# Scaling-Law Analysis of SignSGD: From Feature-Space Linear Regression to LLM Pre-training

**Binghui Li**[1]*, **Jianan Wang**[2]*, **Jinbo Wang**[2]*, **Lean Wang**[3]*, **Zilin Wang**[2]*, **Lei Wu**[1,2,4]†
[1]Center for Machine Learning Research, Peking University
[2]School of Mathematical Sciences, Peking University
[3]School of Computer Science, Peking University [4]AI for Science Institute, Beijing
{libinghui,lean}@pku.edu.cn,leiwu@math.pku.edu.cn
{jiananwang25,wangjinbo,wangzilin}@stu.pku.edu.cn

## Abstract

Despite their widespread use in deep learning, the mechanisms underlying the effectiveness of adaptive gradient methods in large-scale training remain poorly understood. In this work, we provide a scaling-law analysis of SignSGD, a minimal yet expressive optimizer that captures the core coordinate-wise adaptivity shared by more sophisticated adaptive methods. We consider feature-space linear regression with power-law spectra, which allows us to precisely characterize the training dynamics of SignSGD. Specifically, we derive explicit scaling laws for SignSGD that accurately describe the loss dynamics. By further analyzing the data-limited regime, we characterize the phase diagram of SignSGD training and quantify the superiority of SignSGD in data scaling. We also show that SignSGD admits a substantially larger critical batch size than SGD, which gives SignSGD more benefits from large-batch training. Finally, we systematically validate our theoretical predictions through large-scale LLM pre-training experiments, demonstrating that the scaling laws uncovered here extend beyond the controlled model and are predictive of practical training behavior.

## 1 Introduction

Adaptive optimizers, such as Adam (Kingma, 2014), SignSGD (Bernstein et al., 2018), Lion (Chen et al., 2023), and Muon (Jordan et al., 2024; Liu et al., 2025), have demonstrated remarkable empirical success in training large-scale deep learning models. However, the mechanisms underlying their effectiveness in large-scale training remain poorly understood. Most existing theoretical work analyzes adaptive methods through convergence guarantees within the classical stochastic optimization framework (Bernstein et al., 2018; Reddi et al., 2019; Chen et al., 2018; Défossez et al., 2020). These analyses typically rely on highly general assumptions on the objective function and gradient noise, under which adaptive gradient methods perform similarly to vanilla stochastic gradient descent (SGD). As a result, such convergence guarantees cannot explain the empirical advantages of adaptive methods over SGD. More fundamentally, modern large-scale training is governed by the **scaling properties** of these optimizers—namely, how training dynamics vary with the learning rate and batch size, and how they depend on the available data and compute budget. In LLM pre-training, for example, optimizers are routinely configured using data and parameter scaling laws (Kaplan et al., 2020; Yang et al., 2021; Hoffmann et al., 2022; DeepSeek-AI et al., 2024; Li et al., 2025b), as well as hyperparameter scaling rules (Granziol et al., 2022; Malladi et al., 2022), in order to achieve favorable compute efficiency. These scaling properties play a central role in practice, yet they lie largely outside the scope of classical stochastic optimization theory. Addressing this gap motivates us to move beyond classical convergence analysis toward a scaling perspective. As a starting point, we provide a scaling analysis of *SignSGD*:

$$\boldsymbol{\theta}_{k+1} = \boldsymbol{\theta}_k - \eta_k \, \mathrm{sgn}(\mathbf{g}_k),$$

---

*Equal contribution, alphabetical order. †Corresponding author.

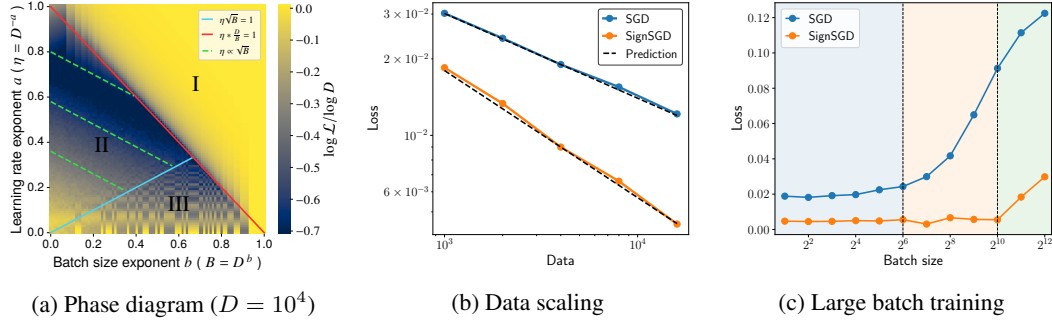

(a) Phase diagram ($D = 10^4$)  (b) Data scaling  (c) Large batch training

Figure 1: **(a)** Phase diagram of SignSGD dynamics under a fixed data budget $D$, with two axes denoting batch size $B = D^b$ and learning rate $\eta = D^{-a}$. Three regimes arise: **burn-in regime, scaling regime and unstable regime**. **(b) Loss versus data** for SignSGD and SGD together with our scaling law predictions. There exists an explicit gap in the scaling efficiency between the optimizers. Solid curves report the final loss (averaged over 400 runs), and dashed lines show theoretical predictions. **(c) Loss versus batch size** at a fixed data budget. As batch size increases, the loss of SGD diverges rapidly, whereas SignSGD remains stable over a wide range of batch sizes. Background colors indicate stable (blue), SGD-unstable (orange), and fully unstable (green) regimes.

where $\boldsymbol{\theta}_k$ denotes the model parameters and $\mathbf{g}_k$ is the stochastic gradient. We focus on this optimizer for two reasons. First, it serves as a minimal yet expressive example for isolating scaling mechanisms shared with more sophisticated adaptive optimizers, such as RMSprop (Hinton et al., 2012) and Adam (Balles & Hennig, 2018; Peng et al., 2025). Second, SignSGD is practically relevant in its own right (Bernstein et al., 2018), and recent work shows that, with a simple momentum variant, it can achieve performance comparable to Adam in LLM pre-training (Zhao et al., 2025; Semenov et al., 2025).

To facilitate a clean and precise analysis, we study SignSGD in a controlled and analytically tractable setting: feature-space linear regression with power-law spectra. While stylized, it captures essential aspects of modern learning dynamics (Li et al., 2025a) and allows us to isolate scaling mechanisms that are difficult to disentangle in more complex models. Notably, such settings have been widely used in previous scaling analyses of vanilla SGD (Lin et al., 2024; Bahri et al., 2024; Bordelon et al., 2025; Li et al., 2025a; Wang et al., 2026).

## 2 SETUP AND MAIN RESULTS

**Notation.** Throughout the paper, we use $\asymp$ to denote equivalence up to a constant factor, and $\lesssim$ (resp. $\gtrsim$) indicates inequality up to a constant factor.

**Feature-space linear regression.** We consider a supervised learning setting in which data points $(\mathbf{x}, y)$ are generated as follows: $\mathbf{x} \sim \mathcal{D}$ and $y = \langle \boldsymbol{\phi}(\mathbf{x}), \boldsymbol{\theta}^* \rangle + \epsilon$ with $\epsilon \sim \mathcal{N}(0, \sigma^2)$, where $\boldsymbol{\phi}(\mathbf{x}) \in \mathbb{R}^d$ denotes the feature representation and $\boldsymbol{\theta}^* \in \mathbb{R}^d$ is the unknown target parameter. We assume Gaussian features $\boldsymbol{\phi}(\mathbf{x}) \sim \mathcal{N}(\mathbf{0}, \mathbf{H})$, where $\mathbf{H} := \mathbb{E}_{\mathbf{x} \sim \mathcal{D}}[\boldsymbol{\phi}(\mathbf{x})\boldsymbol{\phi}(\mathbf{x})^\top]$ is the *feature covariance* matrix. We learn using a linear student model $f(\mathbf{x}; \boldsymbol{\theta}) = \langle \boldsymbol{\phi}(\mathbf{x}), \boldsymbol{\theta} \rangle$ and consider the population risk $\mathcal{R}(\boldsymbol{\theta}) = \frac{1}{2} \mathbb{E}_{\mathbf{x},y}\left[(\langle \boldsymbol{\phi}(\mathbf{x}), \boldsymbol{\theta} \rangle - y)^2\right] = \mathcal{L}(\boldsymbol{\theta}) + \frac{\sigma^2}{2}$, with $\mathcal{L}(\boldsymbol{\theta}) = \frac{1}{2}(\boldsymbol{\theta} - \boldsymbol{\theta}^*)^\top \mathbf{H} (\boldsymbol{\theta} - \boldsymbol{\theta}^*)$. Here $\mathcal{L}(\boldsymbol{\theta})$ is the excess risk. We consider minimizing $\mathcal{R}(\boldsymbol{\theta})$ using **SignSGD**: $\boldsymbol{\theta}_{k+1} = \boldsymbol{\theta}_k - \eta \, \mathrm{sgn}(\nabla \mathcal{R}(\boldsymbol{\theta}_k) + \boldsymbol{\xi}_k)$, with $\boldsymbol{\theta}_0 = \mathbf{0}$, where $\eta > 0$ denotes the learning rate and $\boldsymbol{\xi}_k$ represents the mini-batch gradient noise. For tratability, we model the gradient noise as i.i.d. Gaussian noise, i.e., $\boldsymbol{\xi}_k \sim \mathcal{N}(\mathbf{0}, \mathbf{H}/B)$. Here $B$ is the batch size, the covariance matrix of gradient noise for batch size is aligned with Hessian. The Hessian-aligned noise structure is consistent with (Zhu et al., 2018; Damian et al., 2021; Wu et al., 2022) and is verifiable in the noise-dominated regime $\sigma \gtrsim 1$ (see Appendix C.2).

**Assumption 2.1** (Diagonalized covariance with power-law). $\mathbf{H} = \mathrm{diag}\{\lambda_1, \lambda_2, \ldots, \lambda_d\}$ with (i) (Capacity condition) $\lambda_j \asymp j^{-\beta}$ for some $\beta > 1$ and, (ii) (Source condition) $\lambda_j |\theta_j^*|^2 \asymp j^{-\alpha}$.

The diagonal Hessian assumption is motivated by empirically observed approximately block-diagonal Hessian structure (Zhang et al., 2024c;d; Wang et al., 2025) and should be viewed as an idealized limit that captures the essential properties of SignSGD in practical regimes where blockwise decoupling

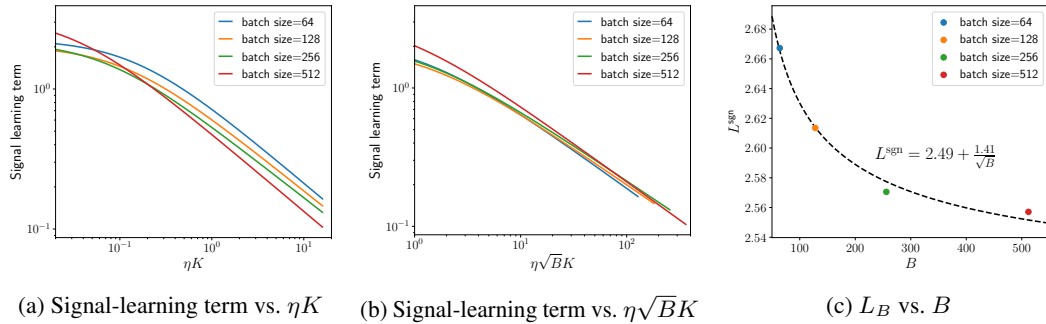

(a) Signal-learning term vs. $\eta K$     (b) Signal-learning term vs. $\eta\sqrt{B}K$     (c) $L_B$ vs. $B$

Figure 2: **Batch size dependence in SignSGD scaling laws**. We fit the loss trajectory using the scaling law, and decompose the total into the signal learning term and the irreducible-plus-noise term $L_B$. **(a)** Signal learning term against intrinsic time $\eta K$. **(b)** Signal learning term against rescaled intrinsic time $\eta K\sqrt{B}$. **(c)** $L_B$ against batch size $B$, with coefficient of determination $R^2 = 0.9891$ between $L^{\text{sgn}}$ and $1/\sqrt{B}$.

holds approximately. The capacity and source conditions are standard in the study of kernel methods and are recently adopted in analysis of scaling laws, with $s := (\alpha - 1)/\beta$ being the **relative difficulty**. See Appendix C.1.

## 3   SCALING LAW FOR THE LOSS DYNAMICS OF SIGNSGD

In this section, we present our main results of this paper. We begin with the following theorem, which establishes the optimization scaling law for SignSGD.

**Theorem 3.1** (Informal version, scaling laws for SignSGD). *Suppose $\alpha \geq \beta > 2$ and $d \gg 1$, for any step $k$, we have*

$$\mathbb{E}[\mathcal{L}(\boldsymbol{\theta}_k)] \approx \begin{cases} (1 - \eta k)^2, & \eta k \lesssim 1 & \textit{(Burn-in regime)}, \\ (\eta\sqrt{B}\,k)^{-2s} + \eta/\sqrt{B}, & \eta k \gtrsim 1 \textit{ and } \eta\sqrt{B} \lesssim 1 & \textit{(Scaling regime)}, \\ (\eta\sqrt{B}\,k)^{-2s} + \eta^2, & \eta k \gtrsim 1 \textit{ and } \eta\sqrt{B} \gtrsim 1 & \textit{(Unstable regime)}. \end{cases} \tag{1}$$

Here, we focus on an easy learning regime where we assume $\alpha \geq \beta > 2$ for simplicity. The full version of Theorem 3.1 and detailed proof are deferred to Appendix F. Theorem 3.1 demonstrates that, the full loss dynamics of SignSGD exhibits a scaling law.

This scaling law of SignSGD further enables a clean analysis in the data-limited regime; the full derivation is given in Appendix B.3.

- **Phase diagram of hyperparameter scaling.** Under a fixed data budget $D$, we characterize how the final loss of SignSGD varies with learning rate $\eta$ and batch size $B$, and get a phase diagram in Figure 1a, from which we observe a sharp phase transition: a burn-in regime where the loss stays nearly constant, a predictable scaling regime where performance depends primarily on the effective learning rate $\eta/\sqrt{B}$, and an unstable regime where oscillation effects dominate.

- **Improved data scaling efficiency.** We investigate how the excess risk scales with data budget $D$: we show that SignSGD achieves a strictly faster data scaling law than vanilla SGD, which is consistently supported by our linear regression experiments in Figure 1b.

- **Effect of large batch training.** We investigate how increasing batch size affects training performance: our analysis predicts that as the batch size increases, the loss of SGD diverges rapidly, whereas SignSGD remains stable over a wide range of batch sizes. See Figure 1c for the experimental result.

## 4 LANGUAGE MODEL EXPERIMENTS

### 4.1 TWO DISTINCT BATCH-SIZE EFFECTS UNDERLYING SIGNSGD

Our scaling-law analysis of SignSGD predicts that increasing the batch size plays two distinct roles in training dynamics. First, in the signal-learning term, the loss decays as $(\eta\sqrt{B}\,k)^{-2s}$, indicating that the effective intrinsic time is rescaled from $T = \eta k$ to $\sqrt{B}\,\eta k$. As a result, under SignSGD, increasing the batch size accelerates signal learning by a factor of $\sqrt{B}$. Second, the noise-accumulation term in SignSGD scales as $\eta/\sqrt{B}$, in contrast to the $\eta/B$ scaling observed in vanilla SGD.

We now empirically verify these two predicted roles of batch size in LLM pre-training. Specifically, we collect loss curves for batch sizes $B \in \{64, 128, 256, 512\}$ and fit curves using previously established SGD scaling law (Lin et al., 2024; Li et al., 2025a) and our proposed SignSGD scaling law:

$$L_{\text{SGD}}(k, B) \approx \underbrace{\underbrace{L_0^{\text{sgd}}}_{\text{irreducible}} + \underbrace{\frac{C^{\text{sgd}}}{B}}_{\text{noise}}}_{\text{loss plateau } L_B} + \underbrace{\frac{1}{(\eta k)^{\alpha}}}_{\text{signal learning}} \quad , \quad L_{\text{SignSGD}}(k, B) \approx \underbrace{\underbrace{L_0^{\text{sgn}}}_{\text{irreducible}} + \underbrace{\frac{C^{\text{sgn}}}{\sqrt{B}}}_{\text{noise}}}_{\text{loss plateau } L_B} + \underbrace{\frac{1}{(\eta\sqrt{B}k)^{\beta}}}_{\text{signal learning}}.$$

In both cases, the third term captures the signal-learning dynamics. Importantly, we do not impose any prior assumption on how the loss plateau $L_B$ depends on the batch size $B$, allowing the effect of batch size on noise accumulation to be learned directly from the data.

The results are reported in Figure 2. Figure 2(a-b) shows that the signal-learning dynamics collapses when plotted against the rescaled intrinsic time $\sqrt{B}\,\eta k$, rather than the original intrinsic time $\eta k$. Figure 2c illustrates the effect of batch size on the noise-accumulation term. Importantly, no prior assumption is made on how the loss plateau $L_B$ depends on the batch size $B$ during fitting. Despite this, the inferred values of $L_B$ consistently follow a $1/\sqrt{B}$ scaling, in contrast to the $1/B$ behavior characteristic of vanilla SGD. Together, these results demonstrate that the two distinct effects of increasing batch size in SignSGD—predicted by our theory—also manifest in LLM pre-training.

### 4.2 SIGNSGD SCALING LAW ACCURATELY PREDICTS LOSS DYNAMICS

We first examine whether our SignSGD scaling law (2) can fit and predict the loss curves of LLM pre-training, in particular capturing the influence of batch size. We train this model using different batch sizes and collect the loss curves. We then use these curves to fit the SignSGD scaling law:

$$F(k, B) = L_0 + C_1(C_2 + \sqrt{B}\,k\eta)^{-a} + C_3\eta/\sqrt{B}, \tag{2}$$

where $\{L_0, C_1, C_2, C_3, a\}$ are fitted parameters. Here, the three terms in (2) represent the irreducible loss, signal learning, and noise accumulation, respectively. Specifically, we train models with batch sizes $B \in \{64, 128, 256\}$ using a constant learning rate $\eta = 5 \times 10^{-4}$. The fitting is performed using the initial $25\%$ of the loss curve for $B = 64$ and the initial $50\%$ for $B = 128$. We then use the fitted law to predict the remaining training steps for $B = 64$ and $B = 128$, as well as the entire loss curve for the unseen batch size $B = 256$.

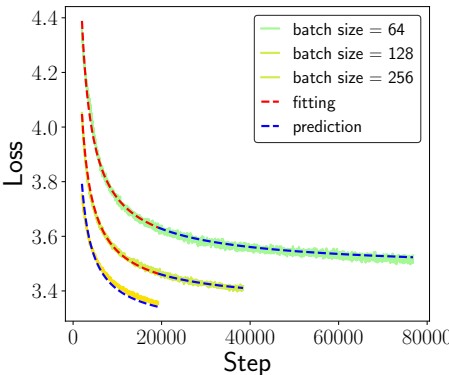

Figure 3: SignSGD scaling law accurately predicts the loss dynamics of LLM pre-training.

As shown in Figure 3, the proposed scaling law accurately matches the empirical loss over the fitting interval and generalizes well to both *longer training horizons* and *unseen batch sizes*. In particular, for batch size $B = 64$, the prediction remains accurate for up to $4\times$ more training steps.

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

# Appendix

## A  SCIENCE OF DL IMPROVEMENT

### A.1  WHAT MODEL ARE WE TARGETING?

We target **SignSGD** as a scientific object of study, a minimal yet expressive optimizer: it isolates the *sign-based coordinate-wise adaptivity* that is widely implicated in the empirical success of modern adaptive optimizers, while remaining analytically tractable.

In particular, we focus on its **scaling behavior**—how optimization dynamics, stability, and data efficiency depend on the learning rate $\eta$, batch size $B$, and a finite data budget $D$.

### A.2  HOW DO OUR RESULTS CONTRIBUTE TO UNDERSTANDING THESE MODELS?

We establish a **functional scaling law** for SignSGD to quantitatively describe the scaling behavior of sign-based (and, more broadly, adaptive) optimizers in large-scale training:

- **Optimization behavior of SignSGD.** We characterize SignSGD across the joint scaling of $(\eta, B)$ via a *phase diagram* with three regimes—*burn-in*, *scaling*, and *unstable*—which provides a principled map for hyperparameter tuning.
- **Why SignSGD (and other adaptive optimizers) outperforms SGD:** We show how sign-based coordinate-wise adaptivity can outperform vanilla SGD through an implicit preconditioning effect, yielding improved scaling exponents and clearer advantages in the large-batch regime.
- **Fit and prediction of loss curve.** Our proposed scaling law accurately matches the empirical loss curves of SignSGD over the fitting interval and generalizes well to both *longer training horizons* and *unseen batch sizes*.

### A.3  HOW DO WE EXPECT OUR SUBMISSION TO INFLUENCE FUTURE WORK?

- **Scaling analysis of optimizers.** This work motivates a *scaling* perspective on optimization—complementary to classical convergence-rate analyses—as a more faithful lens for understanding optimizer behavior in large-scale training.
- **Hyperparameter tuning.** Our theoretical analysis suggests principled design rules for hyperparameter tuning of optimizers, bridging theory and practical recipe design.
- **Design principles beyond SignSGD.** Identifying the benefits of SignSGD motivates new optimizer designs (and analyses) that explicitly target the same mechanism-level sources of data-efficiency gains.

## B  CORE THEORETICAL RESULTS

### B.1  PRELIMINARIES

**Notation.** Throughout the paper, we use $\eqsim$ to denote equivalence up to a constant factor, and $\lesssim$ (resp. $\gtrsim$) indicates inequality up to a constant factor.

**Feature-space linear regression.**   We consider a supervised learning setting in which data points $(\mathbf{x}, y)$ are generated as follows: $\mathbf{x} \sim \mathcal{D}$ and $y = \langle \boldsymbol{\phi}(\mathbf{x}), \boldsymbol{\theta}^* \rangle + \epsilon$ with $\epsilon \sim \mathcal{N}(0, \sigma^2)$, where $\boldsymbol{\phi}(\mathbf{x}) \in \mathbb{R}^d$ denotes the feature representation and $\boldsymbol{\theta}^* \in \mathbb{R}^d$ is the unknown target parameter. We assume Gaussian features $\boldsymbol{\phi}(\mathbf{x}) \sim \mathcal{N}(\mathbf{0}, \mathbf{H})$, where $\mathbf{H} := \mathbb{E}_{\mathbf{x} \sim \mathcal{D}}[\boldsymbol{\phi}(\mathbf{x})\boldsymbol{\phi}(\mathbf{x})^\top]$ is the *feature covariance* matrix. We learn using a linear student model $f(\mathbf{x}; \boldsymbol{\theta}) = \langle \boldsymbol{\phi}(\mathbf{x}), \boldsymbol{\theta} \rangle$ and consider the population risk

$$\mathcal{R}(\boldsymbol{\theta}) = \tfrac{1}{2} \, \mathbb{E}_{\mathbf{x}, y}\big[ (\langle \boldsymbol{\phi}(\mathbf{x}), \boldsymbol{\theta} \rangle - y)^2 \big]. \tag{3}$$

A direct calculation yields $\mathcal{R}(\boldsymbol{\theta}) = \mathcal{L}(\boldsymbol{\theta}) + \frac{\sigma^2}{2}$, with $\mathcal{L}(\boldsymbol{\theta}) = \frac{1}{2}(\boldsymbol{\theta} - \boldsymbol{\theta}^*)^\top \mathbf{H} (\boldsymbol{\theta} - \boldsymbol{\theta}^*)$. Here $\mathcal{L}(\boldsymbol{\theta})$ is the excess risk. We consider minimizing $\mathcal{R}(\boldsymbol{\theta})$ using **SignSGD**:

$$\boldsymbol{\theta}_{k+1} = \boldsymbol{\theta}_k - \eta \, \mathrm{sgn}(\nabla \mathcal{R}(\boldsymbol{\theta}_k) + \boldsymbol{\xi}_k), \qquad \boldsymbol{\theta}_0 = \mathbf{0}, \tag{4}$$

where $\eta > 0$ denotes the learning rate and $\boldsymbol{\xi}_k$ represents the mini-batch gradient noise. We model the gradient noise as i.i.d. Gaussian noise, i.e., $\boldsymbol{\xi}_k \sim \mathcal{N}(\mathbf{0}, \boldsymbol{\Sigma}/B)$. Here $B$ is the batch size and $\boldsymbol{\Sigma}$ denotes the covariance matrix of gradient noise for batch size 1. We further assume:

**Assumption B.1** (Hessian-aligned noise). The gradient noise covariance is aligned with Hessian: $\boldsymbol{\Sigma} \asymp \mathbf{H}$.

This assumption implies an anisotropic, Hessian-aligned noise structure, in which the magnitude of gradient noise along each direction is proportional to the local curvature of the loss landscape. In the feature-space linear regression setting, this structure can be verified explicitly in the noise-dominated regime $\sigma \gtrsim 1$ (see Appendix C.2). Similar alignment between gradient noise and curvature can be validated in deep neural networks (Zhu et al., 2018; Blanc et al., 2020; Ziyin et al., 2021; Damian et al., 2021; Li et al., 2022; Wu et al., 2022; Mori et al., 2021; Wang & Wu, 2023).

**Assumption B.2** (Diagonalized feature covariance). $\mathbf{H} = \text{diag}\{\lambda_1, \lambda_2, \ldots, \lambda_d\}$ with $\lambda_1 \geq \lambda_2 \geq \cdots \geq \lambda_d > 0$.

Under this assumption, the excess risk decomposes as $\mathcal{L}(\boldsymbol{\theta}) = \frac{1}{2} \sum_{j=1}^{d} \lambda_j (\theta_j - \theta_j^*)^2$. This diagonal structure decouples the learning dynamics across coordinates, enabling a precise analysis of SignSGD. In particular, it allows us to identify regimes in which the sign operation leads to qualitatively different—and in certain aspects more favorable—scaling behavior than that of SGD. Moreover, empirical studies suggest that the Hessian of deep neural networks often exhibits an approximately block-diagonal structure (Zhang et al., 2024c; Malinovskii et al., 2024; Zhang et al., 2024d; Kunstner et al., 2024; Zhang et al., 2024b; Ormaniec et al., 2024; Dong et al., 2025; Wang et al., 2025). Our analysis naturally extends to such block-diagonal settings, and Assumption B.2 can be viewed as an idealized limit that captures the essential properties of SignSGD in practical regimes where blockwise decoupling holds approximately.

**Assumption B.3** (Power-law structures). The following conditions hold:

(i) **(Capacity condition)** $\lambda_j \asymp j^{-\beta}$ for some $\beta > 1$.

(ii) **(Source condition)** $\lambda_j |\theta_j^*|^2 \asymp j^{-\alpha}$ for some $\alpha > 1$.

The **capacity exponent** $\beta$ controls the decay rate of the feature spectrum. The source condition quantifies the alignment of the target function and the feature space. We define $s := (\alpha - 1)/\beta$, referred to as the **relative difficulty**. Then, smaller $s$ corresponds to challenging learning problems, in which a larger fraction of the target energy is concentrated in high-frequency components.

Assumption B.3 is standard in the analysis of kernel methods and has recently been widely adopted in the analysis of scaling laws (Paquette et al., 2024; Lin et al., 2024; Bordelon et al., 2025; Li et al., 2025a). A more detailed interpretation of this setup is provided in Appendix C.1.

**Scaling laws for SGD.** As a baseline for understanding SignSGD, we recall the scaling behavior of vanilla SGD:

$$\boldsymbol{\theta}_{k+1} = \boldsymbol{\theta}_k - \eta \big( \nabla \mathcal{R}(\boldsymbol{\theta}_k) + \boldsymbol{\xi}_k \big), \tag{5}$$

with zero initialization $\boldsymbol{\theta}_0 = \mathbf{0}$. Under the structural assumptions introduced above, the loss dynamics of SGD exhibits the following scaling law: for $\eta \lesssim 1$ and $1 \lesssim \eta k \lesssim d^\beta$

$$\mathbb{E}[\mathcal{L}(\boldsymbol{\theta}_k)] \asymp \frac{1}{(\eta k)^s} + \frac{\eta}{B}. \tag{6}$$

This scaling law has been established in prior work (Lin et al., 2024; Li et al., 2025a), and for completeness we provide a self-contained derivation in Appendix E.

Equation (6) reveals a signal–noise decomposition of the loss dynamics of SGD. The *signal-learning term* $(\eta k)^{-s}$ decays as a power law in the intrinsic training time $t = \eta k$, with an exponent determined by the relative difficulty $s$. The *noise-accumulation term $\eta/B$* scales linearly with the learning rate and inversely with the batch size, capturing the contribution of stochastic gradient noise.

## B.2 SCALING LAW FOR THE LOSS DYNAMICS OF SIGNSGD

In this section, we present our main results of this paper. We begin with the following theorem, which establishes the optimization scaling law for SignSGD.

**Theorem B.4** (Informal version, scaling laws for SignSGD). *Suppose $\alpha \geq \beta > 2$ and $d \gg 1$, for any step $k$, we have*

- *(Burn-in regime) When $\eta k \lesssim 1$,*

$$\mathbb{E}[\mathcal{L}(\boldsymbol{\theta}_k)] \approx (1 - \eta k)^2;$$

- *(Scaling regime) When $\eta k \gtrsim 1$ and $\eta\sqrt{B} \lesssim 1$,*

$$\mathbb{E}[\mathcal{L}(\boldsymbol{\theta}_k)] \approx \frac{1}{(\eta\sqrt{B}k)^{2s}} + \frac{\eta}{\sqrt{B}};$$

- *(Unstable regime) When $\eta k \gtrsim 1$ and $\eta\sqrt{B} \gtrsim 1$,*

$$\mathbb{E}[\mathcal{L}(\boldsymbol{\theta}_k)] \approx \frac{1}{(\eta\sqrt{B}k)^{2s}} + \eta^2.$$

Here, we focus on an easy learning regime where we assume $\alpha \geq \beta > 2$ for simplicity. The full version of Theorem B.4 and detailed proof are deferred to Appendix F. Theorem B.4 demonstrates that, the full loss dynamics of SignSGD exhibits a scaling law.

**Comparison with SGD under scaling regime.** Our result reveals, under the scaling regime, the following two insights into how the sign operator impacts scaling behavior.

- **Improved scaling efficiency from preconditioning.** Theorem B.4 shows that the optimization scaling rate of SignSGD is double that of SGD. Intuitively, while the convergence along each coordinate $j \in [d]$ is governed by $\lambda_j$, the global convergence behavior is dictated by the spectral decay of the Hessian. A faster decay induces strong anisotropy, causing the optimization to be bottlenecked by the slow modes corresponding to small eigenvalues. For SGD, given a task difficulty parameter $\alpha$ and a power-law Hessian spectrum $\lambda_j \sim j^{-\beta}$, the final scaling efficiency is $s = \frac{\alpha-1}{\beta}$. In contrast, SignSGD can implicitly leverage the Hessian-aligned noise structure (assuming $\boldsymbol{\Sigma} = \mathbf{H}$) to achieve preconditioning, which effectively transforms the spectrum as $\mathbf{H} \to \boldsymbol{\Sigma}^{-1/2}\mathbf{H} = \mathbf{H}^{1/2}$ (mapping eigenvalues $\lambda_j \to \sqrt{\lambda_j}$). Consequently, the decay rate flattens from $\beta$ to $\beta/2$, doubling the scaling efficiency, i.e., $s \to 2s$. See details in the following proof intuition.

- **Accelerating signal learning with large batch size.** For a fixed learning rate $\eta$ and number of steps $k$, increasing $B$ accelerates signal learning in SignSGD through an additional $\sqrt{B}$ factor, whereas SGD has no such acceleration.

### B.2.1 PROOF INTUITION

Under the assumption of a diagonal feature covariance, we analyze the training dynamics coordinate-wise. Consider the one-dimensional SignSGD update: $\theta_{k+1} = \theta_k - \eta\,\mathrm{sgn}(g_k + \xi_k)$, where we assume $\theta_0 > 0$, the target $\theta^* = 0$, $g_k = \lambda\theta_k$ denotes the full-batch gradient associated with curvature $\lambda$, and $\xi_k \sim \mathcal{N}(0, \sigma^2)$ represents the Hessian-aligned label noise $\sigma^2 = \lambda/B$.

**Burn-in regime.** In the early stage of training, the stochastic gradient signal significantly dominates the noise component $|g_k| \gg \sigma$. Consequently, we have

$$\mathrm{sgn}(g_k + \xi_k) \approx \mathrm{sgn}(g_k) = \mathrm{sgn}(\theta_k).$$

In this regime, the SignSGD dynamics approximates deterministic SignGD, evolving as: $\theta_{k+1} \approx \theta_k - \eta$. Thus, the loss decreases as a quadratic function over iterations, i.e., $\theta_k^2 \approx (\theta_0 - \eta k)^2$. Therefore, the excess risk satisfies

$$\mathcal{L}(\boldsymbol{\theta}_k) \approx \sum_{j=1}^{d} \lambda_j(\theta_0^{(j)} - \eta k)^2 \approx (1 - \eta k)^2.$$

**Scaling regime.** When $\eta k \gtrsim 1$, the parameter enters the vicinity of the optimum, where the noise dominates the gradient signal (i.e., $|g_k| \ll \sigma$). In this regime, the expected update direction can be linearly approximated as $\mathbb{E}[\mathrm{sgn}(g_k + \xi_k)] \approx g_k/\sigma$, which induces stochastic linear dynamics:

$$\theta_{k+1} \approx (1 - \eta\lambda/\sigma)\theta_k + \eta\zeta_k$$
$$= (1 - \eta\sqrt{\lambda B})\theta_k + \eta\zeta_k,$$

where $\zeta_k$ is a zero-mean random variable with problem-independent variance $\mathrm{Var}[\zeta_k] \approx 1$. The stability condition for this system is given by $\eta\sqrt{\lambda B} < 2$. Assuming $\eta\sqrt{B} \lesssim 1$, the parameters remain within the linear stability regime across all coordinates. This implies that the SignSGD dynamics behaves similarly to SGD, characterized by rapid signal learning with normalized stochastic noise. The resulting variance dynamics satisfies

$$\mathbb{E}[\theta_k^2] \approx e^{-2\eta\sqrt{\lambda B}k}|\theta_0|^2 + \eta/\sqrt{\lambda B}.$$

Incorporating the effect of task accumulation ($d \gg 1$), we ultimately derive the loss scaling law:

$$\mathcal{L}(\boldsymbol{\theta}_k) \approx \sum_{j=1}^{d} \lambda_j (e^{-2\eta\sqrt{\lambda_j B}k}|\theta_0^{(j)}|^2 + \eta/\sqrt{\lambda_j B})$$

$$\approx (\eta\sqrt{B}k)^{-2s} + \eta/\sqrt{B}.$$

Notably, our results demonstrate that SignSGD achieves improved scaling efficiency compared to vanilla SGD (improving the rate from $s$ to $2s$). This acceleration arises from the preconditioning effect of the sign operator, under the assumptions of Hessian-aligned noise and a diagonal Hessian.

**Unstable regime.** When $\eta k \gtrsim 1$ and $\eta\sqrt{B} \gtrsim 1$, the stability condition is violated along the sharpest directions (specifically, the top $j^* \approx (\eta\sqrt{B})^{2/\beta}$ coordinates), preventing them from stabilizing in the scaling regime. Instead, these parameters oscillate with a magnitude of order $\mathcal{O}(\eta)$. Consequently, the excess risk can be approximated by decomposing it into unstable and stable components:

$$\mathcal{L}(\boldsymbol{\theta}_k) \approx \sum_{j \leq j^*} \lambda_j \eta^2 + \sum_{j > j^*} \lambda_j (e^{-2\eta\sqrt{\lambda_j B}k} + \eta/\sqrt{B})$$

$$\approx \eta^2 + (\eta\sqrt{B}k)^{-2s} + \eta/\sqrt{B}$$

$$\approx (\eta\sqrt{B}k)^{-2s} + \eta^2$$

### B.3    IMPLICATIONS FOR THE DATA-LIMITED REGIME

Theorem B.4 establishes a scaling law characterizing the loss dynamics of SignSGD. We now analyze its implications for training in a data-limited regime.

Specifically, let $D = KB$ denote the total data budget, where $K$ is the number of training steps and $B$ is the batch size. We use $\mathcal{E}_D(\eta, B)$ to denote the expected excess risk achieved with learning rate $\eta$ and batch size $B$.

### B.3.1    PHASE DIAGRAM OF HYPERPARAMETER SCALING

In the setting of large-scale training, scaling hyperparameters in proportion to the data budget is crucial for improving both training efficiency and effectiveness.

**The burn-in regime is under-trained.** When $\eta K = \eta\frac{D}{B} \lesssim 1$, the loss scales as

$$\mathcal{E}_D(\eta, B) \approx (1 - \eta K)^2 \approx 1,$$

implying that the final loss remains at a constant level due to insufficient training.

**The scaling regime exhibits predictable scaling behavior.** When $\eta K = \eta D/B \gtrsim 1$ and $\eta\sqrt{B} \lesssim 1$, for a fixed budget $D$, the final loss scales as

$$\mathcal{E}_D(\eta, B) \approx (\eta\sqrt{B}K)^{-2s} + \eta/\sqrt{B}$$

$$= (D\eta/\sqrt{B})^{-2s} + \eta/\sqrt{B}.$$

Notably, this expression depends solely on the effective learning rate $\gamma = \eta/\sqrt{B}$. Consequently, to accelerate training, one can scale the learning rate and batch size simultaneously according to $\eta \propto \sqrt{B}$ while maintaining the same final loss. This strategy is known as the square root scaling rule (Granziol et al., 2022; Malladi et al., 2022; Li et al., 2024), which demonstrates a predictable scaling behavior for hyperparameter tuning.

**The unstable regime is suboptimal and uncontrollable.** When $\eta K \gtrsim 1$ and $\eta\sqrt{B} \gtrsim 1$, the square root scaling rule breaks down; consequently, there exists no simple scaling strategy to maintain performance while increasing training efficiency. Since $\eta^2 \gtrsim \eta/\sqrt{B}$, the final loss is dominated by oscillation terms, implying

$$\mathcal{E}_D(\eta, B) \gtrsim (D\eta/\sqrt{B})^{-2s} + \eta/\sqrt{B},$$

which characterizes a suboptimal regime.

**Empirical validation.** To verify our theoretical phase diagram, we conduct numerical experiments on feature-space linear regression. We parameterize the scaling of the learning rate and batch size as power laws of the data budget $D$, specifically $\eta = D^{-a}$ and $B = D^b$. The results, presented in Figure 1a, demonstrate a sharp phase transition that aligns closely with our theoretical predictions. Region I corresponds to the burn-in regime; Region II represents the scaling regime, characterized by the scaling contour $\eta \propto \sqrt{B}$; and Region III denotes the unstable regime, where the final loss exhibits chaotic oscillations.

### B.3.2 IMPROVED DATA SCALING EFFICIENCY

We now focus on the scaling regime to discuss the optimal data scaling law for SignSGD. Given a fixed data budget $D$, and assuming the conditions $\eta K \gtrsim 1$ and $\eta\sqrt{B} \lesssim 1$ hold, we derive the following optimal effective learning rate and the corresponding excess risk:

$$\gamma_D^{*,\mathrm{sgn}} \asymp D^{-\frac{2s}{2s+1}} \quad \text{and} \quad \mathcal{E}_D^{*,\mathrm{sgn}} \asymp D^{-\frac{2s}{2s+1}}.$$

In contrast, SGD exhibits a linear scaling rule. Specifically, the loss scales as $\mathcal{E}_D(\eta, B) = (D(\eta/B))^{-s} + \eta/B$, which is a function solely of the effective learning rate $\gamma = \eta/B$ and implies the following optimal data scaling law:

$$\gamma_D^{*,\mathrm{sgd}} \asymp D^{-\frac{s}{s+1}} \quad \text{and} \quad \mathcal{E}_D^{*,\mathrm{sgd}} \asymp D^{-\frac{s}{s+1}}.$$

These findings highlight the improved data scaling efficiency of SignSGD compared to vanilla SGD. We empirically validate this improvement on feature-space linear regression, with the results presented in Figure 1b. Notably, we extend our theoretical insights to real-world language model pre-training. The results are presented in Figure 6a.

### B.3.3 THE EFFECT OF LARGE BATCH TRAINING

In practice, scaling up the batch size is crucial for maximizing computational throughput on parallel hardware and has become a standard component of large-scale training (McCandlish et al., 2018; Zhang et al., 2024a; Merrill et al., 2025; You et al., 2020; Narayanan et al., 2021). This raises a fundamental theoretical question: how does increasing batch size shape training performance under a fixed data budget? Motivated by this practical consideration, we analyze the behavior of large-batch training in the scaling regime. We use $\mathcal{E}_D^*(B) = \min_{\eta \lesssim 1/\sqrt{B}} \mathcal{E}_D(\eta, B)$ to denote the optimal excess risk for a given data budget $D$ and batch size $B$.

$$\mathcal{E}_D^{*,\mathrm{sgn}}(B) = \begin{cases} D^{-\frac{2s}{2s+1}}, & B \lesssim D^{\frac{2s}{2s+1}}; \\ \left(\frac{D}{B}\right)^{-2s}, & B \gtrsim D^{\frac{2s}{2s+1}}. \end{cases}$$

This result reveals a critical batch size $B_D^{*,\mathrm{sgn}} \asymp D^{\frac{2s}{2s+1}}$, which implies that as long as the batch size does not exceed this threshold (i.e., $B \leq B_D^*$), scaling the learning rate according to the square root rule maintains optimal data efficiency while linearly reducing the total number of training iterations.

In contrast, for vanilla SGD, the stability condition is $\eta \lesssim 1$ (Wu et al., 2018; Cohen et al., 2021; Wu et al., 2022). Under this condition, the excess risk is given by:

$$\mathcal{E}_D^{*,\mathrm{sgd}}(B) = \begin{cases} D^{-\frac{s}{s+1}}, & B \lesssim D^{\frac{s}{s+1}}; \\ \left(\frac{D}{B}\right)^{-s}, & B \gtrsim D^{\frac{s}{s+1}}. \end{cases}$$

This yields a critical batch size $B_D^{*,\text{sgd}} \asymp D^{\frac{s}{s+1}}$, below which increasing the batch size according to the linear scaling rule preserves the optimal data-efficiency rate.

The above analysis highlights two key differences between SignSGD and SGD in large-batch training:

- **Larger critical batch size.** SignSGD admits a substantially larger critical batch size than SGD. In particular,

$$\frac{B_D^{*,\text{sgn}}}{B_D^{*,\text{sgd}}} \asymp D^\nu, \qquad \nu = \frac{s}{(2s+1)(s+1)} > 0,$$

  implying that the gap between the two critical batch sizes grows in a power law with the data budget.

- **Growing advantage in the large-batch regime.** For a fixed data budget $D$, the performance advantage of SignSGD over SGD as the batch size increases exhibits a three-phase behavior: for $B \leq D^{\frac{s}{s+1}}$, the loss gap remains constant; for $D^{\frac{s}{s+1}} \lesssim B \lesssim D^{\frac{2s}{2s+1}}$, the gap grows rapidly, scaling as $B^s$; and for $B \gtrsim D^{\frac{2s}{2s+1}}$, the growth slows but remains increasing with $B$.

Together, these results provide a theoretical explanation for the growing advantage of SignSGD over SGD in large-batch training regimes as the dataset size increases. These offer a potential explanation for why adaptive optimizers outperform SGD in large-batch training settings (Marek et al., 2025; Srećković et al., 2025). We conduct numerical simulations to investigate the effect of large-batch training (see results in Figure 1c). Furthermore, we extend this analysis to large-scale model pre-training experiments, as shown in Figure 6b.

## C  ADDITIONAL BACKGROUND

### C.1  RELATIONSHIP WITH POWER-LAW KERNEL REGRESSION

Our feature-space linear regression setting can be viewed as a finite-dimensional specialization of kernel regression, a classical non-parametric framework that has been extensively studied in the literature (Caponnetto & Vito, 2005; Caponnetto & De Vito, 2007; Spigler et al., 2020; Bordelon et al., 2020; Maloney et al., 2022). In this section, we briefly review the kernel regression setup and the standard *source* and *capacity* conditions under power-law spectra. We then provide an explicit mapping between the parameters in our linear regression model and those in power-law kernel regression, which will be used to interpret our assumptions and results from the kernel-methods viewpoint.

#### C.1.1  KERNEL REGRESSION

In kernel regression, we aim to learn a function from finite samples within a reproducing kernel Hilbert space (RKHS). Let $\mathcal{X}$ be the input space and let $K : \mathcal{X} \times \mathcal{X} \to \mathbb{R}$ be a continuous positive semidefinite kernel. We denote by $\mathcal{H}_K$ the RKHS associated with $K$ defined as the completion of

$$\text{span}\{K(x, \cdot) : x \in \mathcal{X}\},$$

equipped with the unique inner product $\langle \cdot, \cdot \rangle_{\mathcal{H}_K}$ satisfying

$$\langle K(x, \cdot), K(x', \cdot) \rangle_{\mathcal{H}_K} = K(x, x'), \qquad \forall x, x' \in \mathcal{X}.$$

In particular, $\mathcal{H}_K$ has the reproducing property

$$f(x) = \langle f, K(x, \cdot) \rangle_{\mathcal{H}_K}, \qquad \forall f \in \mathcal{H}_K, \ \forall x \in \mathcal{X},$$

and the canonical feature map $\Phi : \mathcal{X} \to \mathcal{H}_K$ defined by $\Phi(x) := K(x, \cdot)$ satisfies

$$K(x, x') = \langle \Phi(x), \Phi(x') \rangle_{\mathcal{H}_K}.$$

We write $\|f\|_{\mathcal{H}_K} := \sqrt{\langle f, f \rangle_{\mathcal{H}_K}}$.

Let $\rho$ be a probability measure on $\mathcal{X} \times \mathbb{R}$ and denote by $\rho_x$ its marginal on $\mathcal{X}$. Let $(x, y) \sim \rho$ and $f$ be an estimator. We measure prediction quality by the squared loss

$$\mathcal{E}(f) := \int_{\mathcal{X} \times \mathbb{R}} (f(x) - y)^2 \, d\rho(x, y).$$

Let $L^2(\rho_x)$ be the space of square-integrable functions on $\mathcal{X}$ w.r.t. $\rho_x$.

**Assumption C.1.** The kernel satisfies

$$\int_{\mathcal{X}} K(x,x)\,d\rho_x(x) < \infty.$$

**Lemma C.2** (Continuous embedding $\mathcal{H}_K \hookrightarrow L^2(\rho_x)$). *Under Assumption C.1, every $f \in \mathcal{H}_K$ belongs to $L^2(\rho_x)$ and the inclusion map $I : \mathcal{H}_K \to L^2(\rho_x)$, $If = f$, is a bounded linear operator. Moreover,*

$$\|If\|_{L^2(\rho_x)}^2 \leq \left( \int_{\mathcal{X}} K(x,x)\,d\rho_x(x) \right) \|f\|_{\mathcal{H}_K}^2, \qquad \forall f \in \mathcal{H}_K.$$

*Proof.* By the reproducing property and Cauchy–Schwarz,

$$|f(x)| = |\langle f, K(x,\cdot)\rangle_{\mathcal{H}_K}| \leq \|f\|_{\mathcal{H}_K} \|K(x,\cdot)\|_{\mathcal{H}_K} = \|f\|_{\mathcal{H}_K} \sqrt{K(x,x)}.$$

Squaring and integrating w.r.t. $\rho_x$ yields the desired result. $\square$

Define the kernel integral operator $T : L^2(\rho_x) \to L^2(\rho_x)$ by

$$(Tg)(x) := \int_{\mathcal{X}} K(x,x')\,g(x')\,d\rho_x(x'), \qquad g \in L^2(\rho_x).$$

Under Assumption C.1, the operator $T : L^2(\rho_x) \to L^2(\rho_x)$ is well-defined and bounded. Moreover, $T$ is positive and self-adjoint, i.e.,

$$\langle g, Tg\rangle_{L^2(\rho_x)} \geq 0, \qquad \langle g, Th\rangle_{L^2(\rho_x)} = \langle Tg, h\rangle_{L^2(\rho_x)}, \qquad \forall g, h \in L^2(\rho_x).$$

In addition, since $K$ is positive semidefinite, the Cauchy–Schwarz inequality in $\mathcal{H}_K$ yields

$$|K(x,x')|^2 \leq K(x,x)\,K(x',x'), \qquad \forall x, x' \in \mathcal{X}.$$

Consequently,

$$\int_{\mathcal{X}}\int_{\mathcal{X}} |K(x,x')|^2\,d\rho_x(x)\,d\rho_x(x') \leq \left( \int_{\mathcal{X}} K(x,x)\,d\rho_x(x) \right)^2 < \infty,$$

which shows that $K \in L^2(\rho_x \otimes \rho_x)$ and hence $T$ is a Hilbert–Schmidt operator on $L^2(\rho_x)$. In particular, $T$ is compact. Therefore, by the spectral theorem, there exist an orthonormal system $\{\phi_j\}_{j \geq 1} \subset L^2(\rho_x)$ and a non-increasing sequence of eigenvalues $\{\mu_j\}_{j \geq 1} \subset [0,\infty)$ with $\mu_j \downarrow 0$. For any $g \in L^2(\rho_x)$, we write the (orthogonal) expansion

$$g = \sum_{j \geq 1} g_j \phi_j + g_\perp, \qquad g_j := \langle g, \phi_j\rangle_{L^2(\rho_x)}, \qquad g_\perp \perp \overline{\mathrm{Ran}(T)}.$$

Then

$$Tg = \sum_{j \geq 1} \mu_j g_j \phi_j.$$

**Source condition.** For any $r \geq 0$, define the fractional power $T^r$ by

$$T^r g := \sum_{j \geq 1} \mu_j^r g_j \phi_j,$$

with domain

$$\mathrm{Dom}(T^r) := \left\{ g \in L^2(\rho_x) : \sum_{j \geq 1} \mu_j^{2r} g_j^2 < \infty \right\}.$$

Let $f_\rho(x) := \mathbb{E}[y \mid x]$ and let $f_{\mathcal{H}}$ denote the $L^2(\rho_x)$-projection of $f_\rho$ onto $\overline{\mathrm{Ran}(T^{1/2})}$ (equivalently, onto the $L^2(\rho_x)$-closure of the RKHS hypothesis space). We impose the following standard regularity assumption.

**Assumption C.3** (Source condition). There exist $r > 0$ and $g \in L^2(\rho_x)$ such that

$$f_{\mathcal{H}} = T^r g, \qquad \text{and} \qquad \|g\|_{L^2(\rho_x)} \leq R$$

for some constant $R > 0$.

In terms of the eigencoordinates $f_{\mathcal{H},j} := \langle f_{\mathcal{H}}, \phi_j \rangle_{L^2(\rho_x)}$, Assumption C.3 is equivalent to

$$\sum_{j \geq 1} \frac{f_{\mathcal{H},j}^2}{\mu_j^{2r}} \leq R^2,$$

or equivalently,

$$f_{\mathcal{H},j} = \mu_j^r g_j, \quad \sum_{j \geq 1} g_j^2 \leq R^2.$$

Thus the source exponent $r$ quantifies how rapidly the target coefficients $\{f_{\mathcal{H},j}\}$ decay relative to the spectrum $\{\mu_j\}$.

**Capacity condition.** To quantify the complexity induced by $(K, \rho_x)$, we impose a standard capacity condition in terms of fractional traces of $T$.

**Assumption C.4** (Capacity condition). There exist $\gamma \in (0, 1]$ and a constant $Q > 0$ such that

$$\mathrm{Tr}(T^\gamma) = \sum_{j \geq 1} \mu_j^\gamma \leq Q.$$

Assumption C.4 is equivalent to $\{\mu_j\}_{j \geq 1} \in \ell^\gamma$, and in particular implies the polynomial upper bound

$$\mu_j \leq Q^{1/\gamma} j^{-1/\gamma}, \qquad \forall j \geq 1.$$

Thus the exponent $\gamma$ uniquely characterizes the decay rate of the spectrum.

### C.1.2 CONNECTION WITH FEATURE-SPACE LINEAR REGRESSION.

We next relate the above source–capacity framework to the feature-space linear regression setup in Section B.1. The goal is to identify the spectral quantities in kernel regression with the corresponding objects in linear regression.

Specialize to $\mathcal{X} = \mathbb{R}^d$ and identify $\rho_x = \mathcal{D}$. Consider the linear kernel

$$K(\mathbf{x}, \mathbf{x}') := \langle \mathbf{x}, \mathbf{x}' \rangle_{\mathbb{R}^d}.$$

Let

$$\mathbf{H} := \mathbb{E}_{\mathbf{x} \sim \mathcal{D}}[\mathbf{x} \mathbf{x}^\top]$$

be the data covariance matrix. Let $\mathbf{H} u_j = \lambda_j u_j$ with $\|u_j\|_2 = 1$ and $\lambda_1 \geq \cdots \geq \lambda_d > 0$.

**Lemma C.5** (Spectral identification for the linear kernel). *Define the integral operator $T : L^2(\rho_x) \to L^2(\rho_x)$ by*

$$(Tg)(\mathbf{x}) := \int_{\mathbb{R}^d} \langle \mathbf{x}, \mathbf{x}' \rangle \, g(\mathbf{x}') \, d\rho_x(\mathbf{x}').$$

*Then $\mathrm{Ran}(T)$ is contained in the $d$-dimensional linear function class $\{\mathbf{x} \mapsto \langle v, \mathbf{x} \rangle : v \in \mathbb{R}^d\}$ and hence $\mathrm{rank}(T) \leq d$. Moreover, for each $j \in [d]$, the function*

$$\phi_j(\mathbf{x}) := \frac{\langle u_j, \mathbf{x} \rangle}{\sqrt{\lambda_j}}$$

*belongs to $L^2(\rho_x)$, satisfies $\|\phi_j\|_{L^2(\rho_x)} = 1$, and is an eigenfunction of $T$ with eigenvalue $\lambda_j$:*

$$T\phi_j = \lambda_j \phi_j.$$

*Consequently, the nonzero eigenvalues of $T$ coincide with $\{\lambda_j\}_{j=1}^d$ (counting multiplicities), i.e., $\mu_j = \lambda_j$ for $j \in [d]$.*

*Proof.* Fix $g \in L^2(\rho_x)$ and define

$$m(g) := \mathbb{E}_{\mathbf{x}' \sim \rho_x}[\mathbf{x}' g(\mathbf{x}')] \in \mathbb{R}^d.$$

Then, for all $\mathbf{x} \in \mathbb{R}^d$,

$$(Tg)(\mathbf{x}) = \int \langle \mathbf{x}, \mathbf{x}' \rangle g(\mathbf{x}') \, d\rho_x(\mathbf{x}') = \Big\langle \mathbf{x}, \, \mathbb{E}[\mathbf{x}' g(\mathbf{x}')] \Big\rangle = \langle \mathbf{x}, m(g) \rangle,$$

so $\mathrm{Ran}(T) \subseteq \{\mathbf{x} \mapsto \langle v, \mathbf{x} \rangle : v \in \mathbb{R}^d\}$ and hence $\mathrm{rank}(T) \leq d$.

Next, fix $j \in [d]$ and let $\phi_j(\mathbf{x}) = \langle u_j, \mathbf{x} \rangle / \sqrt{\lambda_j}$. Its $L^2(\rho_x)$-norm satisfies

$$\|\phi_j\|^2_{L^2(\rho_x)} = \frac{1}{\lambda_j} \mathbb{E}[\langle u_j, \mathbf{x} \rangle^2] = \frac{1}{\lambda_j} u_j^\top \mathbb{E}[\mathbf{x}\mathbf{x}^\top] u_j = \frac{1}{\lambda_j} u_j^\top \mathbf{H} u_j = 1.$$

Similarly, for $i \neq j$,

$$\langle \phi_i, \phi_j \rangle_{L^2(\rho_x)} = \frac{1}{\sqrt{\lambda_i \lambda_j}} u_i^\top \mathbf{H} u_j = \frac{1}{\sqrt{\lambda_i \lambda_j}} u_i^\top (\lambda_j u_j) = 0,$$

so $\{\phi_j\}_{j=1}^d$ is orthonormal.

Moreover,

$$m(\phi_j) = \mathbb{E}\left[\mathbf{x}' \frac{\langle u_j, \mathbf{x}' \rangle}{\sqrt{\lambda_j}}\right] = \frac{1}{\sqrt{\lambda_j}} \mathbb{E}[\mathbf{x}'\mathbf{x}'^\top] u_j = \frac{1}{\sqrt{\lambda_j}} \mathbf{H} u_j = \sqrt{\lambda_j}\, u_j,$$

and therefore, for all $\mathbf{x}$,

$$(T\phi_j)(\mathbf{x}) = \langle \mathbf{x}, m(\phi_j) \rangle = \langle \mathbf{x}, \sqrt{\lambda_j} u_j \rangle = \lambda_j \frac{\langle u_j, \mathbf{x} \rangle}{\sqrt{\lambda_j}} = \lambda_j \phi_j(\mathbf{x}).$$

Thus $(\lambda_j, \phi_j)$ is an eigenpair of $T$. Since $T$ has rank at most $d$, these $d$ eigenpairs account for all nonzero eigenvalues (counting multiplicities), and hence $\mu_j = \lambda_j$ for $j \in [d]$. $\qquad\square$

Consider the linear predictor class $f_{\boldsymbol{\theta}}(\mathbf{x}) := \langle \boldsymbol{\theta}, \mathbf{x} \rangle$. Writing $\theta_j := \langle \boldsymbol{\theta}, u_j \rangle$ and $\theta_j^* := \langle \boldsymbol{\theta}^*, u_j \rangle$, Lemma C.5 yields the eigen-expansion

$$f_{\boldsymbol{\theta}}(\mathbf{x}) = \sum_{j=1}^d \sqrt{\lambda_j}\, \theta_j\, \phi_j(\mathbf{x}), \qquad f_{\boldsymbol{\theta}^*}(\mathbf{x}) = \sum_{j=1}^d \sqrt{\lambda_j}\, \theta_j^*\, \phi_j(\mathbf{x}),$$

and hence

$$\|f_{\boldsymbol{\theta}} - f_{\boldsymbol{\theta}^*}\|^2_{L^2(\rho_x)} = \sum_{j=1}^d \lambda_j (\theta_j - \theta_j^*)^2.$$

**Parameter correspondence under power laws.** Under Lemma C.5, we have $\mu_j = \lambda_j$ for $j \in [d]$. Assume further that the Hessian eigenvalues satisfy $\lambda_j \asymp j^{-\beta}$ with $\beta > 1$ (Assumption on power-decay Hessian eigenvalues). Then

$$\mathrm{Tr}(T^\gamma) = \sum_{j=1}^d \mu_j^\gamma = \sum_{j=1}^d \lambda_j^\gamma \asymp \sum_{j=1}^d j^{-\beta\gamma}.$$

In the large-$d$ regime, the threshold for boundedness occurs at $\beta\gamma = 1$. Accordingly, the *critical* trace exponent equals $\gamma_\star = 1/\beta$, yielding the identification

$$\gamma_\star = \frac{1}{\beta}. \tag{7}$$

If Assumption C.4 holds for some $\gamma$ uniformly for the arbitrary $d$ under the power-law spectrum, then necessarily $\gamma > \gamma_\star$ under this power-law model.

Moreover, in the linear model $y = \langle \boldsymbol{\theta}^*, \mathbf{x} \rangle + \epsilon$ with $\mathbb{E}[\epsilon \mid \mathbf{x}] = 0$, the regression function is $f_\rho(\mathbf{x}) = \mathbb{E}[y \mid \mathbf{x}] = \langle \boldsymbol{\theta}^*, \mathbf{x} \rangle$. Its eigencoordinates satisfy

$$\langle f_\rho, \phi_j \rangle_{L^2(\rho_x)} = \mathbb{E}\left[\langle \boldsymbol{\theta}^*, \mathbf{x} \rangle \frac{\langle u_j, \mathbf{x} \rangle}{\sqrt{\lambda_j}}\right] = \frac{1}{\sqrt{\lambda_j}} \boldsymbol{\theta}^{*\top} \mathbb{E}[\mathbf{x}\mathbf{x}^\top] u_j = \frac{1}{\sqrt{\lambda_j}} \boldsymbol{\theta}^{*\top} \mathbf{H} u_j = \sqrt{\lambda_j}\, \theta_j^*,$$

and hence

$$\langle f_\rho, \phi_j \rangle^2_{L^2(\rho_x)} = \lambda_j (\theta_j^*)^2.$$

Therefore, Assumption 2.1, namely $\lambda_j(\theta_j^*)^2 \asymp j^{-\alpha}$, is exactly a power-law decay assumption on the squared target coefficients in the eigenbasis of $T$.

Combining $\mu_j = \lambda_j \asymp j^{-\beta}$ and $\langle f_\rho, \phi_j \rangle^2 \asymp j^{-\alpha}$ with the source characterization yields

$$\sum_{j \geq 1} \frac{\langle f_\rho, \phi_j \rangle^2_{L^2(\rho_x)}}{\mu_j^{2r}} \asymp \sum_{j \geq 1} j^{-(\alpha - 2r\beta)}.$$

Consequently, $f_\rho \in \mathrm{Ran}(T^r)$ holds if and only if $\alpha - 2r\beta > 1$. Define the (critical) source exponent

$$r_\star := \frac{\alpha - 1}{2\beta}. \tag{8}$$

Then $f_\rho \in \mathrm{Ran}(T^r)$ for all $r < r_\star$, and $r_\star$ is the maximal admissible exponent under the power-law model. Together, (7) and (8) establish an explicit correspondence between the parameters in feature-space linear regression and those in power-law kernel regression, thereby clarifying the relationship between the two settings.

*Remark* C.6 (Parameter mapping). The equalities $\gamma_\star = 1/\beta$ and $r_\star = (\alpha - 1)/(2\beta)$ are exact under the two-sided power-law assumption $\mu_j \asymp j^{-\beta}$ and $\langle f_\rho, \phi_j \rangle^2 \asymp j^{-\alpha}$. If only one-sided polynomial bounds hold (or the decay is not exactly polynomial), the same reasoning yields corresponding bounds on $\gamma_\star$ and $r_\star$ rather than equalities.

## C.2   NOISE STRUCTURE

In this section, we investigate the structure of the gradient noise in the feature-space linear regression model. In particular, we show that the covariance of the gradient noise is aligned with the Hessian matrix $H$, thereby validating the Hessian-aligned noise assumption.

Let $\mathcal{B}_k = \{(\mathbf{x}_{k,i}, y_{k,i})\}_{i=1}^B$ denote the mini-batch sampled at iteration $k$. The corresponding mini-batch empirical loss is

$$\ell_{\mathcal{B}_k}(\boldsymbol{\theta}) = \frac{1}{2B} \sum_{i=1}^B \left( \langle \boldsymbol{\phi}(\mathbf{x}_{k,i}), \boldsymbol{\theta} \rangle - y_{k,i} \right)^2.$$

Accordingly, the SignSGD update takes the form

$$\boldsymbol{\theta}_{k+1} = \boldsymbol{\theta}_k - \eta \, \mathrm{sgn}\!\left( \nabla \ell_{\mathcal{B}_k}(\boldsymbol{\theta}_k) \right).$$

At iteration $k$, the mini-batch samples satisfy $\mathbf{x}_{k,i} \sim \mathcal{D}$ and $y_{k,i} = \langle \boldsymbol{\phi}(\mathbf{x}_{k,i}), \boldsymbol{\theta}^* \rangle + \epsilon_{k,i}$, where $\epsilon_{k,i} \overset{\text{i.i.d.}}{\sim} \mathcal{N}(0, \sigma^2)$. Consequently, the mini-batch gradient admits the form of

$$\nabla \ell_{\mathcal{B}_k}(\boldsymbol{\theta}) = \frac{1}{B} \sum_{i=1}^B \left( \langle \boldsymbol{\phi}(\mathbf{x}_{k,i}), \boldsymbol{\theta} \rangle - y_{k,i} \right) \boldsymbol{\phi}(\mathbf{x}_{k,i})$$

$$= \frac{1}{B} \sum_{i=1}^B \left( \langle \boldsymbol{\phi}(\mathbf{x}_{k,i}), \boldsymbol{\theta} - \boldsymbol{\theta}^* \rangle - \epsilon_{k,i} \right) \boldsymbol{\phi}(\mathbf{x}_{k,i})$$

$$= \frac{1}{B} \sum_{i=1}^B \left( \boldsymbol{\phi}(\mathbf{x}_{k,i}) \boldsymbol{\phi}(\mathbf{x}_{k,i})^\top (\boldsymbol{\theta} - \boldsymbol{\theta}^*) - \epsilon_{k,i} \, \boldsymbol{\phi}(\mathbf{x}_{k,i}) \right)$$

On the other hand, the population risk is $\mathcal{R}(\boldsymbol{\theta}) = \frac{1}{2} \mathbb{E}_{\mathbf{x},y} \left[ (\langle \boldsymbol{\phi}(\mathbf{x}), \boldsymbol{\theta} \rangle - y)^2 \right]$. Its gradient is given by

$$\nabla \mathcal{R}(\boldsymbol{\theta}) = \mathbb{E}_{\mathbf{x},y}[(\langle \boldsymbol{\phi}(\mathbf{x}), \boldsymbol{\theta} \rangle - y) \, \boldsymbol{\phi}(\mathbf{x})]$$

$$= \mathbb{E}_{\mathbf{x}} \left[ \boldsymbol{\phi}(\mathbf{x}) \boldsymbol{\phi}(\mathbf{x})^\top \right] (\boldsymbol{\theta} - \boldsymbol{\theta}^*),$$

Therefore, the gradient noise

$$\boldsymbol{\xi}_k = \nabla \ell_{\mathcal{B}_k}(\boldsymbol{\theta}_k) - \nabla \mathcal{R}(\boldsymbol{\theta}_k) = \left( \frac{1}{B} \sum_{i=1}^B \boldsymbol{\phi}(\mathbf{x}_{k,i}) \boldsymbol{\phi}(\mathbf{x}_{k,i})^\top - \mathbb{E}_{\mathbf{x}} \left[ \boldsymbol{\phi}(\mathbf{x}) \boldsymbol{\phi}(\mathbf{x})^\top \right] \right) (\boldsymbol{\theta}_k - \boldsymbol{\theta}^*) - \frac{1}{B} \sum_{i=1}^B \epsilon_{k,i} \, \boldsymbol{\phi}(\mathbf{x}_{k,i}).$$

Since $\phi(\mathbf{x}) \sim \mathcal{N}(\mathbf{0}, \mathbf{H})$, we can compute the conditional covariance of $\boldsymbol{\xi}_k$ given $\boldsymbol{\theta}_k$ explicitly. Let $\boldsymbol{\delta}_k := \boldsymbol{\theta}_k - \boldsymbol{\theta}^*$, $\boldsymbol{\phi}_{k,i} := \phi(\mathbf{x}_{k,i})$. Then $\mathbb{E}[\boldsymbol{\xi}_k \mid \boldsymbol{\theta}_k] = \mathbf{0}$ and

$$
\mathbb{E}\big[\boldsymbol{\xi}_k \boldsymbol{\xi}_k^\top \mid \boldsymbol{\theta}_k\big] = \mathbb{E}\left[\left(\frac{1}{B}\sum_{i=1}^{B}\big((\boldsymbol{\phi}_{k,i}\boldsymbol{\phi}_{k,i}^\top - \mathbf{H})\boldsymbol{\delta}_k - \epsilon_{k,i}\boldsymbol{\phi}_{k,i}\big)\right)\left(\frac{1}{B}\sum_{j=1}^{B}\big((\boldsymbol{\phi}_{k,j}\boldsymbol{\phi}_{k,j}^\top - \mathbf{H})\boldsymbol{\delta}_k - \epsilon_{k,j}\boldsymbol{\phi}_{k,j}\big)\right)^\top \,\Bigg|\, \boldsymbol{\theta}_k\right]
$$

$$
= \frac{1}{B}\,\mathbb{E}\big[\big((\boldsymbol{\phi}\boldsymbol{\phi}^\top - \mathbf{H})\boldsymbol{\delta}_k - \epsilon\boldsymbol{\phi}\big)\big((\boldsymbol{\phi}\boldsymbol{\phi}^\top - \mathbf{H})\boldsymbol{\delta}_k - \epsilon\boldsymbol{\phi}\big)^\top\big]
$$

$$
= \frac{1}{B}\,\mathbb{E}\big[(\boldsymbol{\phi}\boldsymbol{\phi}^\top - \mathbf{H})\boldsymbol{\delta}_k\boldsymbol{\delta}_k^\top(\boldsymbol{\phi}\boldsymbol{\phi}^\top - \mathbf{H})\big] + \frac{1}{B}\,\mathbb{E}\big[\epsilon^2\,\boldsymbol{\phi}\boldsymbol{\phi}^\top\big],
$$

Since $\mathbb{E}[\epsilon^2] = \sigma^2$, we have $\mathbb{E}[\epsilon^2\,\boldsymbol{\phi}\boldsymbol{\phi}^\top] = \sigma^2\,\mathbb{E}[\boldsymbol{\phi}\boldsymbol{\phi}^\top] = \sigma^2\mathbf{H}$. Moreover, for $\boldsymbol{\phi} \sim \mathcal{N}(\mathbf{0}, \mathbf{H})$,

$$
\mathbb{E}\big[(\boldsymbol{\phi}\boldsymbol{\phi}^\top - \mathbf{H})\boldsymbol{\delta}_k\boldsymbol{\delta}_k^\top(\boldsymbol{\phi}\boldsymbol{\phi}^\top - \mathbf{H})\big] = (\boldsymbol{\delta}_k^\top \mathbf{H}\boldsymbol{\delta}_k)\,\mathbf{H} + \mathbf{H}\boldsymbol{\delta}_k\boldsymbol{\delta}_k^\top\mathbf{H},
$$

and hence

$$
\mathbb{E}\big[\boldsymbol{\xi}_k \boldsymbol{\xi}_k^\top \mid \boldsymbol{\theta}_k\big] = \frac{1}{B}\Big((\sigma^2 + \boldsymbol{\delta}_k^\top\mathbf{H}\boldsymbol{\delta}_k)\,\mathbf{H} + \mathbf{H}\boldsymbol{\delta}_k\boldsymbol{\delta}_k^\top\mathbf{H}\Big).
$$

In particular, using $\mathbf{H}\boldsymbol{\delta}_k\boldsymbol{\delta}_k^\top\mathbf{H} \preceq (\boldsymbol{\delta}_k^\top\mathbf{H}\boldsymbol{\delta}_k)\,\mathbf{H}$, we obtain the PSD bounds

$$
\frac{\sigma^2 + \boldsymbol{\delta}_k^\top\mathbf{H}\boldsymbol{\delta}_k}{B}\,\mathbf{H} \preceq \mathbb{E}\big[\boldsymbol{\xi}_k\boldsymbol{\xi}_k^\top \mid \boldsymbol{\theta}_k\big] \preceq \frac{\sigma^2 + 2\boldsymbol{\delta}_k^\top\mathbf{H}\boldsymbol{\delta}_k}{B}\,\mathbf{H}.
$$

Consequently, if the iterates stay in a regime where the population loss is bounded, i.e., $\mathcal{R}(\boldsymbol{\theta}_k) \leq C$, then $\|\boldsymbol{\delta}_k\|_{\mathbf{H}}^2 = \boldsymbol{\delta}_k^\top\mathbf{H}\boldsymbol{\delta}_k \leq 2C$. If we further assume $\sigma^2 \gtrsim 1$, the above PSD bounds imply

$$
\mathbb{E}\big[\boldsymbol{\xi}_k\boldsymbol{\xi}_k^\top \mid \boldsymbol{\theta}_k\big] \approx \frac{\sigma^2}{B}\,\mathbf{H}.
$$

We note that the noise structure in Section 2 posits the stronger Gaussian surrogate $\boldsymbol{\xi}_k \sim \mathcal{N}(\mathbf{0}, \mathbf{H}/B)$. Our derivation does not establish exact Gaussianity of $\boldsymbol{\xi}_k$; rather, it shows that the covariance of the gradient noise is spectrally aligned with $\mathbf{H}$ and scales as $1/B$. This provides theoretical support for Assumption B.1, while the Gaussian form can be viewed as an approximation. Moreover, since $\boldsymbol{\xi}_k$ is an average of i.i.d. terms, the central limit theorem suggests that its distribution may be well-approximated by a Gaussian for sufficiently large $B$.

## D  Additional experimental details

### D.1  Experimental setup.

For language model experiments, we perform extensive pre-training runs to evaluate the proposed scaling law, using a LLaMA-style model (Touvron et al., 2023) (approximately 100M parameters) up to 10B tokens.

### D.2  Details of Figure 2

Figure 4 provides the fitting results of all batch sizes in the experiment of Figure 2.

For the fitting details in Figure 2c, we fit the parameters in the function $L_B = L_0 + \frac{C}{\sqrt{B}}$ using a least-squares fit. Figure 5 shows an alternative visualization of Figure 2c.

### D.3  Details of Figure 6a

The two fitted lines in Figure 6a are obtained by jointly fitting the model $L(D) = L_0 + C_1 D^{-\alpha}$ to both datasets. We treat $L_0$ as a shared parameter, while each curve has its own $C_1$ and $\alpha$. The fitting proceeds by performing a grid search over candidate exponents $(\alpha_1, \alpha_2)$, and for each pair, we use least-squares to estimate $L_0$, $C_1^{(1)}$, and $C_1^{(2)}$, computing the mean squared error over both curves. The optimal exponents are selected as those minimizing the combined MSE, and the corresponding $C_1$ values define the two fitted lines.

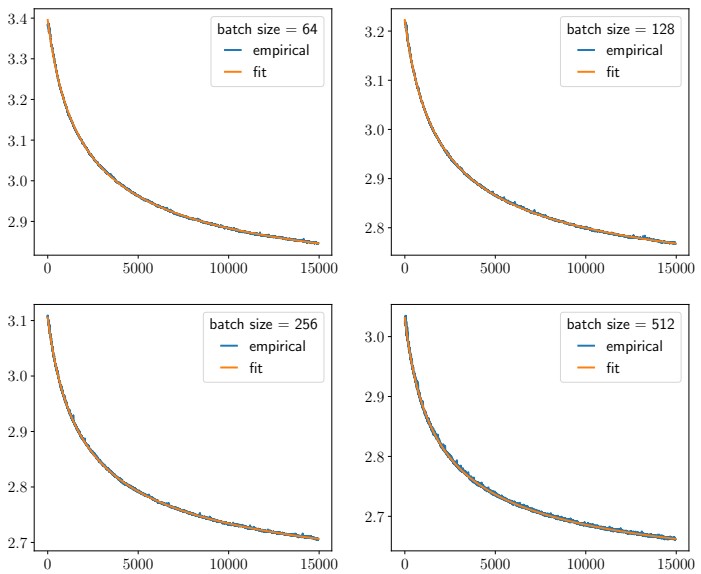

Figure 4: Curve fitting results of Figure 2

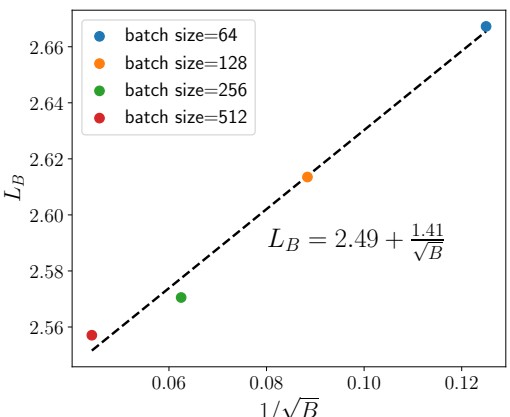

Figure 5: **Relationship between $L_B$ and batch size**. $L_B$ vs. $\frac{1}{\sqrt{B}}$.

## D.4 SIGNSGD OUTPERFORMS SGD IN LANGUAGE MODEL PRE-TRAINING AS PREDICTED BY THEORY

We now examine whether the performance advantage of SignSGD over SGD in LLM pre-training manifests in the manner predicted by our theory.

- **Data efficiency.** We train models using SignSGD and SGD under varying data budgets. For each data size, we tune the learning rate and batch size, and fit the resulting optimal loss $L$ as a function of the data size $D$ using the power law $L = L_0 + CD^{-\alpha}$. The irreducible loss $L_0$ is shared between SignSGD and SGD in the fitting procedure. Figure 6a reports the fitted data-scaling curves. We observe that SignSGD indeed achieves a substantially larger data-scaling exponent ($\alpha = 1.04$) than SGD ($\alpha = 0.67$), indicating superior data efficiency. In addition, SignSGD exhibits a significantly smaller multiplicative constant $C$ than SGD—an empirical advantage that goes beyond the predictions of our theory.

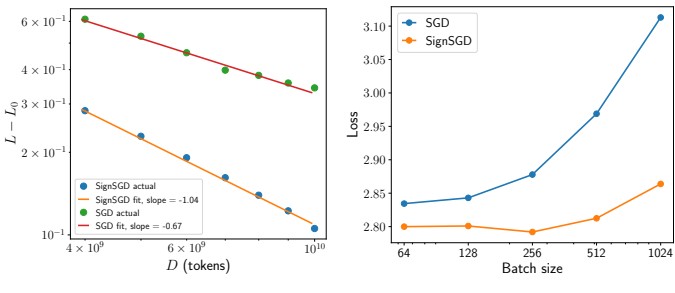

(a) Data scaling law for LLM     (b) Large batch training for LLM

Figure 6: **(a)** Data scaling law for LLM pre-training under optimally tuned hyperparameters; fitted slopes are $-1.04$ and $-0.67$ respectively. **(b)** The effect of batch size for final performance under a fixed data budget between SignSGD and SGD.

- **Large-batch training.** Figure 6b compares the final-step loss of SignSGD and SGD under a fixed data budget across different batch sizes. The learning rates are optimally tuned for each batch size. As the batch size increases, the performance gap remains approximately constant in the small-batch regime and then widens dramatically in the large-batch regime. Similar behavior has also been reported in Marek et al. (2025). These numerical results are consistent with our theoretical predictions: SignSGD has a much larger critical batch size than SGD and consequently, SignSGD benefits more from large-batch training than SGD.

### D.5 PHASE DIAGRAM OF SIGNSGD IN LANGUAGE MODELS

Figure 7 shows the results of SignSGD in language models.

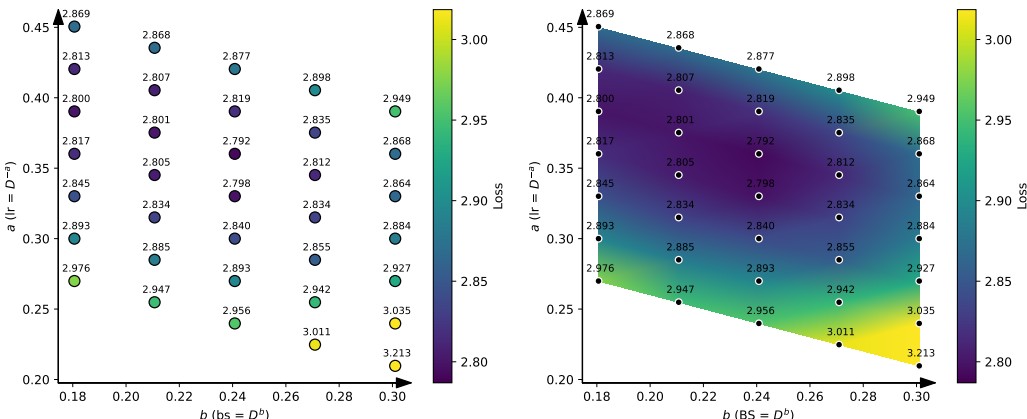

Figure 7: **Phase diagram of SignSGD in language models**. Real data points **(left)** and linear interpolation of these points **(right)**. In scaling regime, loss contours align with slope $-1/2$, corresponding to constant $\eta/\sqrt{B}$. The scaling boundary follows a slope of $1/2$, consistent with the condition $\eta\sqrt{B} \approx 1$.

Following the same procedure as in Figure 1a, we construct the phase diagram of SignSGD for language models with $D = 10B$. The result is shown in Figure 7. From the figure, we observe the following results.

First, the phase diagram has a scaling regime, and within the scaling regime, the loss remains approximately constant along curves satisfying $\eta/\sqrt{B} = \text{const}$. This confirms that, even in large-scale LLM training, the dependence of the loss on the learning rate and batch size satisfies the square root rule.

In practice, this means that, to achieve optimal loss, increasing the batch size requires a proportional increase of the learning rate. Tuning $\eta$ independently of $B$ would fail to track the optimal trajectory.

Second, the boundary separating the scaling regime from the unstable regime follows an approximately linear trend with slope close to $1/2$ in the phase diagram. This behavior corresponds to the stability condition $\eta\sqrt{B} \approx 1$ derived before. As a consequence, the batch size required to remain in the scaling regime depends explicitly on the learning rate. Larger learning rates necessitate smaller batch sizes to maintain stability.

### D.6 PHASE DIAGRAM IN FEATURE-SPACE LINEAR REGRESSION

**Phase diagram of SGD.** Figure 8 shows the phase diagram of SGD, from which we can observe two results. First, the contour lines exhibit a slope of $-1$, in contrast to the slope $-1/2$ observed for signSGD. This difference indicates that the effective learning rate of SGD scales as $\eta/B$ ($\eta/\sqrt{B}$ for SignSGD). Second, when $\eta \gtrsim 1$, SGD diverges.

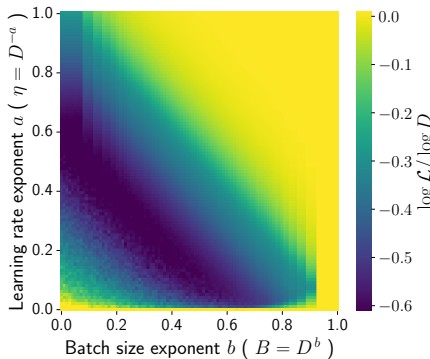

Figure 8: SGD phase diagram in feature-space linear regression

**Phase diagram of other optimizers.** Figure 9 shows the phase diagrams of other optimizers, including SGD with momentum, RMSProp, Signum, and Adam. Compared with SGD and SignSGD, these methods introduce additional state variables, which lead to more complicated phase-diagram structures. In particular, SignSGD in Figure 1 (left) and Adam in Figure 10 (right) exhibit similar phase diagram behaviors. However, Adam displays a noticeably larger scaling regime than SignSGD, indicating improved stability over a wider range of learning rates and batch sizes.

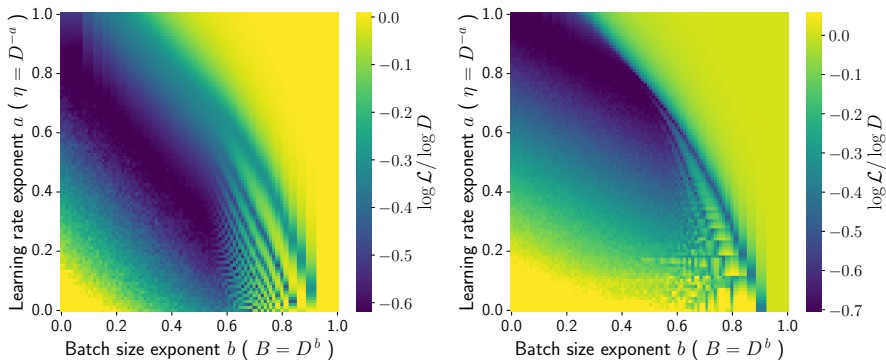

Figure 9: Phase diagram of other optimizers in feature-space linear regression. **SGD with momentum (left) and Signum (right)**.

## E DERIVATION OF SGD SCALING LAW

In this section, we derive the scaling law for SGD. We first state a lemma that will be useful in the analysis. As a warm-up, we also derive the scaling law for gradient flow (GF), which follows as a simpler special case.

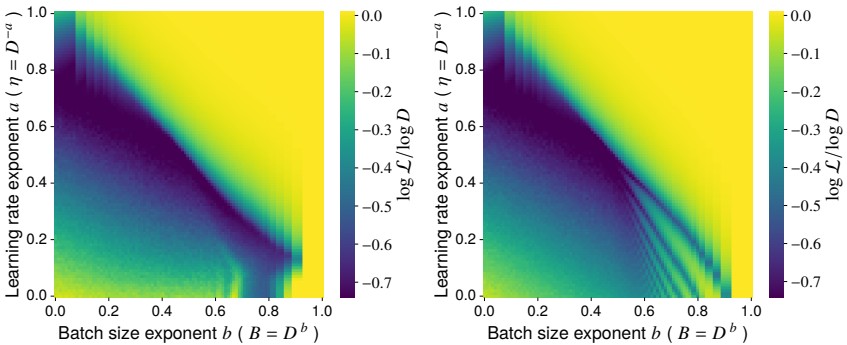

Figure 10: Phase diagram of other optimizers in feature-space linear regression. **RMSProp (left) and Adam (right)**.

**Lemma E.1.** *Let $d \in \mathbb{N}$ and let $f : [1, d] \to [0, \infty)$ be integrable. Assume $f$ is* unimodal *on $[1, d]$: there exists $x_0 \in [1, d]$ such that $f$ is non-decreasing on $[1, x_0]$ and non-increasing on $[x_0, d]$. Let*

$$M := \sup_{x \in [1,d]} f(x).$$

*Then the following symmetric estimate holds:*

$$\left| \sum_{j=1}^{d} f(j) - \int_{1}^{d} f(x)\, dx \right| \leq M.$$

*Proof.* If $x_0 = 1$ or $x_0 = d$, then $f$ is monotone on $[1, d]$.

*(1) $x_0 = 1$, $f$ is non-decreasing.* For each integer $j = 1, \ldots, d-1$ and all $x \in [j, j+1]$, we have

$$f(j) \leq f(x) \leq f(j+1),$$

so integrating gives

$$f(j) \leq \int_{j}^{j+1} f(x)\, dx \leq f(j+1).$$

Summing over $j = 1, \ldots, d-1$ gives

$$\sum_{j=1}^{d-1} f(j) \leq \int_{1}^{d} f(x)\, dx \leq \sum_{j=2}^{d} f(j),$$

and adding $f(d)$ to the left inequality and $f(1)$ to the right inequality gives

$$\sum_{j=1}^{d} f(j) \leq \int_{1}^{d} f(x)\, dx + f(d), \qquad \sum_{j=1}^{d} f(j) \geq \int_{1}^{d} f(x)\, dx + f(1).$$

Since $f(d), f(1) \leq M$, we obtain

$$\left| \sum_{j=1}^{d} f(j) - \int_{1}^{d} f(x)\, dx \right| \leq M.$$

*(2) $x_0 = d$, $f$ is non-increasing.* Similarly, for each $j = 1, \ldots, d-1$ and $x \in [j, j+1]$,

$$f(j+1) \leq f(x) \leq f(j),$$

which gives

$$\sum_{j=2}^{d} f(j) \leq \int_{1}^{d} f(x)\, dx \leq \sum_{j=1}^{d-1} f(j),$$

hence

$$\int_1^d f(x)\,dx + f(d) \le \sum_{j=1}^d f(j) \le \int_1^d f(x)\,dx + f(1),$$

and again $\left|\sum f(j) - \int f\right| \le M$.

(3) $1 < x_0 < d$. Set $k := \lfloor x_0 \rfloor \in \{1, 2, \ldots, d-1\}$, so that $x_0 \in [k, k+1]$.

*(3a) Proof of* $\sum f(j) \le \int f + M$. For $j = 1, \ldots, k-1$, the interval $[j, j+1] \subset [1, x_0]$ and $f$ is non-decreasing, so

$$f(j) \le \int_j^{j+1} f(x)\,dx.$$

For $j = k+2, \ldots, d$, the interval $[j-1, j] \subset [x_0, d]$ and $f$ is non-increasing, so

$$f(j) \le \int_{j-1}^j f(x)\,dx.$$

Summing the above gives

$$\sum_{\substack{1 \le j \le d \\ j \ne k, k+1}} f(j) \le \int_1^d f(x)\,dx - \int_k^{k+1} f(x)\,dx.$$

Hence

$$\sum_{j=1}^d f(j) \le \int_1^d f(x)\,dx + \left(f(k) + f(k+1) - \int_k^{k+1} f(x)\,dx\right).$$

Since $f$ is non-decreasing on $[k, x_0]$ and non-increasing on $[x_0, k+1]$, we have $f(x) \ge \min\{f(k), f(k+1)\}$ for all $x \in [k, k+1]$, so

$$f(k) + f(k+1) - \int_k^{k+1} f(x)\,dx \le \max\{f(k), f(k+1)\} \le M.$$

Thus $\sum_{j=1}^d f(j) \le \int_1^d f(x)\,dx + M$.

*(3b) Proof of* $\int f \le \sum f(j) + M$. Decompose the integral:

$$\int_1^d f(x)\,dx = \sum_{j=1}^{d-1} \int_j^{j+1} f(x)\,dx.$$

For $j = 1, \ldots, k-1$, $[j, j+1] \subset [1, x_0]$ and $f$ is non-decreasing, so $\int_j^{j+1} f \le f(j+1)$. For $j = k+1, \ldots, d-1$, $[j, j+1] \subset [x_0, d]$ and $f$ is non-increasing, so $\int_j^{j+1} f \le f(j)$. Thus

$$\int_1^d f(x)\,dx - \int_k^{k+1} f(x)\,dx \le \sum_{j=2}^{d-1} f(j) \le \sum_{j=1}^d f(j),$$

and since $\int_k^{k+1} f(x)\,dx \le M$, we obtain $\int_1^d f(x)\,dx \le \sum_{j=1}^d f(j) + M$.

Combining all cases gives the desired estimate. □

### E.1 SCALING LAW OF GRADIENT FLOW

We consider the gradient flow (GF) method to solve the feature-space linear regression problem. Let $\boldsymbol{\theta}(t)$ denote the parameter at the continuous-time $t$, which satisfies the following ODE:

$$\dot{\boldsymbol{\theta}}(t) = -\nabla\mathcal{R}(\boldsymbol{\theta}(t)), \tag{9}$$

We have the following theorem.

**Theorem E.2** (Scaling law for GF). *When $1 \lesssim t \lesssim d^\beta$, we have*

$$\mathcal{L}(\boldsymbol{\theta}(t)) \approx t^{-s},$$

*where $s := \frac{\alpha-1}{\beta}$ denotes the relative task difficulty.*

The following lemma will be useful.

**Lemma E.3.** *Let $\beta > 1$ and $\alpha > 1$. Consider the integral*

$$I(t,d) := \int_1^d j^{-\alpha} \exp(-tj^{-\beta}) \, \mathrm{d}j.$$

*Then the following asymptotic behaviors hold:*

$$I(t,d) \approx \begin{cases} t^{-s}, & 1 \lesssim t \lesssim d^\beta, \\ t^{-1} d^{-\beta(s-1)} \exp(-td^{-\beta}), & t \gtrsim d^\beta, \end{cases}$$

*where $s := (\alpha - 1)/\beta$.*

*Proof.* Perform the change of variables

$$z = tj^{-\beta}, \qquad j = \left(\frac{z}{t}\right)^{-1/\beta}, \qquad \mathrm{d}j = \frac{1}{\beta} t^{1/\beta} z^{-1/\beta - 1} \, \mathrm{d}z.$$

Then the integral becomes

$$I(t,d) = \int_1^d j^{-\alpha} e^{-tj^{-\beta}} \, \mathrm{d}j$$

$$= \frac{1}{\beta} t^{-s} \int_{td^{-\beta}}^t z^{s-1} e^{-z} \, \mathrm{d}z.$$

where $s := (\alpha - 1)/\beta$.

*Case 1: $1 \lesssim t \lesssim d^\beta$.*

In this regime, the lower limit satisfies $td^{-\beta} \lesssim 1$. Since $z^{s-1}e^{-z}$ is integrable on $(0,\infty)$ for $s > 0$, and the interval $\left[td^{-\beta}, t\right]$ captures a constant fraction of the mass of this integrable function when $1 \lesssim t \lesssim d^\beta$, we have

$$\int_{td^{-\beta}}^t z^{s-1} e^{-z} \, \mathrm{d}z \approx \int_0^\infty z^{s-1} e^{-z} \, \mathrm{d}z = \Gamma(s).$$

As a result,

$$I(t,d) \approx t^{-s}.$$

*Case 2: $t \gtrsim d^\beta$.*

In this regime, the lower limit satisfies $td^{-\beta} \gtrsim 1$. For sufficiently large $z$, the function $z^{s-1}e^{-z}$ is positive, unimodal, and monotonically decreasing. Moreover, by a standard tail asymptotic of the incomplete Gamma function, as a result, we have

$$\int_x^\infty z^{s-1} e^{-z} \, \mathrm{d}z \approx x^{s-1} e^{-x}, \qquad x \to \infty.$$

Applying this result with $x = td^{-\beta}$, and noting that truncating the upper limit at $t$ only removes an exponentially smaller tail, we obtain

$$\int_{td^{-\beta}}^t z^{s-1} e^{-z} \, \mathrm{d}z \approx (td^{-\beta})^{s-1} e^{-td^{-\beta}}.$$

Multiplying by the prefactor $t^{-s}$ yields

$$I(t,d) \approx t^{-1} d^{-\beta(s-1)} e^{-td^{-\beta}}.$$

This completes the proof.

$\square$

With Lemma E.1 and E.3, we can now begin the proof of Theorem E.2.

*Proof of Theorem E.2.* Due to the separable quadratic structure of the loss function, the gradient flow dynamics (9) decouples across coordinates. In particular, for each coordinate $j \in [d]$, the parameter $\theta_j(t)$ evolves according to the linear ODE

$$\dot{\theta}_j(t) = -\lambda_j(\theta_j(t) - \theta_j^*), \qquad \theta_j(0) = 0.$$

Solving the above equation gives

$$\theta_j(t) = \theta_j^*(1 - e^{-\lambda_j t}).$$

Consequently, the loss associated with the $j$-th sub-task at time $t$ is given by

$$\mathcal{L}_j(t) = \tfrac{1}{2}\lambda_j(\theta_j(t) - \theta_j^*)^2 = \tfrac{1}{2}\lambda_j(\theta_j^*)^2 e^{-2\lambda_j t}.$$

The total loss is the sum of all sub-task losses:

$$\mathcal{L}(\boldsymbol{\theta}(t)) = \sum_{j=1}^{d} \tfrac{1}{2}\lambda_j(\theta_j^*)^2 e^{-2\lambda_j t}.$$

Under the assumed power-law scaling $\lambda_j \asymp j^{-\beta}$ and $\lambda_j(\theta_j^*)^2 \asymp j^{-\alpha}$, this sum can be written as

$$\mathcal{L}(\boldsymbol{\theta}(t)) \asymp \sum_{j=1}^{d} j^{-\alpha} \exp(-2tj^{-\beta}).$$

We now justify that the above sum is equivalent, up to a multiplicative constant, to its continuous integral approximation. That is

$$\sum_{j=1}^{d} j^{-\alpha} \exp(-2tj^{-\beta}) \asymp \int_1^d x^{-\alpha} \exp(-2tx^{-\beta})\,\mathrm{d}x.$$

Define
$$f(x) := x^{-\alpha} \exp(-2tx^{-\beta}), \qquad x \in [1, d].$$

A direct computation shows that $f$ is unimodal on $(0, \infty)$. Indeed,

$$f'(x) = x^{-\alpha-1} e^{-2tx^{-\beta}}(-\alpha + 2t\beta x^{-\beta}),$$

so $f$ attains its unique maximum at

$$x_* = \left(\frac{2\beta}{\alpha}t\right)^{1/\beta}.$$

Evaluating $f$ at $x_*$ gives
$$f(x_*) = x_*^{-\alpha} \exp(-2tx_*^{-\beta}) \asymp t^{-\alpha/\beta},$$

where the exponential factor is a constant independent of $t$. Consequently,

$$\sup_{x \in [1,d]} f(x) \asymp t^{-\alpha/\beta}.$$

Since $f$ is unimodal on $[1, d]$, Lemma E.1 implies

$$\sum_{j=1}^{d} f(j) \asymp \int_1^d f(x)\,\mathrm{d}x + t^{-\alpha/\beta}.$$

Applying Lemma E.3, we obtain

$$\int_1^d f(x)\,\mathrm{d}x \asymp t^{-(\alpha-1)/\beta}, \qquad 1 \lesssim t \lesssim d^\beta.$$

Since $\alpha > 1$, the discretization error satisfies

$$t^{-\alpha/\beta} = o\left(t^{-(\alpha-1)/\beta}\right),$$

and is therefore negligible compared to the integral contribution. As a result,

$$\sum_{j=1}^{d} j^{-\alpha} \exp(-2tj^{-\beta}) \approx t^{-(\alpha-1)/\beta}, \qquad 1 \lesssim t \lesssim d^{\beta}.$$

Therefore, in this time regime, the total loss exhibits a power-law decay

$$\mathcal{L}(\boldsymbol{\theta}(t)) \approx t^{-s}, \qquad s = \frac{\alpha - 1}{\beta},$$

which completes the proof.

$\square$

## E.2  SCALING LAW OF SGD

In this section, we establish the full scaling law for SGD. In particular, we prove the following theorem, which is a more detailed version of (6).

**Theorem E.4.** *Suppose that $\eta \lesssim 1$, and let $\mathcal{L}(\boldsymbol{\theta}_k)$ denote the loss after $k$ steps of SGD update in (5). Then*

$$\mathbb{E}[\mathcal{L}(\boldsymbol{\theta}_k)] \approx \begin{cases} \dfrac{1}{(\eta k)^s} + \dfrac{\eta}{B}\left[1 - (\eta k)^{-(1-1/\beta)}\right], & 1 \lesssim \eta k \lesssim d^{\beta}, \\ (\eta k)^{-1} d^{-\beta(s-1)} \exp(-\Theta(\eta k d^{-\beta})) + \dfrac{\eta}{B}, & \eta k \gtrsim d^{\beta}. \end{cases} \tag{10}$$

We note that this theorem includes both the leading-order terms (e.g. $\frac{\eta}{B}$) and subdominant corrections (e.g. the $(\eta k)^{-(1-1/\beta)}$ decay). By ignoring the subdominant terms and considers the typical scaling regime $\eta k \lesssim d^{\beta}$, one immediately recovers the original scaling law for SGD of (6).

Before the proof of Theorem E.4, we will first prove the following lemma.

**Lemma E.5.** *Let $\beta > 1$. Consider the integral*

$$I(\eta, k, d) := \int_1^d j^{-\beta}\left(1 - \exp(-\eta k j^{-\beta})\right) \mathrm{d}j.$$

*Then the following asymptotic behaviors hold:*

$$I(\eta, k, d) \approx \begin{cases} 1 - (\eta k)^{\frac{1}{\beta}-1}, & 1 \lesssim \eta k \lesssim d^{\beta}, \\ 1, & \eta k \gtrsim d^{\beta}. \end{cases}$$

*Proof.* We perform the change of variables

$$z = \eta k j^{-\beta}, \qquad j = \left(\frac{z}{\eta k}\right)^{-\frac{1}{\beta}}, \qquad \mathrm{d}j = \frac{1}{\beta}(\eta k)^{\frac{1}{\beta}} z^{-\frac{1}{\beta}-1} \mathrm{d}z.$$

Then the integral becomes

$$I(\eta, k, d) = \frac{1}{\beta}(\eta k)^{\frac{1}{\beta}-1} \int_{\eta k d^{-\beta}}^{\eta k} z^{-\frac{1}{\beta}}\left(1 - e^{-z}\right) \mathrm{d}z. \tag{11}$$

We analyze (11) in different regimes.

*Case 1: $1 \lesssim \eta k \lesssim d^{\beta}$.* We split the integral in (11) into two regions.

If $z \in [\eta k d^{-\beta}, 1]$, since $z$ is small, we have $1 - e^{-z} \approx z$.

Thus, the integral over this region is

$$\int_{\eta k d^{-\beta}}^{1} z^{-1/\beta}(1-e^{-z})\,\mathrm{d}z \;\approxeq\; \int_{\eta k d^{-\beta}}^{1} z^{1-1/\beta}\,\mathrm{d}z = \frac{1-(\eta k d^{-\beta})^{2-1/\beta}}{2-1/\beta} \;\approxeq\; 1-(\eta k)^{-(2-1/\beta)}d^{\beta(2-1/\beta)}.$$

If $z \in [1, \eta k]$, we have $1 - e^{-z} \approxeq 1$, so

$$\int_{1}^{\eta k} z^{-1/\beta}(1-e^{-z})\,\mathrm{d}z \;\approxeq\; \int_{1}^{\eta k} z^{-1/\beta}\,\mathrm{d}z = \frac{(\eta k)^{1-1/\beta}-1}{1-1/\beta} \;\approxeq\; (\eta k)^{1-1/\beta}.$$

Adding the two contributions and multiplying by the prefactor $(\eta k)^{1/\beta-1}$ gives

$$I(\eta, k, d) \;\approxeq\; (\eta k)^{1/\beta-1}\Big[1 - (\eta k)^{-(2-1/\beta)}d^{\beta(2-1/\beta)} + (\eta k)^{1-1/\beta}\Big].$$

Observing that the small correction $(\eta k)^{-(2-1/\beta)}d^{\beta(2-1/\beta)}$ is subdominant, we arrive at

$$I(\eta, k, d) \;\approxeq\; 1 - (\eta k)^{-(1-1/\beta)},$$

which reproduces the expected leading-order behavior.

*Case 2:* $\eta k \gtrsim d^{\beta}$. Now $\eta k d^{-\beta} \gtrsim 1$, so $1 - e^{-z} \approxeq 1$ throughout the integration range. Thus,

$$\int_{\eta k d^{-\beta}}^{\eta k} z^{-1/\beta}(1-e^{-z})\,\mathrm{d}z \;\approxeq\; \int_{\eta k d^{-\beta}}^{\eta k} z^{-1/\beta}\,\mathrm{d}z = \frac{(\eta k)^{1-1/\beta}-(\eta k d^{-\beta})^{1-1/\beta}}{1-1/\beta} \;\approxeq\; (\eta k)^{1-1/\beta}.$$

Multiplying by $(\eta k)^{1/\beta-1}$ gives

$$I(\eta, k, d) \;\approxeq\; 1 - d^{-(1-1/\beta)} \;\approxeq\; 1.$$

$\square$

*Proof of Theorem E.4.* We now compute the SGD scaling law. Since the loss is quadratic and the gradient noise is Hessian-aligned (Assumption B.1), the SGD dynamics decouples across coordinates and we may analyze each coordinate independently.

Fix a coordinate $j$. The SGD update reads

$$\theta_{k+1}^{(j)} = \theta_k^{(j)} - \eta\big(\lambda_j(\theta_k^{(j)} - \theta_j^*) + \xi_k^{(j)}\big), \tag{12}$$

where $\xi_k^{(j)} \sim \mathcal{N}(0, \lambda_j/B)$. Define the estimation error $\delta_k^{(j)} := \theta_k^{(j)} - \theta_j^*$, so that $\delta_0^{(j)} = -\theta_j^*$. Subtracting $\theta_j^*$ from both sides of (12) gives

$$\delta_{k+1}^{(j)} = (1 - \eta\lambda_j)\delta_k^{(j)} - \eta\xi_k^{(j)}.$$

Squaring and taking expectation yields the recursion

$$\mathbb{E}[(\delta_{k+1}^{(j)})^2] = (1-\eta\lambda_j)^2\,\mathbb{E}[(\delta_k^{(j)})^2] + \eta^2\,\mathbb{E}[(\xi_k^{(j)})^2] = (1-\eta\lambda_j)^2\,\mathbb{E}[(\delta_k^{(j)})^2] + \frac{\eta^2\lambda_j}{B}. \tag{13}$$

Unrolling (13) gives the closed form

$$\begin{aligned}
\mathbb{E}[(\delta_k^{(j)})^2] &= (1-\eta\lambda_j)^{2k}(\delta_0^{(j)})^2 + \frac{\eta^2\lambda_j}{B}\sum_{r=0}^{k-1}(1-\eta\lambda_j)^{2r} \\
&= (1-\eta\lambda_j)^{2k}(\delta_0^{(j)})^2 + \frac{\eta^2\lambda_j}{B}\cdot\frac{1-(1-\eta\lambda_j)^{2k}}{1-(1-\eta\lambda_j)^2} \\
&= (1-\eta\lambda_j)^{2k}(\delta_0^{(j)})^2 + \frac{\eta}{B}\cdot\frac{1-(1-\eta\lambda_j)^{2k}}{2-\eta\lambda_j}.
\end{aligned} \tag{14}$$

We assume $\eta \leq 1/2$ and thus $\eta\lambda_j \leq 1/2$ for all $j$. Then $2 - \eta\lambda_j \in [3/2, 2]$, hence $\frac{1}{2-\eta\lambda_j} \approxeq 1$ as a multiplicative constant. To convert $(1-\eta\lambda_j)^{2k}$ into exponentials without changing the exponent by an uncontrolled constant, we use the two-sided bound

$$-2x \leq \log(1-x) \leq -x, \qquad \forall x \in (0, 1/2).$$

With $x = \eta\lambda_j$ and multiplying by $2k$, we obtain

$$-4\eta\lambda_j k \leq 2k\log(1 - \eta\lambda_j) \leq -2\eta\lambda_j k,$$

and therefore

$$\exp(-4\eta\lambda_j k) \leq (1 - \eta\lambda_j)^{2k} \leq \exp(-2\eta\lambda_j k). \tag{15}$$

Plugging (15) into (14) yields an upper bound of the form

$$\mathbb{E}[(\delta_k^{(j)})^2] \lesssim (\delta_0^{(j)})^2 \exp(-2\eta\lambda_j k) + \frac{\eta}{B}\Big(1 - \exp(-4\eta\lambda_j k)\Big), \tag{16}$$

and similarly a lower bound

$$\mathbb{E}[(\delta_k^{(j)})^2] \gtrsim (\delta_0^{(j)})^2 \exp(-4\eta\lambda_j k) + \frac{\eta}{B}\Big(1 - \exp(-2\eta\lambda_j k)\Big). \tag{17}$$

Since $\mathcal{L}(\boldsymbol{\theta}_k) = \frac{1}{2}\sum_{j=1}^d \lambda_j (\delta_k^{(j)})^2$, we have

$$\mathbb{E}[\mathcal{L}(\boldsymbol{\theta}_k)] = \frac{1}{2}\sum_{j=1}^d \lambda_j \,\mathbb{E}[(\delta_k^{(j)})^2].$$

Using (16) and the eigenvalue structure $\lambda_j \asymp j^{-\beta}$, together with the same sum-to-integral argument as in Theorem E.2, we obtain

$$\mathbb{E}[\mathcal{L}(\boldsymbol{\theta}_k)] \lesssim \frac{1}{2}\int_1^d j^{-\beta}(\delta_0^{(j)})^2 \exp(-2\eta k j^{-\beta})\,\mathrm{d}j$$

$$+ \frac{\eta}{2B}\int_1^d j^{-\beta}\Big(1 - \exp(-4\eta k j^{-\beta})\Big)\,\mathrm{d}j.$$

The first integral, which corresponds to signal learning, equals $\int_1^d j^{-\alpha}\exp(-2\eta k j^{-\beta})\,\mathrm{d}j$, hence by Lemma E.3,

$$\int_1^d j^{-\alpha}\exp(-2\eta k j^{-\beta})\,\mathrm{d}j \asymp \begin{cases} (\eta k)^{-s}, & 1 \lesssim \eta k \lesssim d^\beta, \\ (\eta k)^{-1}d^{-\beta(s-1)}\exp(-2\eta k d^{-\beta}), & \eta k \gtrsim d^\beta. \end{cases}$$

where $s = (\alpha - 1)/\beta$. The second integral represents the noise accumulation and matches the form in Lemma E.5 up to an absolute multiplicative constant $c = 4$, which we absorb into the $\asymp$ notation. Consequently,

$$\frac{\eta}{2B}\int_1^d j^{-\beta}\Big(1 - \exp(-4\eta k j^{-\beta})\Big)\,\mathrm{d}j \asymp \begin{cases} \dfrac{\eta}{B}[1 - (\eta k)^{-(1-1/\beta)}], & 1 \lesssim \eta k \lesssim d^\beta, \\ \dfrac{\eta}{B}, & \eta k \gtrsim d^\beta. \end{cases}$$

Combining the two gives the stated upper bound:

$$\mathbb{E}[\mathcal{L}(\boldsymbol{\theta}_k)] \lesssim \begin{cases} (\eta k)^{-s} + \dfrac{\eta}{B}[1 - (\eta k)^{-(1-1/\beta)}], & 1 \lesssim \eta k \lesssim d^\beta, \\ (\eta k)^{-1}d^{-\beta(s-1)}\exp(-2\eta k d^{-\beta}) + \dfrac{\eta}{B}, & \eta k \gtrsim d^\beta. \end{cases}$$

The lower bound follows analogously from (17), where the signal exponential uses constant $c = 4$ and the noise exponential uses $c = 2$. It follows that

$$\mathbb{E}[\mathcal{L}(\boldsymbol{\theta}_k)] \gtrsim \begin{cases} (\eta k)^{-s} + \dfrac{\eta}{B}[1 - (\eta k)^{-(1-1/\beta)}], & 1 \lesssim \eta k \lesssim d^\beta, \\ (\eta k)^{-1}d^{-\beta(s-1)}\exp(-4\eta k d^{-\beta}) + \dfrac{\eta}{B}, & \eta k \gtrsim d^\beta. \end{cases}$$

This completes the proof. $\qquad\square$

## F  DERIVATION OF SIGNSGD SCALING LAW

In this section we show how to derive the scaling law of SignSGD. This is based on the analysis of SignSGD training dynamics.

## F.1 ANALYSIS OF SIGNSGD TRAINING DYNAMICS

Under Assumptions B.1 and B.2, the SignSGD dynamics decouple across coordinates. Let $\boldsymbol{\theta}_k = (\theta_k^{(j)}, \theta_k^{(j)}, \ldots, \theta_k^{(j)})^\top$ and define the coordinate-wise error $\delta_k^{(j)} = \theta_k^{(j)} - \theta_j^*$ for each $j \in [d]$ at step $k$. Then, for each coordinate, SignSGD update is simplified as

$$\delta_{k+1}^{(j)} = \delta_k^{(j)} - \eta \, \mathrm{sgn}\big(\lambda_j \delta_k^{(j)} + \xi_k^{(j)}\big), \tag{18}$$

with $\delta_0^{(j)} = -\theta_j^*$ and $\xi_k^{(j)} \sim \mathcal{N}(0, \lambda_j/B)$.

The analysis naturally splits into two steps: we first study the one-dimensional dynamics for each coordinate, and then aggregate the results across coordinates.

Let $m_k^{(j)} = \mathbb{E}[\mathrm{sgn}(\lambda_j \delta_k^{(j)} + \xi_k^{(j)})|\delta_k^{(j)}]$ denote the drift term and $n_k^{(j)} = \mathrm{sgn}(\lambda_j \delta_k^{(j)} + \xi_k^{(j)}) - m_k^{(j)}$ the noise term. We have $\mathbb{E}[n_k^{(j)}|\delta_k^{(j)}] = 0$ and $\mathrm{Var}[n_k^{(j)}|\delta_k^{(j)}] = 1 - (m_k^{(j)})^2$. The SignSGD update can be written as

$$\delta_{k+1}^{(j)} = \delta_k^{(j)} - \eta(m_k^{(j)} + n_k^{(j)}).$$

Define a function $\phi_a(x) := \mathbb{E}_{\xi \sim \mathcal{N}(0,a^2)}[\mathrm{sgn}(x + \xi)]$ that can be used to describe the drift term $m_k$. We have $m_k^{(j)} = \mathbb{E}_{\xi_k^{(j)}}[\mathrm{sgn}(\lambda_j \delta_k^{(j)} + \xi_k^{(j)})|\delta_k^{(j)}] = \mathbb{E}[\mathrm{sgn}(\delta_k^{(j)} + \xi_k^{(j)}/\lambda_j)|\delta_k^{(j)}] = \phi_{1/\sqrt{\lambda_j B}}(\delta_k^{(j)})$. It can be shown that $\phi_a$ has a saturation region (far from origin) and a linear region (near origin):

$$\phi_a(x) \approx \begin{cases} \mathrm{sgn}(x), & |x| \gg a \\ \sqrt{\frac{2}{\pi}} \frac{x}{a}, & |x| \ll a \end{cases} \tag{19}$$

We call $\phi_a$ the smoothed sign function. This function shows us the dependence of $m_k^{(j)}$ on $\delta_k^{(j)}$. See Figure 11a for an illustration.

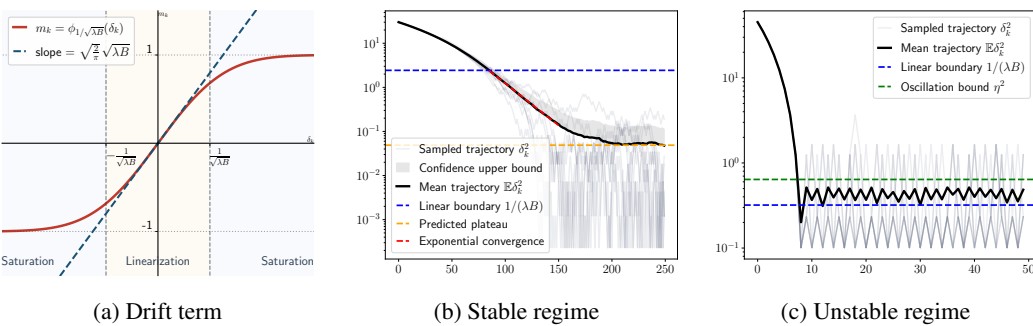

|  (a) Drift term | (b) Stable regime | (c) Unstable regime |

Figure 11: **An illustration of the 1D SignSGD dynamics. (a)** The drift term (expected sign update) $m(\delta_k) = \mathbb{E}[\mathrm{sgn}(\lambda\delta_k + \xi_k) \mid \delta_k]$ is approximately linear in $\delta_k$ near 0 and saturates to $\pm 1$ when $|\delta_k|$ is large, dividing the dynamics into a linear region and a saturation region. **(b)–(c)** Trajectories of the 1D dynamics. We plot $\delta_k^2$ (together with its mean) on a log scale. **(b)** $\eta\sqrt{\lambda B} \lesssim 1$. After a burn-in period, the expected squared residual exhibits exponential decay, as predicted by the linear recurrence (23). Moreover, the confidence upper bound (mean + standard deviation) stays below the linear boundary, suggesting that the trajectory is highly unlikely to escape the linear region. The dynamics eventually reach a plateau consistent with the theory, on the order of $\eta/\sqrt{\lambda B}$. **(c)** $\eta\sqrt{\lambda B} \gtrsim 1$. The dynamics cannot stably converge in the linear region: the trajectory repeatedly jumps in and out of the region, and the expected squared residual $\mathbb{E}[\delta_k^2]$ remains at the level of $\eta^2$.

**Saturation region** $(|\delta_k^{(j)}| \gg 1/\sqrt{\lambda_j B})$**.** In this region, we have $m_k^{(j)} \approx \mathrm{sgn}(\delta_k^{(j)})$ and $\mathrm{Var}[n_k^{(j)} \mid \delta_k^{(j)}] \approx 0$. Therefore, SignSGD is well approximated by the deterministic update

$$\delta_{k+1}^{(j)} = \delta_k^{(j)} - \eta \, \mathrm{sgn}(\delta_k^{(j)}),$$

which coincides with the update of SignGD.

**Linear region** ($|\delta_k^{(j)}| \ll 1/\sqrt{\lambda_j B}$). In this region, we have $m_k^{(j)} \approx \sqrt{\lambda_j B}\,\delta_k^{(j)}$, and hence the update can be approximated by

$$\delta_{k+1}^{(j)} = (1 - \eta\sqrt{\lambda_j B})\delta_k^{(j)} - \eta n_k^{(j)}. \tag{20}$$

Formally, we define the linear region for the stochastic process in (18) as $\mathcal{R}_j = [-\frac{1}{\sqrt{\lambda_j B}}, \frac{1}{\sqrt{\lambda_j B}}]$. Leveraging the properties of the smoothed sign function, it holds that $\mathbb{E}_{\xi_k^{(j)}}[\mathrm{sgn}(\lambda_j \delta_k^{(j)} + \xi_k^{(j)})] = \Theta(\sqrt{\lambda_j B}\,\delta_k^{(j)})$ for any $\delta_k^{(j)} \in \mathcal{R}_j$. To further study the stochastic dynamics in (18), we define the burn-in time

$$k_j^* = \inf\{k \in \mathbb{N} \mid \delta_k^{(j)} \in \mathcal{R}_j\}, \tag{21}$$

which represents the first time the process $\delta_k^{(j)}$ enters the linear region. Note that $\lambda(\delta_0^{(j)})^2 \approx j^{-\alpha}$. If $j \gtrsim B^{1/\alpha}$, the initial residual $\delta_0^{(j)} \in \mathcal{R}_j$ and thus $k_j^* = 0$. Conversely, in the regime where $1 \le j \lesssim B^{1/\alpha}$, burn-in time is described by the following lemma.

**Lemma F.1.** *For $1 \le j \lesssim B^{1/\alpha}$ and suppose that $\eta\sqrt{\lambda_j B} \lesssim 1$, with probability at least $1 - \exp(-\Omega(|\theta_j^*|/\eta))$, it holds that $k_j^* \approx |\theta_j^*|/\eta$.*

The proof can be found in Appendix H.1. Intuitively, so long as the residual $\delta_k^{(j)}$ remains far from the linear region $\mathcal{R}_j$, i.e., $|\delta_k^{(j)}| \gg \frac{1}{\sqrt{\lambda_j B}}$, the update (18) is effectively governed by the standard SignGD dynamics, i.e., the conditionally expected next-step residual $\mathbb{E}[\delta_{k+1}^{(j)} \mid \delta_k^{(j)}] = \delta_k^{(j)} - \eta\phi_{\sqrt{\lambda_j/B}}(\lambda_j \delta_k^{(j)}) \approx \delta_k^{(j)} - \eta\,\mathrm{sgn}(\lambda_j \delta_k^{(j)})$, thus necessitating $\Theta(|\theta_j^*|/\eta)$ iterations for the iterate to reach $\mathcal{R}_j$.

When $\eta\sqrt{\lambda_j B} \gtrsim 1$, it is not appropriate to define the burn-in time via (21), since the width of the linear region is of the same order or smaller than the step size, i.e., $2/\sqrt{\lambda_j B} \lesssim \eta$. In this case, the trajectory may not enter the linear region; instead, it can jump across it in a single update. Therefore, we define

$$k_j^{**} = \inf\{k \in \mathbb{N} \mid |\delta_k^{(j)}| \le \eta\}. \tag{22}$$

We can bound the burn-in time by the following lemma. The proof is given in Appendix H.2.

**Lemma F.2.** *Suppose that $\eta\sqrt{\lambda_j B} \gtrsim 1$. Then with probability at least $1 - \exp(-\Omega(|\theta_j^*|/\eta))$, it holds that $k_j^{**} \approx |\theta_j^*|/\eta$.*

**Attractivity and oscillation.** Once the residual enters the linear region $\mathcal{R}_j$, the update (18) can be well-approximated by the linear dynamics (20). Note that the stability condition for this linear system is $\eta\sqrt{\lambda_j B} \lesssim 1$, which means that the region $\mathcal{R}_j$ is attractive under this condition. However, from a rigorous stochastic perspective, the inherent randomness of the gradient noise $\xi_k^{(j)}$ ensures that the probability of the residual escaping the region $\mathcal{R}_j$ remains strictly positive. While numerical simulations empirically confirm that such an escape is an exceedingly rare event (see Figure 11b), we introduce the following idealized assumption to simplify our analysis.

**Assumption F.3** (Attractivity). *Suppose $\eta\sqrt{\lambda_j B} \lesssim 1$. Then, for all $k \ge k_j^*$, the residual satisfies $\delta_k^{(j)} \in \mathcal{R}_j$.*

We provide an intuitive rationale for this attractivity assumption. When $\eta\sqrt{\lambda_j B} \lesssim 1$, the deterministic part in 20 is contractive and drives $\delta_k^{(j)}$ to exponentially converge to zero. Moreover, the variance of the stochastic part remains controlled: since $\mathrm{Var}[\eta n_k^{(j)} \mid \delta_k^{(j)}] = \eta^2(1 - (m_k^{(j)})^2) \le \eta^2 \lesssim 1/(\lambda_j B)$. Consequently, the residual magnitude $|\delta_k^{(j)}| \le \frac{1}{\lambda_j B}$ typically holds.

We then provide a result describing the dynamics when the stability condition is violated.

**Proposition F.4** (Oscillation). *Suppose $\eta\sqrt{\lambda_j B} \gtrsim 1$. Then, for all $k \ge k_j^{**}$, the residual satisfies $\mathbb{E}[(\delta_k^{(j)})^2 \mid k_j^*] \lesssim \eta^2$ and $\max\{|\delta_k^{(j)}|, |\delta_{k+1}^{(j)}|\} \gtrsim \eta$.*

We defer the proof to Appendix H.2. Proposition F.4 shows that the residual oscillates at a scale of $\mathcal{O}(\eta)$ in the absence of the stability condition. See numerical validation in Figure 11c.

Next, we establish the following lemma to characterize the evolution of the loss under the update (18).

**Lemma F.5** (Linear recurrence)**.** *Suppose* $\eta\sqrt{\lambda_j B} \lesssim 1$*. Then, for all* $k \geq k_j^*$*, we have*

$$\mathbb{E}[(\delta_{k+1}^{(j)})^2 \mid k_j^*] = (1 - \Theta(\eta\sqrt{\lambda_j B}))\,\mathbb{E}[(\delta_k^{(j)})^2 \mid k_j^*] + \eta^2. \tag{23}$$

This result follows directly from the linear approximation $m_k^{(j)} \approx \sqrt{\lambda_j B}\,\delta_k^{(j)}$, with the complete proof provided in Appendix H.3.

## F.2 SCALING LAW IN THE LINEAR PHASE

In this section, we prove the full scaling law for SignSGD in the linear phase. The following is the formal version of the second case in Theorem 3.1.

**Theorem F.6.** *Suppose that* $\eta k \gtrsim \max\{B^{\frac{\beta-\alpha}{2\alpha}}, 1\}, \eta\sqrt{B} \lesssim 1$*. We denote the loss after* $k$ *steps of SignSGD by* $\mathcal{L}(\boldsymbol{\theta}_k)$*. Assume that* $\mathcal{L}(\boldsymbol{\theta}_k) \lesssim 1$*. Then we have*

- *If* $\beta > 2$

$$\mathbb{E}[\mathcal{L}(\boldsymbol{\theta}_k)] \approx \begin{cases} \dfrac{1}{\eta\sqrt{B}k^{2s}} + \dfrac{\eta}{\sqrt{B}}\Big[1 - (\eta\sqrt{B}k)^{-(1-2/\beta)}\Big], & B^{\frac{\beta}{2\alpha}} \lesssim \eta\sqrt{B}k \lesssim d^{\beta/2}, \\ (\eta\sqrt{B}k)^{-1}d^{-\frac{\beta}{2}(2s-1)}\exp(-\Theta(\eta\sqrt{B}kd^{-\beta/2})) + \dfrac{\eta}{\sqrt{B}}, & \eta\sqrt{B}k \gtrsim d^{\beta/2}. \end{cases} \tag{24}$$

- *If* $\beta = 2$

$$\mathbb{E}[\mathcal{L}(\boldsymbol{\theta}_k)] \approx \begin{cases} \dfrac{1}{\eta\sqrt{B}k^{2s}} + \dfrac{\eta}{\sqrt{B}}\log(\eta\sqrt{B}k), & B^{\frac{\beta}{2\alpha}} \lesssim \eta\sqrt{B}k \lesssim d^{\beta/2}, \\ (\eta\sqrt{B}k)^{-1}d^{-\frac{\beta}{2}(2s-1)}\exp(-\Theta(\eta\sqrt{B}kd^{-\beta/2})) + \dfrac{\eta}{\sqrt{B}}\log d, & \eta\sqrt{B}k \gtrsim d^{\beta/2}. \end{cases} \tag{25}$$

- *If* $1 < \beta < 2$

$$\mathbb{E}[\mathcal{L}(\boldsymbol{\theta}_k)] \approx \begin{cases} \dfrac{1}{(\eta\sqrt{B}k)^{2s}} + \dfrac{\eta}{\sqrt{B}}(\eta\sqrt{B}k)^{\frac{2}{\beta}-1}, & B^{\frac{\beta}{2\alpha}} \lesssim \eta\sqrt{B}k \lesssim d^{\beta/2}, \\ (\eta\sqrt{B}k)^{-1}d^{-\frac{\beta}{2}(2s-1)}\exp(-\Theta(\eta\sqrt{B}kd^{-\beta/2})) + \dfrac{\eta}{\sqrt{B}}d^{1-\frac{\beta}{2}}, & \eta\sqrt{B}k \gtrsim d^{\beta/2}. \end{cases} \tag{26}$$

Similar to Appendix E.2, this formula contains both the leading-order terms and subdominant corrections. It is easy to recover the original scaling law for SignSGD in Theorem F.6 by ignoring the subdominant terms.

We first establish two auxiliary lemmas.

**Lemma F.7.** *Consider the integral*

$$I(\eta, B, k, d) := \int_1^d j^{-1}\big(1 - \exp(-\eta\sqrt{B}kj^{-1})\big)\,\mathrm{d}j.$$

*Then the following asymptotic behaviors hold:*

$$I(\eta, B, k, d) \approx \begin{cases} \log(\eta\sqrt{B}k), & 1 \lesssim \eta\sqrt{B}k \lesssim d, \\ \log d, & \eta\sqrt{B}k \gtrsim d. \end{cases}$$

*Proof.* We perform the change of variables

$$z = \eta\sqrt{B}k\,j^{-1}, \qquad \mathrm{d}j = -\frac{\eta\sqrt{B}k}{z^2}\,\mathrm{d}z.$$

When $j = 1$, we have $z = \eta\sqrt{B}k$, and when $j = d$, we have $z = \eta\sqrt{B}k/d$. Hence the integral becomes

$$\frac{\eta}{\sqrt{B}} \int_{\eta\sqrt{B}k/d}^{\eta\sqrt{B}k} \frac{1 - e^{-z}}{z}\, dz. \tag{27}$$

We distinguish two regimes.

(i) $1 \lesssim \eta\sqrt{B}k \lesssim d$. In this regime we have

$$\frac{\eta\sqrt{B}k}{d} \lesssim 1 \lesssim \eta\sqrt{B}k,$$

and we split the integral in (27) as

$$\int_{\eta\sqrt{B}k/d}^{\eta\sqrt{B}k} = \int_{\eta\sqrt{B}k/d}^{1} + \int_{1}^{\eta\sqrt{B}k}.$$

For $z \in (0, 1)$, we use $1 - e^{-z} \approx z$, which gives

$$\int_{\eta\sqrt{B}k/d}^{1} \frac{1 - e^{-z}}{z}\, dz \approx 1.$$

For $z \geq 1$, we use $1 - e^{-z} \approx 1$, which gives

$$\int_{1}^{\eta\sqrt{B}k} \frac{1 - e^{-z}}{z}\, dz \approx \int_{1}^{\eta\sqrt{B}k} \frac{1}{z}\, dz = \log(\eta\sqrt{B}k).$$

Combining the two contributions, we obtain

$$\frac{\eta}{\sqrt{B}} \int_{1}^{d} j^{-1}\left(1 - \exp(-\eta\sqrt{B}kj^{-1})\right) dj \approx \frac{\eta}{\sqrt{B}} \log(\eta\sqrt{B}k),$$

(ii) $\eta\sqrt{B}k \gtrsim d$. In this case, $\frac{\eta\sqrt{B}k}{d} \gtrsim 1$, so that $z \gtrsim 1$ throughout the integration range. Hence $1 - e^{-z} \approx 1$, and (27) satisfies

$$\frac{\eta}{\sqrt{B}} \int_{\eta\sqrt{B}k/d}^{\eta\sqrt{B}k} \frac{1 - e^{-z}}{z}\, dz \approx \frac{\eta}{\sqrt{B}} \int_{\eta\sqrt{B}k/d}^{\eta\sqrt{B}k} \frac{1}{z}\, dz = \frac{\eta}{\sqrt{B}} \log d.$$

$\square$

**Lemma F.8.** *Let $1 < \beta < 2$. Consider the integral*

$$I(\eta, B, k, d) := \int_{1}^{d} j^{-\beta/2}\left(1 - \exp(-\eta\sqrt{B}kj^{-\beta/2})\right) dj.$$

*Then the following asymptotic behaviors hold:*

$$I(\eta, B, k, d) \approx \begin{cases} (\eta\sqrt{B}k)^{2/\beta-1}, & 1 \lesssim \eta\sqrt{B}k \lesssim d^{\beta/2}, \\ d^{1-\beta/2}, & \eta\sqrt{B}k \gtrsim d^{\beta/2}. \end{cases}$$

*Proof.* We perform the change of variables

$$z = \eta\sqrt{B}k\, j^{-\beta/2}, \qquad dj = -\frac{2}{\beta}(\eta\sqrt{B}k)^{2/\beta} z^{-(2/\beta+1)}\, dz.$$

When $j = 1$, we have $z = \eta\sqrt{B}k$, and when $j = d$, we have $z = \eta\sqrt{B}k\, d^{-\beta/2}$. Hence the integral becomes

$$\frac{\eta}{\sqrt{B}}(\eta\sqrt{B}k)^{2/\beta-1} \int_{\eta\sqrt{B}kd^{-\beta/2}}^{\eta\sqrt{B}k} (1 - e^{-z})\, z^{-2/\beta}\, dz. \tag{28}$$

We again distinguish two regimes.

(i) $1 \lesssim \eta\sqrt{B}k \lesssim d^{\beta/2}$. In this regime, $\eta\sqrt{B}kd^{-\beta/2} \lesssim 1 \lesssim \eta\sqrt{B}k$. We split the integral at $z = 1$.

For $z \in (0, 1)$, we use $1 - e^{-z} \asymp z$, which gives

$$\int_{\eta\sqrt{B}kd^{-\beta/2}}^{1} \left(1 - e^{-z}\right) z^{-2/\beta}\, dz \;\asymp\; \int_{\eta\sqrt{B}kd^{-\beta/2}}^{1} z^{1-2/\beta}\, dz \;\asymp\; 1,$$

since $1 - 2/\beta > -1$ for $\beta > 1$.

For $z \geq 1$, we use $1 - e^{-z} \asymp 1$, which gives

$$\int_{1}^{\eta\sqrt{B}k} z^{-2/\beta}\, dz \;\asymp\; \int_{1}^{\eta\sqrt{B}k} z^{-2/\beta}\, dz \;\asymp\; 1,$$

because $2/\beta > 1$ for $\beta < 2$.

Combining the two parts, we conclude that

$$\frac{\eta}{\sqrt{B}} \int_{1}^{d} j^{-\beta/2}\left(1 - \exp(-\eta\sqrt{B}kj^{-\beta/2})\right) dj \;\asymp\; \frac{\eta}{\sqrt{B}}(\eta\sqrt{B}k)^{2/\beta-1}.$$

(ii) $\eta\sqrt{B}k \gtrsim d^{\beta/2}$.

In this case, $\eta\sqrt{B}kd^{-\beta/2} \gtrsim 1$, so that the entire integration range satisfies $z \gtrsim 1$. Consequently, $1 - e^{-z} \asymp 1$ uniformly over the integration domain. Therefore,

$$\int_{\eta\sqrt{B}kd^{-\beta/2}}^{\eta\sqrt{B}k} \left(1 - e^{-z}\right) z^{-2/\beta}\, dz \;\asymp\; \int_{\eta\sqrt{B}kd^{-\beta/2}}^{\eta\sqrt{B}k} z^{-2/\beta}\, dz = \left[\frac{z^{1-2/\beta}}{1-2/\beta}\right]_{\eta\sqrt{B}kd^{-\beta/2}}^{\eta\sqrt{B}k}.$$

Since $1 < \beta < 2$, we have $1 - 2/\beta < 0$, and thus the integral is dominated by its lower limit. So we have

$$\int_{\eta\sqrt{B}kd^{-\beta/2}}^{\eta\sqrt{B}k} \left(1 - e^{-z}\right) z^{-2/\beta}\, dz \;\asymp\; (\eta\sqrt{B}k)^{1-2/\beta}d^{1-\beta/2}.$$

Substituting this into (28), we finally obtain

$$\frac{\eta}{\sqrt{B}} \int_{1}^{d} j^{-\beta/2}\left(1 - \exp(-\eta\sqrt{B}kj^{-\beta/2})\right) dj \;\asymp\; \frac{\eta}{\sqrt{B}}d^{1-\beta/2}.$$

$\square$

*Proof of Theorem F.6.* Since $\eta\sqrt{B} \lesssim 1$, we have $\eta\sqrt{\lambda_j B} \lesssim 1$ for all $j$. Recall that the burn-in time $k_j^*$ in (21) is the first time the $j$-th coordinate enters the linear region $\mathcal{R}_j := \left[-\frac{1}{\sqrt{\lambda_j B}}, \frac{1}{\sqrt{\lambda_j B}}\right]$.

Hence, by Lemma F.5, the following one-step conditional recursion holds *for all $k \geq k_j^*$*:

$$\mathbb{E}[(\delta_{k+1}^{(j)})^2 \mid k_j^*] = \left(1 - \Theta(\eta\sqrt{\lambda_j B})\right) \mathbb{E}[(\delta_k^{(j)})^2 \mid k_j^*] + \eta^2. \tag{29}$$

Note that when $j \gtrsim B^{1/\alpha}$, we have $k_j^* = 0$ (see Lemma H.1). On the other hand, when $j \lesssim B^{1/\alpha}$, Lemma F.1 provides a high-probability estimate for $k_j^*$. In the following analysis we deal with the two groups separately and then combine them.

**Large-indices group.** For $j \gtrsim B^{1/\alpha}$, we have $k_j^* = 0$, so we can unroll (29) from 0 to $k$:

$$\mathbb{E}[(\delta_k^{(j)})^2] = \left(1 - \Theta(\eta\sqrt{\lambda_j B})\right)^k (\delta_0^{(j)})^2 + \Theta(\eta^2) \sum_{r=0}^{k-1} \left(1 - \Theta(\eta\sqrt{\lambda_j B})\right)^r$$

$$\asymp \left(1 - \eta\sqrt{\lambda_j B}\right)^k (\delta_0^{(j)})^2 + \frac{\eta}{\sqrt{\lambda_j B}}\left(1 - \left(1 - \Theta(\eta\sqrt{\lambda_j B})\right)^k\right) \tag{30}$$

$$\asymp (\delta_0^{(j)})^2 \exp\left(-\Theta(\eta\sqrt{\lambda_j B})k\right) + \frac{\eta}{\sqrt{\lambda_j B}}\left(1 - \exp\left(-\Theta(\eta\sqrt{\lambda_j B})k\right)\right),$$

where the last transition uses $\eta\sqrt{\lambda_j B} \lesssim 1$ and the standard equivalence $(1-\Theta(x))^k \asymp \exp(-\Theta(xk))$ for $x \in (0, 1]$, as is discussed in the proof of Theorem E.4.

**Small-indices group.** For $j \lesssim B^{1/\alpha}$, we unroll (29) from the burn-in time $k_j^*$. Conditioning on $k_j^*$ and for all $k \geq k_j^*$,

$$\mathbb{E}[(\delta_k^{(j)})^2 \mid k_j^*] \asymp \mathbb{E}[(\delta_{k_j^*}^{(j)})^2 \mid k_j^*]\exp\big(-\Theta(\eta\sqrt{\lambda_j B})(k-k_j^*)\big)+\frac{\eta}{\sqrt{\lambda_j B}}\Big(1-\exp\big(-\Theta(\eta\sqrt{\lambda_j B})(k-k_j^*)\big)\Big).$$
(31)

We now control $k_j^*$ uniformly over all $j \lesssim B^{1/\alpha}$. Define the "good" event on which all burn-in times are well-controlled as

$$\mathcal{A} := \bigcap_{j \lesssim B^{1/\alpha}} \left\{ k_j^* \asymp |\theta_j^*|/\eta \right\}.$$
(32)

Applying Lemma F.1 and taking a union bound over $j \lesssim B^{1/\alpha}$ yields

$$\mathbb{P}(\mathcal{A}^c) \leq \sum_{j \lesssim B^{1/\alpha}} \exp\big(-\Omega(|\theta_j^*|/\eta)\big) =: p_{\text{st}}^*.$$
(33)

Define

$$m_B := \min_{1 \leq j \lesssim B^{1/\alpha}} |\theta_j^*|, \qquad M_B := \max_{1 \leq j \lesssim B^{1/\alpha}} |\theta_j^*|.$$

Then we have an upper bound for the probability $p_{\text{st}}^*$

$$p_{\text{st}}^* \lesssim B^{1/\alpha} \exp\big(-\Omega(m_B/\eta)\big).$$
(34)

Moreover, by Assumption B.3,

$$m_B \asymp \min\left\{1, \ B^{\frac{\beta-\alpha}{2\alpha}}\right\}, \qquad M_B \asymp \max\left\{1, \ B^{\frac{\beta-\alpha}{2\alpha}}\right\}.$$
(35)

Under the condition $\eta k \gtrsim M_B$, on the event $\mathcal{A}$ we have for all $j \lesssim B^{1/\alpha}$ that

$$k_j^* \asymp |\theta_j^*|/\eta \leq M_B/\eta \lesssim ck,$$

and therefore

$$k - k_j^* = \Theta(k).$$
(36)

Next we estimate $\mathbb{E}[(\delta_{k_j^*}^{(j)})^2 \mid k_j^*]$. By the definition of $k_j^*$ as the first entrance time into $\mathcal{R}_j$ and since each update has magnitude at most $\eta$, we have deterministically

$$\frac{1}{\sqrt{\lambda_j B}} - \eta \leq |\delta_{k_j^*}^{(j)}| \leq \frac{1}{\sqrt{\lambda_j B}}.$$

Assume $\eta\sqrt{B} \leq 1/2$. Then $\eta\sqrt{\lambda_j B} \leq 1/2$ for all $j$, and thus the above inequality implies

$$|\delta_{k_j^*}^{(j)}| \asymp \frac{1}{\sqrt{\lambda_j B}},$$

which yields

$$\mathbb{E}[(\delta_{k_j^*}^{(j)})^2 \mid k_j^*] \asymp \frac{1}{\lambda_j B}.$$
(37)

Plugging (36) and (37) into (31) yields, on $\mathcal{A}$,

$$\mathbb{E}[(\delta_k^{(j)})^2 \mid \mathcal{A}] \asymp \frac{1}{\lambda_j B}\exp\big(-\Theta(\eta\sqrt{\lambda_j B}k)\big)+\frac{\eta}{\sqrt{\lambda_j B}}\Big(1-\exp\big(-\Theta(\eta\sqrt{\lambda_j B}k)\big)\Big), \qquad j \lesssim B^{1/\alpha}.$$
(38)

**Total loss.** Combining (30) and (38), we obtain the following bound on the final loss

$$
\begin{aligned}
\mathbb{E}[\mathcal{L}(\boldsymbol{\theta}_k)] &= \frac{1}{2}\sum_{j=1}^{d}\lambda_j\,\mathbb{E}[(\delta_k^{(j)})^2]\\
&\asymp \frac{1}{2}\sum_{j>B^{1/\alpha}}\lambda_j(\delta_0^{(j)})^2\exp\!\big(-\Theta(\eta\sqrt{\lambda_j B}k)\big) + \frac{\eta}{2\sqrt{B}}\sum_{j>B^{1/\alpha}}\sqrt{\lambda_j}\Big(1-\exp\!\big(-\Theta(\eta\sqrt{\lambda_j B}k)\big)\Big)\\
&\quad + \frac{1}{2B}\sum_{j\le B^{1/\alpha}}\exp\!\big(-\Theta(\eta\sqrt{\lambda_j B}k)\big) + \frac{\eta}{2\sqrt{B}}\sum_{j\le B^{1/\alpha}}\sqrt{\lambda_j}\Big(1-\exp\!\big(-\Theta(\eta\sqrt{\lambda_j B}k)\big)\Big) + \mathcal{L}^c,\\
&\lesssim \underbrace{\frac{1}{2}\sum_{j>B^{1/\alpha}}\lambda_j(\delta_0^{(j)})^2\exp\!\big(-\Theta(\eta\sqrt{\lambda_j B}k)\big)}_{\text{Term A}} + \underbrace{\frac{1}{2B}\sum_{j\le B^{1/\alpha}}\exp\!\big(-\Theta(\eta\sqrt{\lambda_j B}k)\big)}_{\text{Term B}}\\
&\quad + \underbrace{\frac{\eta}{2\sqrt{B}}\sum_{j=1}^{d}\sqrt{\lambda_j}\Big(1-\exp\!\big(-\Theta(\eta\sqrt{\lambda_j B}k)\big)\Big)}_{\text{Term C}} + \mathcal{L}^c,
\end{aligned}
\tag{39}
$$

where $\mathcal{L}^c := \mathbb{E}[\mathcal{L}(\boldsymbol{\theta}_k)\mathbf{1}_{\mathcal{A}^c}]$ denotes the contribution to the expected loss from the complement event $\mathcal{A}^c$.

**Term A.** Consider the first term in the summation. Recall that $\lambda_j(\delta_0^{(j)})^2 \asymp j^{-\alpha}$ and $\lambda_j \asymp j^{-\beta}$. Applying the sum-to-integral approximation of Lemma E.1, we obtain

$$
\frac{1}{2}\sum_{j>B^{1/\alpha}}\lambda_j(\delta_0^{(j)})^2\exp\!\big(-\Theta(\eta\sqrt{\lambda_j B}\,k)\big) \;\asymp\; \frac{1}{2}\int_{B^{1/\alpha}}^{d} j^{-\alpha}\exp\!\big(-\Theta(\eta\sqrt{B}\,k)\,j^{-\beta/2}\big)\,\mathrm{d}j.
$$

By an argument analogous to Lemma E.3, the integral admits the following scaling:

$$
\int_{B^{1/\alpha}}^{d} j^{-\alpha}\exp\!\big(-\Theta(\eta\sqrt{B}\,k)\,j^{-\beta/2}\big)\,\mathrm{d}j \;\asymp\;
\begin{cases}
(\eta\sqrt{B}\,k)^{-2s}, & B^{\frac{\beta}{2\alpha}} \lesssim \eta\sqrt{B}\,k \lesssim d^{\beta},\\
(\eta\sqrt{B}\,k)^{-1}d^{-\frac{\beta}{2}(2s-1)}\exp\!\big(-\Theta(\eta\sqrt{B}\,k\,d^{-\beta/2})\big), & \eta\sqrt{B}\,k \gtrsim d^{\beta}.
\end{cases}
$$

Note that the lower bound $B^{\frac{\beta}{2\alpha}} \lesssim \eta\sqrt{B}\,k$ is equivalent to $\eta k \gtrsim B^{\frac{\beta-\alpha}{2\alpha}}$, which is ensured by our assumptions.

**Term B.** The second term can be bounded as

$$
\frac{1}{2B}\sum_{j\le B^{1/\alpha}}\exp\!\big(-\Theta(\eta\sqrt{\lambda_j B}\,k)\big) \;\lesssim\; B^{1/\alpha-1}\exp\!\Big(-\Theta\!\big(\eta\,B^{\frac{\alpha-\beta}{2\alpha}}\,k\big)\Big).
$$

Moreover, under the condition $\eta k \gtrsim B^{\frac{\beta-\alpha}{2\alpha}}$, the exponent satisfies $\eta\,B^{\frac{\alpha-\beta}{2\alpha}}\,k = \Omega(1)$, and can be arbitrarily large. Hence the exponential factor is strongly suppressed, and Term B is negligible compared with the leading terms.

**Term C.** Now we consider the third term

$$
\frac{\eta}{2\sqrt{B}}\sum_{j=1}^{d}\sqrt{\lambda_j}\Big(1-\exp\!\big(-\Theta(\eta\sqrt{\lambda_j B}k)\big)\Big)
$$

Using the eigenvalue decay $\lambda_j \asymp j^{-\beta}$ and the sum-to-integral approximation (Lemma E.1), we obtain

$$
\frac{\eta}{2\sqrt{B}}\sum_{j=1}^{d}\sqrt{\lambda_j}\Big(1-\exp\!\big(-\Theta(\eta\sqrt{\lambda_j B}k)\big)\Big) \asymp \frac{\eta}{2\sqrt{B}}\int_{1}^{d} j^{-\beta/2}\big(1-\exp(-2\eta\sqrt{B}kj^{-\beta/2})\big)\,\mathrm{d}j.
$$

We need to consider three cases.

*Case 1:* $\beta > 2$. We directly apply Lemma E.5 and get

$$\frac{\eta}{\sqrt{B}} \int_1^d j^{-\beta/2} \big(1 - \exp(-2\eta\sqrt{B}kj^{-\beta/2})\big) \, \mathrm{d}j \asymp \begin{cases} \frac{\eta}{\sqrt{B}} \Big[1 - (\eta\sqrt{B}k)^{-(1-2/\beta)}\Big], & 1 \lesssim \eta\sqrt{B}k \lesssim d^{\beta/2}, \\ \frac{\eta}{\sqrt{B}}, & \eta\sqrt{B}k \gtrsim d^{\beta/2}. \end{cases}$$

*Case 2:* $\beta = 2$. Applying Lemma F.7, we obtain

$$\frac{\eta}{\sqrt{B}} \int_1^d j^{-1} \Big(1 - \exp(-2\eta\sqrt{B}kj^{-1})\Big) \, \mathrm{d}j \asymp \begin{cases} \frac{\eta}{\sqrt{B}} \log(\eta\sqrt{B}k), & 1 \lesssim \eta\sqrt{B}k \lesssim d, \\ \frac{\eta}{\sqrt{B}} \log d, & \eta\sqrt{B}k \gtrsim d. \end{cases}$$

*Case 3:* $1 < \beta < 2$. Applying Lemma F.8, we obtain

$$\frac{\eta}{\sqrt{B}} \int_1^d j^{-\beta/2} \Big(1 - \exp(-2\eta\sqrt{B}kj^{-\beta/2})\Big) \, \mathrm{d}j \asymp \begin{cases} \frac{\eta}{\sqrt{B}} (\eta\sqrt{B}k)^{2/\beta-1}, & 1 \lesssim \eta\sqrt{B}k \lesssim d^{\beta/2}, \\ \frac{\eta}{\sqrt{B}} d^{1-\beta/2}, & \eta\sqrt{B}k \gtrsim d^{\beta/2}. \end{cases}$$

**Term $\mathcal{L}^c$.** This term can be controlled by the failure probability:

$$\mathcal{L}^c \le \sup_k \mathcal{L}(\boldsymbol{\theta}_k) \cdot \mathbb{P}(\mathcal{A}^c) \lesssim p_{\mathrm{st}}^* \lesssim B^{1/\alpha} \exp\big(-\Omega(m_B/\eta)\big), \tag{40}$$

where we used the assumption $\mathcal{L}(\boldsymbol{\theta}_k) \lesssim 1$ with (33)–(34). Here $m_B \asymp \min\{1, B^{\frac{\beta-\alpha}{2\alpha}}\}$ as in (35). Under the stability condition $\eta\sqrt{B} \lesssim 1$, the exponent $\Omega(m_B/\eta)$ is large, so $\mathcal{L}^c$ is exponentially small and therefore negligible compared with the leading terms.

Finally, combining the above estimates, we can get the desired results.

$\square$

## F.3 SCALING LAW BEYOND THE LINEAR PHASE

In this section, we prove the scaling law in the remaining two phases, namely the burn-in phase and the unstable phase, corresponding to the first and third cases of Theorem 3.1. The proof relies on the training dynamics analysis in Appendix F.1.

### F.3.1 BURN-IN PHASE

**Theorem F.9.** *Suppose $\eta k \lesssim 1$. Then*

$$\mathbb{E}[\mathcal{L}(\boldsymbol{\theta}_k)] \asymp (1 - \eta k)^2.$$

*Proof.* The condition $\eta k \lesssim 1$ means that the training horizon is shorter than the burn-in time of the leading coordinates, so the dynamics are still dominated by the early-stage behavior (saturation region before burn-in).

We first prove a lower bound by focusing on the first coordinate. Since $|\theta_1^*| \asymp 1$, we have $|\delta_0^{(1)}| = |\theta_1^*| \asymp 1$. Moreover, by Lemma H.3 we have $k_1^* \gtrsim |\theta_1^*|/\eta \asymp 1/\eta$, and the assumption $\eta k \lesssim 1$ implies $k \lesssim k_1^*$. Hence, throughout the first $k$ steps, the first coordinate remains in the saturation region, where the update behaves (with high probability) like $\delta_{t+1}^{(1)} = \delta_t^{(1)} - \eta \operatorname{sgn}(\delta_t^{(1)})$. Therefore,

$$|\delta_k^{(1)}| \asymp |\delta_0^{(1)}| - \eta k \asymp 1 - \eta k,$$

and consequently

$$\mathbb{E}[\mathcal{L}(\boldsymbol{\theta}_k)] \gtrsim \lambda_1 \, \mathbb{E}[(\delta_k^{(1)})^2] \asymp (1 - \eta k)^2.$$

For the upper bound, we bound the contribution of the remaining coordinates. For any $j \ge 2$, regardless of whether the iterate is in the saturation region (where the update has step size $\eta$) or in the linear region (where it follows a linear recurrence; see Lemma F.5), we may use the crude bound

$$\mathbb{E}[(\delta_k^{(j)})^2] \lesssim (\delta_0^{(j)})^2 + k\eta^2.$$

Summing over $j \geq 2$ gives

$$\sum_{j=2}^{d} \lambda_j \, \mathbb{E}[(\delta_k^{(j)})^2] \lesssim \sum_{j=2}^{d} \lambda_j (\delta_0^{(j)})^2 + \sum_{j=2}^{d} \lambda_j k \eta^2$$

$$\lesssim \sum_{j=2}^{d} j^{-\alpha} + \left(\sum_{j=2}^{d} j^{-\beta}\right) \eta,$$

where we used $\lambda_j (\delta_0^{(j)})^2 \asymp j^{-\alpha}$ and $\lambda_j \asymp j^{-\beta}$, and also $k\eta^2 \leq \eta$ since $\eta k \lesssim 1$. Both sums are $\mathcal{O}(1)$ and thus the loss contribution is $\mathcal{O}(1)$. Therefore,

$$\mathbb{E}[\mathcal{L}(\boldsymbol{\theta}_k)] \lesssim (1 - \eta k)^2 + \mathcal{O}(1).$$

Since $\eta k \lesssim 1$ implies $(1 - \eta k)^2 = \Theta(1)$, the $\mathcal{O}(1)$ term can be absorbed into the multiplicative constants, yielding $\mathbb{E}[\mathcal{L}(\boldsymbol{\theta}_k)] \lesssim (1 - \eta k)^2$. Combining with the lower bound proves the claim. $\square$

### F.3.2 UNSTABLE PHASE

**Theorem F.10.** *Suppose* $\eta k \gtrsim 1$, $\eta\sqrt{B} \gtrsim 1$ *and* $\eta\sqrt{B}k \lesssim d^{\beta/2}$. *Then*

$$\mathbb{E}[\mathcal{L}(\boldsymbol{\theta}_k)] \asymp \frac{1}{(\eta\sqrt{B}k)^{2s}} + \eta^2.$$

*Proof.* Since $\eta k \gtrsim 1$, the burn-in stage occupies only a negligible fraction of the total training horizon. As in the proof of Theorem F.6, its contribution to the loss can be dominated (ignoring exponentially small terms). The key difference from the stable scaling regime is that $\eta\sqrt{B} \gtrsim 1$, which creates oscillatory directions in the linear region.

Define the index threshold

$$j^* := \sup\left\{j : \eta\sqrt{\lambda_j B} \lesssim 1\right\}.$$

Using $\lambda_j \asymp j^{-\beta}$, we have $j^* \asymp (\eta\sqrt{B})^{2/\beta}$. For $j \leq j^*$ we have $\eta\sqrt{\lambda_j B} \gtrsim 1$, and these coordinates exhibit oscillations with typical magnitude $\Theta(\eta)$ (see Proposition F.4), implying $\mathbb{E}[(\delta_k^{(j)})^2] \asymp \eta^2$. For $j > j^*$ we have $\eta\sqrt{\lambda_j B} \lesssim 1$, and these coordinates evolve stably in the linear region, contributing the same decay term as in the stable-scaling analysis.

Splitting the loss accordingly, we obtain

$$\mathbb{E}[\mathcal{L}(\boldsymbol{\theta}_k)] = \frac{1}{2} \sum_{j \leq j^*} \lambda_j \, \mathbb{E}[(\delta_k^{(j)})^2] + \frac{1}{2} \sum_{j > j^*} \lambda_j \, \mathbb{E}[(\delta_k^{(j)})^2]$$

$$\asymp \sum_{j \leq j^*} \lambda_j \eta^2 + \sum_{j > j^*} \lambda_j (\delta_0^{(j)})^2 \exp\left(-\Theta(\eta\sqrt{\lambda_j B}\, k)\right).$$

The first term is $\Theta(\eta^2)$ since $\sum_{j=1}^{\infty} \lambda_j < \infty$ for $\beta > 1$. The second term follows the same derivation as in the proof of Theorem F.6. Under the condition $\eta\sqrt{B}\, k \lesssim d^{\beta/2}$, we have

$$\sum_{j > j^*} \lambda_j (\delta_0^{(j)})^2 \exp\left(-\Theta(\eta\sqrt{\lambda_j B}\, k)\right) \asymp (\eta\sqrt{B}\, k)^{-2s}.$$

Combining the two contributions proves the claim. $\square$

## G   SCALING ANALYSIS

### G.1   DATA SCALING

In this section we provide the analysis of data efficiency. We discuss how the optimal loss scales with $D$ when the hyperparameters are well tuned. Let $K$ be the total training steps. Denote $\mathcal{E}_K := \mathbb{E}[\mathcal{L}(\boldsymbol{\theta}_K)]$, and $\mathcal{E}_{\mathrm{sgn}}^*(D), \mathcal{E}_{\mathrm{sgd}}^*(D)$ as the optimal loss given a fixed data budget $D$ for SignSGD and SGD, respectively.

**Theorem G.1** (Data scaling for SignSGD). *Suppose $B^{\max\{\frac{\beta-\alpha}{2\alpha}+1,\,1\}} \lesssim \eta D \lesssim d^{\beta/2}B, \eta\sqrt{B} \lesssim 1$, and assume $\mathcal{L}(\boldsymbol{\theta}_K) \lesssim 1$ for all $K$. Then, given a data budget $D$, we have:*

- *If $\beta > 2$,*
$$\mathcal{E}^*_{\mathrm{sgn}}(D) \asymp D^{-\frac{2s}{2s+1}}.$$

- *If $\beta = 2$,*
$$\mathcal{E}^*_{\mathrm{sgn}}(D) \asymp (D/\log D)^{-\frac{2s}{2s+1}}.$$

- *If $\beta < 2$,*
$$\mathcal{E}^*_{\mathrm{sgn}}(D) \asymp D^{-\frac{s\beta}{s\beta+1}}.$$

*Proof.* We omit the subscript sgn when there is no ambiguity. The proof starts from Theorem F.6,

$$\mathcal{E}_K \;\asymp\; \left(\eta\sqrt{B}\,K\right)^{-2s} \;+\; \begin{cases} \dfrac{\eta}{\sqrt{B}}, & \beta > 2, \\[2mm] \dfrac{\eta}{\sqrt{B}}\log\left(\eta\sqrt{B}\,K\right), & \beta = 2, \\[2mm] \dfrac{\eta}{\sqrt{B}}\left(\eta\sqrt{B}\,K\right)^{2/\beta-1}, & 1 < \beta < 2, \end{cases}$$

we substitute $K = D/B$ to obtain

$$\mathcal{E}_K \;\asymp\; \left(\frac{\eta D}{\sqrt{B}}\right)^{-2s} \;+\; \begin{cases} \dfrac{\eta}{\sqrt{B}}, & \beta > 2, \\[2mm] \dfrac{\eta}{\sqrt{B}}\log\left(\dfrac{\eta D}{\sqrt{B}}\right), & \beta = 2, \\[2mm] \dfrac{\eta}{\sqrt{B}}\left(\dfrac{\eta D}{\sqrt{B}}\right)^{2/\beta-1}, & 1 < \beta < 2. \end{cases}$$

For a fixed data budget $D$, the estimate depends on $(\eta, B)$ only through the effective learning rate $\tilde{\eta} := \frac{\eta}{\sqrt{B}}$. Indeed, in terms of $\tilde{\eta}$ we have

$$\mathcal{E}_D(\tilde{\eta}) \;\asymp\; (\tilde{\eta}D)^{-2s} \;+\; \begin{cases} \tilde{\eta}, & \beta > 2, \\ \tilde{\eta}\log(\tilde{\eta}D), & \beta = 2, \\ \tilde{\eta}(\tilde{\eta}D)^{2/\beta-1}, & 1 < \beta < 2, \end{cases}$$

which makes it explicit that keeping $\eta/\sqrt{B}$ fixed leaves the loss bound invariant. This is the square root scaling rule.

With this change of variable, the step size constraint becomes $\eta\sqrt{B} = \tilde{\eta}B \lesssim 1$. The burn-in accommodation assumption becomes $\eta K = \tilde{\eta}\frac{D}{\sqrt{B}} \gtrsim 1$,

We now optimize loss over $\tilde{\eta} = \eta/\sqrt{B}$ for a fixed $D$.

*Case 1: $\beta > 2$.* We minimize $(\tilde{\eta}D)^{-2s} + \tilde{\eta}$. Balancing the two terms gives

$$(\tilde{\eta}D)^{-2s} \asymp \tilde{\eta} \quad \Longrightarrow \quad \tilde{\eta}^* \asymp D^{-\frac{2s}{2s+1}}.$$

The resulting optimal loss is

$$\mathcal{E}^*(D) \;\asymp\; \tilde{\eta}^* \;\asymp\; D^{-\frac{2s}{2s+1}}, \qquad (\beta > 2).$$

*Case 2: $\beta = 2$.* In the critical case $\beta = 2$, the bound takes the form

$$\mathcal{E}_D(\tilde{\eta}) \;\asymp\; (\tilde{\eta}D)^{-2s} \;+\; \tilde{\eta}\log(\tilde{\eta}D).$$

To determine the optimal $\tilde{\eta}$, we balance the two terms, that is, $(\tilde{\eta}D)^{-2s} \asymp \tilde{\eta}\log(\tilde{\eta}D)$. Rewriting the equation gives $\tilde{\eta}^{2s+1} \asymp D^{-2s}\left[\log(\tilde{\eta}D)\right]^{-1}$. Taking $(2s+1)$-th roots on both sides gives $\tilde{\eta} \asymp D^{-\frac{2s}{2s+1}}\left[\log(\tilde{\eta}D)\right]^{-\frac{1}{2s+1}}$.

Since the optimal $\tilde{\eta}$ decays polynomially in $D$, we have $\log(\tilde{\eta}D) \asymp \log D$ up to constant factors. Therefore,

$$\tilde{\eta}^* \asymp D^{-\frac{2s}{2s+1}} (\log D)^{-\frac{1}{2s+1}}.$$

Substituting this choice back into the bound, we obtain

$$\mathcal{E}^*(D) \asymp (\tilde{\eta}^* D)^{-2s} \asymp (D/\log D)^{-\frac{2s}{2s+1}}, \qquad (\beta = 2).$$

*Case 3:* $1 < \beta < 2$. In this regime, the noise term is $\tilde{\eta}(\tilde{\eta}D)^{2/\beta-1} = \tilde{\eta}^{2/\beta} D^{2/\beta-1}$, hence we minimize

$$(\tilde{\eta}D)^{-2s} + \tilde{\eta}^{2/\beta} D^{2/\beta-1}.$$

Balancing the two terms yields

$$(\tilde{\eta}D)^{-2s} \asymp \tilde{\eta}^{2/\beta} D^{2/\beta-1} \quad \Longrightarrow \quad \tilde{\eta}^* \asymp D^{-\frac{2s\beta+2-\beta}{2s\beta+2}}.$$

Plugging back, the optimal loss scales as

$$\mathcal{E}^*(D) \asymp (\tilde{\eta}^* D)^{-2s} \asymp D^{-\frac{s\beta}{s\beta+1}}, \qquad (1 < \beta < 2).$$

Finally, we verify that the optimal $\tilde{\eta}^*$ satisfies the constraints. The step size condition $\eta\sqrt{B} = \tilde{\eta}^* B \lesssim 1$ requires

$$B \lesssim \begin{cases} D^{\frac{2s}{2s+1}}, & \beta > 2, \\ D^{\frac{2s}{2s+1}} (\log D)^{\frac{1}{2s+1}}, & \beta = 2, \\ D^{\frac{2s\beta+2-\beta}{2s\beta+2}}, & 1 < \beta < 2, \end{cases}$$

while the burn-in condition $\eta K = \tilde{\eta}^* D/\sqrt{B} \gtrsim 1$ imposes

$$B \lesssim \begin{cases} D, & \beta > 2, \\ (D/\log D)^{\frac{2}{2s+1}}, & \beta = 2, \\ D^{\frac{2}{2s\beta+2}}, & 1 < \beta < 2. \end{cases}$$

These conditions are compatible in all cases for sufficiently large $D$, ensuring that the chosen $\tilde{\eta}^*$ and $B$ satisfy both the step size and burn-in requirements.

$\square$

**Theorem G.2** (Data scaling for SGD). *Suppose that the learning rate $\eta$ satisfy $\eta \lesssim 1$, . Then, given a data budget $D$, we have*

$$\mathcal{E}^*_{\text{sgd}}(D) \asymp D^{-\frac{s}{s+1}}.$$

*Proof.* Recall the data scaling formula

$$\mathcal{E}_K \asymp (\eta K)^{-s} + \frac{\eta}{B},$$

we introduce the effective learning rate $\tilde{\eta} := \eta/B$, which gives

$$\mathcal{E}_D(\tilde{\eta}) \asymp (\tilde{\eta}D)^{-s} + \tilde{\eta}.$$

Balancing the two terms for optimal performance:

$$(\tilde{\eta}D)^{-s} \asymp \tilde{\eta} \quad \Longrightarrow \quad \tilde{\eta}^* \asymp D^{-\frac{s}{s+1}}.$$

Substituting back, the optimal loss scales as

$$\mathcal{E}^*(D) \asymp (\tilde{\eta}^* D)^{-s} \asymp D^{-\frac{s}{s+1}}.$$

Note that the constraint $\eta \lesssim 1$ is also easily satisfied. $\square$

## G.2 LARGE BATCH TRAINING

In this section, we calculate the scaling behavior in large batch training.

### G.2.1 FIXED $D$ SETTING

For SignSGD and SGD, we let $\mathcal{E}^*_{\text{sgn}}(D, B), \mathcal{E}^*_{\text{sgd}}(D, B)$ denote the optimal loss given data $D$ and batch size $B$, where the learning rate $\eta$ can be optimally tuned.

**Theorem G.3** (Fixed data budget scaling for SignSGD). *Fix a data budget $D = KB$. Assume the stability condition $\eta\sqrt{B} \lesssim 1$ holds. Then, optimally tuning the learning rate $\eta$, the optimal loss of SignSGD satisfies*

- *If $\beta > 2$*

$$\mathcal{E}^*_{\text{sgn}}(D, B) \;\asymp\; \begin{cases} D^{-\frac{2s}{2s+1}}, & B \lesssim D^{\frac{2s}{2s+1}}, \\ (B/D)^{2s}, & B \gtrsim D^{\frac{2s}{2s+1}}. \end{cases}$$

- *If $\beta = 2$*

$$\mathcal{E}^*_{\text{sgn}}(D, B) \;\asymp\; \begin{cases} (D/\log D)^{-\frac{2s}{2s+1}}, & B \lesssim D^{\frac{2s}{2s+1}} (\log D)^{\frac{1}{2s+1}}, \\ (B/D)^{2s}, & B \gtrsim D^{\frac{2s}{2s+1}} (\log D)^{\frac{1}{2s+1}}. \end{cases}$$

- *If $1 < \beta < 2$*

$$\mathcal{E}^*_{\text{sgn}}(D, B) \;\asymp\; \begin{cases} D^{-\frac{s\beta}{s\beta+1}}, & B \lesssim D^{\frac{2s\beta+2-\beta}{2s\beta+2}}, \\ (B/D)^{2s}, & B \gtrsim D^{\frac{2s\beta+2-\beta}{2s\beta+2}}. \end{cases}$$

*Proof.* We start from the loss bound in the fixed-data regime (from Theorem G.1),

$$\mathcal{E}_D(\tilde\eta) \;\asymp\; (\tilde\eta D)^{-2s} + \begin{cases} \tilde\eta, & \beta > 2, \\ \tilde\eta \log(\tilde\eta D), & \beta = 2, \\ \tilde\eta(\tilde\eta D)^{2/\beta - 1}, & 1 < \beta < 2, \end{cases}$$

with $\tilde\eta \coloneqq \eta/\sqrt{B}$.

We distinguish regimes based on whether the stability constraint $\eta\sqrt{B} = \tilde\eta B \lesssim 1$ is active.

*Case 1: $\beta > 2$, small batch $B \lesssim D^{2s/(2s+1)}$.* Balancing $(\tilde\eta D)^{-2s} \sim \tilde\eta$ gives

$$\tilde\eta^* \;\asymp\; D^{-\frac{2s}{2s+1}}, \qquad \mathcal{E}^*_{\text{sgn}}(D, B) \;\asymp\; D^{-\frac{2s}{2s+1}}.$$

Feasibility requires $\tilde\eta B \lesssim 1 \implies B \lesssim D^{\frac{2s}{2s+1}}$.

*Case 2: $\beta = 2$, small batch $B \lesssim D^{\frac{2s}{2s+1}} (\log D)^{\frac{1}{2s+1}}$.* Balancing $(\tilde\eta D)^{-2s} \sim \tilde\eta \log(\tilde\eta D)$ gives

$$\tilde\eta^* \;\asymp\; D^{-\frac{2s}{2s+1}} (\log D)^{-\frac{1}{2s+1}}, \qquad \mathcal{E}^*_{\text{sgn}}(D, B) \;\asymp\; (D/\log D)^{-2s/(2s+1)}.$$

*Case 3: $1 < \beta < 2$, small batch $B \lesssim D^{\frac{2s\beta+2-\beta}{2s\beta+2}}$.* Balancing $(\tilde\eta D)^{-2s} \sim \tilde\eta(\tilde\eta D)^{2/\beta - 1} = \tilde\eta^{2/\beta} D^{2/\beta - 1}$ gives

$$\tilde\eta^* \;\asymp\; D^{-\frac{2s\beta+2-\beta}{2s\beta+2}}, \qquad \mathcal{E}^*_{\text{sgn}}(D, B) \;\asymp\; D^{-\frac{s\beta}{s\beta+1}}.$$

In all cases, if $B$ exceeds the threshold above, the stability constraint $\tilde\eta B \lesssim 1$ becomes active. Then the effective learning rate saturates at

$$\tilde\eta^* \;\asymp\; 1/B, \qquad \mathcal{E}^*_{\text{sgn}}(D, B) \;\asymp\; (B/D)^{2s}.$$

$\square$

**Theorem G.4** (Fixed data budget scaling for SGD). *Fix a data budget $D = KB$. Assume the stability condition $\eta \lesssim 1$ holds for SGD. Then, optimally tuning the learning rate $\eta$, the optimal loss of SGD satisfies*

$$\mathcal{E}^*_{\text{sgd}}(D, B) \;\asymp\; \begin{cases} D^{-\frac{s}{s+1}}, & B \lesssim D^{\frac{s}{s+1}}, \\ (B/D)^{s}, & B \gtrsim D^{\frac{s}{s+1}}. \end{cases}$$

*Proof.* For SGD, the loss satisfies:

$$h_{\text{sgd}}(\eta, B, D) \asymp (\eta D/B)^{-s} + \eta/B.$$

Balancing the two terms gives

$$(\eta D/B)^{-s} \asymp \eta/B \implies \eta^* \asymp D^{-s/(s+1)}B.$$

Substituting back into the loss, we obtain

$$\mathcal{E}^*_{\text{sgd}}(D, B) \asymp D^{-\frac{s}{s+1}}.$$

This choice is feasible as long as $\eta^* \lesssim 1$.

If $B \gtrsim D^{s/(s+1)}$, the stability condition $\eta \lesssim 1$ becomes active, so $\eta^* \asymp 1$. Plugging into the loss gives

$$\mathcal{E}^*_{\text{sgd}}(D, B) \asymp (B/D)^s.$$

$\square$

### G.2.2 FIXED $K$ SETTING

We investigate the role of batch size in a complementary setting, where the number of iterations $K$ is fixed while the batch size $B$ varies. We denote the optimal loss as $\mathcal{E}^*_{\text{sgn}}(K, B)$ and $\mathcal{E}^*_{\text{sgd}}(K, B)$ respectively.

**Theorem G.5** (Fix training steps). *Suppose $\beta < \alpha$. For a fixed step K, if we optimally tune learning rate $\eta$ within the stability region. Then we have*

- *For SGD*

$$\mathcal{E}^*_{\text{sgd}}(K, B) = \begin{cases} (BK)^{-\frac{s}{s+1}}, & B < K^s, \\ K^{-s}, & B > K^s. \end{cases}$$

- *For SignSGD*

  - *If $\beta > 2, s < 1$*

$$\mathcal{E}^*_{\text{sgn}}(K, B) = \begin{cases} (BK)^{-\frac{2s}{2s+1}}, & B < K^{2s}, \\ K^{-2s}, & K^{2s} < B < K^2. \end{cases}$$

  - *If $\beta > 2, s > 1$*

$$\mathcal{E}^*_{\text{sgn}}(K, B) = \begin{cases} (BK)^{-\frac{2s}{2s+1}}, & B < K^{\frac{2}{2s-1}}, \\ \dfrac{1}{K\sqrt{B}}, & K^{\frac{2}{2s-1}} < B < K^2. \end{cases}$$

  - *If $1 < \beta < 2, s < \frac{3}{2} - \frac{1}{\beta}$*

$$\mathcal{E}^*_{\text{sgn}}(K, B) = \begin{cases} (BK)^{-\frac{s\beta}{s\beta+1}}, & B < K^{2s+\frac{2}{\beta}-1}, \\ K^{-2s}, & K^{2s+\frac{2}{\beta}-1} < B < K^2. \end{cases}$$

  - *If $1 < \beta < 2, s > \frac{3}{2} - \frac{1}{\beta}$*

$$\mathcal{E}^*_{\text{sgn}}(K, B) = \begin{cases} (BK)^{-\frac{s\beta}{s\beta+1}}, & B < K^{\frac{\beta}{s\beta-\beta+1}}, \\ K^{-1}B^{\frac{1}{\beta}-1}, & K^{\frac{\beta}{s\beta-\beta+1}} < B < K^2. \end{cases}$$

*Proof for SGD in Theorem G.5.* For SGD, the expected excess risk satisfies

$$\mathcal{E}_K \ \asymp \ (\eta K)^{-s} + \frac{\eta}{B},$$

where the only constraint on the learning rate is $\eta \lesssim 1$.

Balancing the two terms, $(\eta K)^{-s} \asymp \frac{\eta}{B}$ we have

$$\eta^* \ \asymp \ B^{\frac{1}{s+1}} K^{-\frac{s}{s+1}}, \qquad \mathcal{E}^* \ \asymp \ (BK)^{-\frac{s}{s+1}}.$$

Since we have $\eta \lesssim 1$, substituting $\eta^*$ gives $B \lesssim K^s$.

If $B \lesssim K^s$, the unconstrained optimum is feasible and

$$\mathcal{E}^*_{\text{sgd}}(K, B) \ \asymp \ (BK)^{-\frac{s}{s+1}}.$$

If $B \gtrsim K^s$, the constraint $\eta \lesssim 1$ becomes active. Taking $\eta \asymp 1$ yields

$$\mathcal{E}^*_{\text{sgd}}(K, B) \ \asymp \ K^{-s}.$$

This completes the proof for SGD. $\qquad\qquad\qquad\qquad\qquad\qquad\qquad\qquad\qquad\qquad$ $\square$

*Proof for SignSGD in Theorem G.5.* Now we come to the proof for SignSGD. To begin with, we prove the case $\beta > 2$. We first need to mention the conditions $\eta\sqrt{B} \lesssim 1$ and $\eta K \gtrsim 1$, which together imply a natural regime $B \lesssim K^2$.

If $\eta$ is set freely, balancing the two terms $(\eta\sqrt{B}K)^{-2s} \asymp \frac{\eta}{\sqrt{B}}$ yields

$$\eta^* = B^{\frac{1-2s}{4s+2}} K^{-\frac{2s}{2s+1}}, \qquad \mathcal{E}^* = (BK)^{-\frac{2s}{2s+1}}.$$

We now check whether the optimal choice $\eta^*$ violates the constraints $\eta\sqrt{B} \lesssim 1$ and $\eta K \gtrsim 1$.

**Condition 1:** $\eta\sqrt{B} \lesssim 1$. Substituting $\eta^*$ yields

$$\eta^* B \ \asymp \ B^{\frac{1}{2s+1}} K^{-\frac{2s}{2s+1}}.$$

Thus, the condition $\eta^*\sqrt{B} \lesssim 1$ is equivalent to

$$B \ \lesssim \ K^{2s}.$$

**Condition 2:** $\eta K \gtrsim 1$. Similarly,

$$\eta^* K \ \asymp \ B^{\frac{1-2s}{4s+2}} K^{\frac{1}{2s+1}}.$$

The validity of this condition depends on the value of $s$.

*Case 1:* $s < \frac{1}{2}$. In this case, the exponent $\frac{1-2s}{4s+2}$ is non-negative, we have $\eta^* K \gtrsim 1$ automatically. Therefore, the only possible active constraint is $B \lesssim K^{2s}$. If this condition holds, the unconstrained optimum is feasible and

$$\mathcal{E}^*_{\text{sgn}}(K, B) \ \asymp \ (BK)^{-\frac{2s}{2s+1}}.$$

Otherwise, when $K^{2s} \lesssim B \lesssim K^2$, the constraint $\eta\sqrt{B} \lesssim 1$ becomes active. Taking $\eta \asymp B^{-1}$ yields

$$\mathcal{E}^*_{\text{sgn}}(K, B) \ \asymp \ K^{-2s}.$$

*Case 2:* $\frac{1}{2} < s < 1$. In this regime, the exponent $\frac{1-2s}{4s+2}$ is negative, so the condition $\eta^* K \gtrsim 1$ yields an additional constraint

$$B \ \lesssim \ K^{\frac{2}{2s-1}}.$$

Since $s < 1$, we have $2s < \frac{2}{2s-1}$, and hence $K^{2s} \lesssim K^{\frac{2}{2s-1}}$. Therefore, the dominant constraint is still $B \lesssim K^{2s}$. The resulting behavior is identical to Case 1:

$$\mathcal{E}^*_{\text{sgn}}(K, B) \asymp \begin{cases} (BK)^{-\frac{2s}{2s+1}}, & B \lesssim K^{2s}, \\ K^{-2s}, & K^{2s} \lesssim B \lesssim K^2. \end{cases}$$

*Case 3: $s > 1$.* In this case, the exponent $\frac{1-2s}{4s+2}$ is negative, and the condition $\eta^* K \gtrsim 1$ imposes a stricter constraint

$$B \lesssim K^{\frac{2}{2s-1}}.$$

When $B \lesssim K^{\frac{2}{2s-1}}$, both constraints are satisfied and

$$\mathcal{E}_{\mathrm{sgn}}^*(K, B) \asymp (BK)^{-\frac{2s}{2s+1}}.$$

When $K^{\frac{2}{2s-1}} \lesssim B \lesssim K^2$, the constraint $\eta K \gtrsim 1$ is violated, and the optimal learning rate is given by $\eta \asymp K^{-1}$. Substituting this choice yields

$$\mathcal{E}_{\mathrm{sgn}}^*(K, B) \asymp \frac{1}{K\sqrt{B}}.$$

This completes the proof for the case $\beta > 2$.

Now we prove the case $1 < \beta < 2$. We follow the same strategy as in the case $\beta > 2$.

Balancing the signal and noise terms in the scaling law, $(\eta\sqrt{B}K)^{-2s} \asymp \frac{\eta}{\sqrt{B}}(\eta\sqrt{B}K)^{\frac{2}{\beta}-1}$, we get the unconstrained optimal learning rate

$$\eta^* \asymp B^{-\frac{1}{2}+\frac{\beta}{2s\beta+2}} K^{-1+\frac{\beta}{2s\beta+2}}, \qquad \mathcal{E}^* \asymp (BK)^{-\frac{s\beta}{s\beta+1}}.$$

We now check whether the optimal choice $\eta^*$ violates the constraints $\eta\sqrt{B} \lesssim 1$ and $\eta K \gtrsim 1$.

**Condition 1: $\eta\sqrt{B} \lesssim 1$.** Substituting $\eta^*$ yields

$$\eta^*\sqrt{B} \asymp B^{\frac{\beta}{2s\beta+2}} K^{-1+\frac{\beta}{2s\beta+2}}.$$

Thus, the condition $\eta^*\sqrt{B} \lesssim 1$ is equivalent to

$$B \lesssim K^{2s+\frac{2}{\beta}-1}.$$

**Condition 2: $\eta K \gtrsim 1$.** Similarly,

$$\eta^* K \asymp B^{-\frac{1}{2}+\frac{\beta}{2s\beta+2}} K^{\frac{\beta}{2s\beta+2}}.$$

The validity of this condition depends on the value of $s, \beta$.

*Case 1: $s < 1 - \frac{1}{\beta}$.*

In this case, $-\frac{1}{2} + \frac{\beta}{2s\beta+2} \geq 0$, and hence the exponent of $B$ in $\eta^* K$ is non-negative. Since $B \lesssim K^2$, the condition $\eta^* K \gtrsim 1$ holds automatically. Therefore, the only possible active constraint is $\eta\sqrt{B} \lesssim 1$.

If $B \lesssim K^{2s+\frac{2}{\beta}-1}$, the unconstrained optimum is feasible and

$$\mathcal{E}_{\mathrm{sgn}}^*(K, B) \asymp (BK)^{-\frac{s\beta}{s\beta+1}}.$$

Otherwise, when $K^{2s+\frac{2}{\beta}-1} \lesssim B \lesssim K^2$, the constraint $\eta\sqrt{B} \lesssim 1$ becomes active. Taking $\eta \asymp B^{-1}$ yields

$$\mathcal{E}_{\mathrm{sgn}}^*(K, B) \asymp K^{-2s}.$$

*Case 2: $1 - \frac{1}{\beta} < s < \frac{3}{2} - \frac{1}{\beta}$.*

In this regime, the exponent $-\frac{1}{2} + \frac{\beta}{2s\beta+2}$ is negative, and hence the condition $\eta^* K \gtrsim 1$ imposes an additional constraint

$$B \lesssim K^{\frac{\beta}{s\beta-\beta+1}}.$$

Since $2s + \frac{2}{\beta} - 1 < \frac{\beta}{s\beta-\beta+1}$, the dominant constraint is still $\eta\sqrt{B} \lesssim 1$, namely $B \lesssim K^{2s+\frac{2}{\beta}-1}$.

Therefore, the resulting behavior is identical to Case 1:

$$
\mathcal{E}^*_{\mathrm{sgn}}(K, B) \approx
\begin{cases}
(BK)^{-\frac{s\beta}{s\beta+1}}, & B \lesssim K^{2s+\frac{2}{\beta}-1}, \\
K^{-2s}, & K^{2s+\frac{2}{\beta}-1} \lesssim B \lesssim K^2.
\end{cases}
$$

*Case 3: $s > \frac{3}{2} - \frac{1}{\beta}$.*

In this case, the constraint $\eta^* K \gtrsim 1$ becomes more restrictive, namely $B \lesssim K^{\frac{\beta}{s\beta-\beta+1}}$. When this condition holds, both constraints are satisfied and

$$
\mathcal{E}^*_{\mathrm{sgn}}(K, B) \approx (BK)^{-\frac{s\beta}{s\beta+1}}.
$$

When $K^{\frac{\beta}{s\beta-\beta+1}} \lesssim B \lesssim K^2$, the constraint $\eta K \gtrsim 1$ is violated. The optimal learning rate is then $\eta \approx K^{-1}$, yielding

$$
\mathcal{E}^*_{\mathrm{sgn}}(K, B) \approx K^{-1} B^{\frac{1}{\beta}-1}.
$$

This completes the proof for the case $1 < \beta < 2$. □

A corresponding experiment is shown in Figure 12. As $B$ increases, the SignSGD loss decreases more rapidly, and its saturation point—beyond which larger batches bring little additional benefit—occurs at a substantially larger $B$ than for SGD.

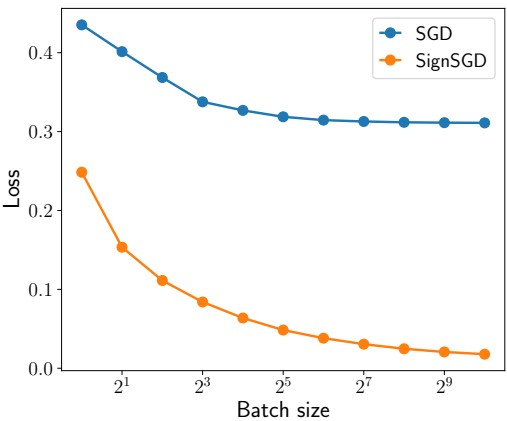

Figure 12: Large batch training when fixed $K$.

# H    OTHER PROOFS

## H.1    BURN-IN TIME

In this section we prove Lemma F.1 in the main text and provide a detailed analysis of the burn-in time of the SignSGD update.

The one-dimensional stochastic dynamics

$$
\delta^{(j)}_{k+1} = \delta^{(j)}_k - \eta \, \mathrm{sgn}\big(\lambda_j \delta^{(j)}_k + \xi^{(j)}_k\big),
$$

where $\delta^{(j)}_0 = -\theta^*_j$ and $\xi^{(j)}_k \sim \mathcal{N}(0, \frac{\lambda_j}{B})$ are i.i.d. Recall that Assumption B.3 implies the power-law decay

$$
\lambda_j \approx j^{-\beta}, \qquad \theta^*_j \approx j^{\frac{\beta-\alpha}{2}}.
$$

Define the linear region

$$
r_j := \frac{1}{\sqrt{\lambda_j B}}, \qquad \mathcal{R}_j = [-r_j, r_j],
$$

and the burn-in time

$$k_j^* = \inf\{k \in \mathbb{N} \mid \delta_k^{(j)} \in \mathcal{R}_j\}.$$

We first show that for sufficiently large $j$, the dynamics is initialized in the linear region.

**Lemma H.1.** *For $j \gtrsim B^{1/\alpha}$, we have $k_j^* = 0$.*

*Proof.* Using $\lambda_j \asymp j^{-\beta}$ and $\theta_j^* \asymp j^{(\beta-\alpha)/2}$, we obtain

$$|\theta_j^*|\sqrt{\lambda_j B} \asymp j^{(\beta-\alpha)/2} \cdot j^{-\beta/2} \cdot \sqrt{B} = \frac{\sqrt{B}}{j^{\alpha/2}}.$$

Thus for $j \gtrsim B^{1/\alpha}$ we have $|\theta_j^*|\sqrt{\lambda_j B} \lesssim 1$, i.e. $|\theta_j^*| \lesssim r_j$. Hence $\delta_0^{(j)} = -\theta_j^* \in \mathcal{R}_j$ and $k_j^* = 0$. $\qquad\square$

We now analyze the hitting time when the initialization is outside $\mathcal{R}_j$. We will use the following classical concentration inequality for martingales.

**Lemma H.2** (Azuma's inequality (Theorem 7.2.1 in Alon & Spencer (2016))). *Let $X_0, \ldots, X_m$ be a martingale with $X_0 = 0$ and $|X_{i+1} - X_i| \leq 1$ for all $0 \leq i < m$. Then it holds for any $\lambda > 0$ that*

$$\mathbb{P}[X_m > \lambda\sqrt{m}] < e^{-\lambda^2/2}.$$

The next lemma is stated in terms of the intrinsic scale $r_j$ and will directly imply Lemma F.1.

**Lemma H.3.** *Fix $j$ and define $r_j := 1/\sqrt{\lambda_j B}$. Assume*

$$|\theta_j^*| > r_j, \qquad \eta \leq r_j.$$

*Define*

$$m_j := \left\lceil \frac{|\theta_j^*| - r_j}{\eta} \right\rceil.$$

*Then $k_j^* \geq m_j$ deterministically, and with probability at least*

$$1 - \exp\left(-\frac{3}{16} m_j\right),$$

*it holds that*

$$k_j^* \leq 6\, m_j.$$

*In particular, under $|\theta_j^*| \geq 2r_j$ we have $m_j \asymp |\theta_j^*|/\eta$ and hence*

$$\mathbb{P}\left(k_j^* \asymp \frac{|\theta_j^*|}{\eta}\right) \geq 1 - \exp\left(-\frac{3}{32} \frac{|\theta_j^*|}{\eta}\right).$$

*Proof.* We omit the subscript $j$ when there is no ambiguity. Write $\delta_k := \delta_k^{(j)}$, $\lambda := \lambda_j$, and $r := r_j = 1/\sqrt{\lambda B}$. On the event $\{k < k_j^*\}$ we have $|\delta_k| > r$ and therefore

$$\mathbb{P}(\mathrm{sgn}(\lambda\delta_k + \xi_k) = \mathrm{sgn}(\delta_k)\,|\,\mathcal{F}_k) = \Phi\left(|\delta_k|\sqrt{\lambda B}\right) \geq \Phi(1),$$

where $\mathcal{F}_k$ is the history up to time $k$. Define

$$I_k := \mathbf{1}\left\{\mathrm{sgn}(\lambda\delta_k + \xi_k) = \mathrm{sgn}(\delta_k)\right\}, \qquad W_k := 2I_k - 1 \in \{-1, +1\}.$$

Then for all $k < k_j^*$,

$$\mathbb{E}[W_k \mid \mathcal{F}_k] = 2\,\mathbb{P}(I_k = 1 \mid \mathcal{F}_k) - 1 \geq 2\Phi(1) - 1 =: \mu_0 > \frac{2}{3}.$$

Since $\eta \leq r$ and $|\delta_k| > r$ for $k < k_j^*$, the sign of $\delta_k$ cannot flip in one update before entering $\mathcal{R}_j$. Consequently, for all $k < k_j^*$,

$$|\delta_{k+1}| = \begin{cases} |\delta_k| - \eta, & W_k = +1, \\ |\delta_k| + \eta, & W_k = -1, \end{cases}$$

or equivalently $|\delta_{k+1}| = |\delta_k| - \eta W_k$. Iterating up to any time $t \leq k_j^*$ yields

$$|\delta_t| = |\delta_0| - \eta \sum_{k=0}^{t-1} W_k = |\theta_j^*| - \eta \sum_{k=0}^{t-1} W_k.$$

Therefore, if $\sum_{k=0}^{t-1} W_k \geq (|\theta_j^*| - r)/\eta$, then $|\delta_t| \leq r$ and hence $k_j^* \leq t$. On the other hand, each step can decrease $|\delta_k|$ by at most $\eta$, so necessarily $t \geq \lceil (|\theta_j^*| - r)/\eta \rceil =: m_j$, which implies the deterministic lower bound $k_j^* \geq m_j$.

For the upper tail, define

$$D_k := W_k - \mathbb{E}[W_k \mid \mathcal{F}_k], \qquad M_t := \sum_{k=0}^{t-1} D_k.$$

Then $(M_t)_{t \geq 0}$ is a martingale and $|D_k| \leq 2$ almost surely. Moreover, for any $t$,

$$\sum_{k=0}^{t-1} W_k = \sum_{k=0}^{t-1} \mathbb{E}[W_k \mid \mathcal{F}_k] + M_t \geq \mu_0 t + M_t \geq \frac{2}{3}t + M_t.$$

Let $t := 6m_j$. Then $2t/3 = 4m_j$, and hence

$$\mathbb{P}\left(\sum_{k=0}^{t-1} W_k < m_j\right) \leq \mathbb{P}(M_t < m_j - 4m_j) = \mathbb{P}(M_t \leq -3m_j).$$

Set $Y_s := M_s/2$. Then $(Y_s)_{s \geq 0}$ is a martingale with $Y_0 = 0$ and $|Y_s - Y_{s-1}| \leq 1$. Applying Lemma H.2 to the martingale $(-Y_s)_{s \geq 0}$ with $\lambda := \frac{3m_j}{2\sqrt{t}}$ gives

$$\mathbb{P}(M_t \leq -3m_j) = \mathbb{P}\left(-Y_t \geq \frac{3m_j}{2}\right) < \exp\left(-\frac{\lambda^2}{2}\right) = \exp\left(-\frac{9m_j^2}{8t}\right) = \exp\left(-\frac{3}{16}m_j\right).$$

Thus with probability at least $1 - \exp(-3m_j/16)$ we have $\sum_{k=0}^{t-1} W_k \geq m_j$, hence $k_j^* \leq t = 6m_j$.

Finally, under $|\theta_j^*| \geq 2r_j$ we have

$$\frac{|\theta_j^*|}{2\eta} \leq \frac{|\theta_j^*| - r_j}{\eta} \leq \frac{|\theta_j^*|}{\eta},$$

so $m_j \asymp |\theta_j^*|/\eta$, and also

$$\exp\left(-\frac{3}{16}m_j\right) \leq \exp\left(-\frac{3}{32}\frac{|\theta_j^*|}{\eta}\right),$$

which yields the desired probability bound.

$\square$

**Proof of Lemma F.1** Lemma F.1 assumes $1 \leq j \lesssim B^{1/\alpha}$ and $\eta\sqrt{\lambda_j B} \lesssim 1$. Since $r_j = 1/\sqrt{\lambda_j B}$, the second condition is equivalent to $\eta \lesssim r_j$, and in particular $\eta \leq r_j$ up to adjusting universal constants.

Next, using $\lambda_j \asymp j^{-\beta}$ and $\theta_j^* \asymp j^{(\beta-\alpha)/2}$,

$$\frac{|\theta_j^*|}{r_j} = |\theta_j^*|\sqrt{\lambda_j B} \asymp \frac{\sqrt{B}}{j^{\alpha/2}}.$$

Hence, for $1 \leq j \lesssim B^{1/\alpha}$, the ratio $|\theta_j^*|/r_j$ is bounded below by a positive constant, and in particular $|\theta_j^*| \geq 2r_j$ (again up to universal constants). Therefore the assumptions of Lemma H.3 hold, and it yields

$$\mathbb{P}\left(k_j^* \asymp \frac{|\theta_j^*|}{\eta}\right) \geq 1 - \exp\left(-\Omega\left(\frac{|\theta_j^*|}{\eta}\right)\right),$$

which concludes the proof of Lemma F.1.

## H.2 OSCILLATION

In this section we make a detailed analysis of the oscillation in the large-learning-rate regime, i.e., when $\eta\sqrt{\lambda_j B} \gtrsim 1$. In this case, the recurrence in the linear region (23) does not converge. So the dynamics cannot be stabilized in the linear region and will quickly escape. We will show that the trajectory does not escape very far. Instead, it oscillates around the scale $\Theta(\eta)$.

Fix an index $j$ and abbreviate $\lambda := \lambda_j$ and $\theta^* := \theta_j^*$. Let $\delta_k := \theta_k^{(j)} - \theta^*$ denote the $j$-th coordinate error. In the linearized regime, the update takes the form

$$\delta_{k+1} = \delta_k - \eta \, \mathrm{sgn}(\lambda\delta_k + \xi_k), \qquad \xi_k \overset{i.i.d.}{\sim} \mathcal{N}\left(0, \frac{\lambda}{B}\right).$$

Assume the large-step regime

$$\eta\sqrt{\lambda B} \geq 1.$$

Recall the burn-in time

$$k_j^{**} := \inf\{k \geq 0 : |\delta_k| \leq \eta\}.$$

**Proof of Lemma F.2.** If $|\theta^*| \leq \eta$, then $|\delta_0| = |\theta^*| \leq \eta$ and hence $k_j^{**} = 0$. Otherwise define

$$m := \left\lceil \frac{|\theta^*| - \eta}{\eta} \right\rceil.$$

Then $k_j^{**} \geq m$ deterministically, since $|\delta|$ can shrink by at most $\eta$ per step before entering $[-\eta, \eta]$.

On the event $\{k < k_j^{**}\}$ we have $|\delta_k| > \eta$, so one update of size $\eta$ cannot cross $0$ before the process enters $[-\eta, \eta]$. Moreover,

$$\mathbb{P}\big(\mathrm{sgn}(\lambda\delta_k + \xi_k) = \mathrm{sgn}(\delta_k) \,\big|\, \mathcal{F}_k\big) = \Phi\big(|\delta_k|\sqrt{\lambda B}\big) \geq \Phi\big(\eta\sqrt{\lambda B}\big) \geq \Phi(1),$$

where we used $\eta\sqrt{\lambda B} \geq 1$. Thus while $k < k_j^{**}$, the process $|\delta_k|$ dominates a biased nearest-neighbor walk on the lattice $\{|\theta^*| - \ell\eta : \ell \in \mathbb{Z}_{\geq 0}\}$ that steps *toward* $0$ with probability at least $p_0 := \Phi(1)$. A standard Chernoff/Hoeffding bound (identical to the argument in Appendix H.1) then yields

$$\mathbb{P}\big(k_j^{**} \leq 6m\big) \geq 1 - \exp\left(-\frac{3}{16}m\right), \qquad \text{equivalently} \qquad \mathbb{P}(k_j^{**} > 6m) \leq \exp(-3m/16).$$

Since $m = \Theta(|\theta^*|/\eta)$ whenever $|\theta^*| > \eta$, we conclude that with probability at least $1 - \exp\big(-\Omega(|\theta_j^*|/\eta)\big)$,

$$k_j^{**} \in [m, 6m] \quad \Longrightarrow \quad k_j^{**} \asymp |\theta_j^*|/\eta.$$

This proves Lemma F.2. $\qquad\qquad\square$

**Theorem H.4.** *Suppose $\eta\sqrt{\lambda B} \geq 1$. There exists a universal constant $C < \infty$ such that*

$$\sup_{t \geq 0} \mathbb{E}\left[\delta_{k_j^{**}+t}^2\right] \leq C\eta^2.$$

*Proof.* Let $p_0 := \Phi(1)$ and $q_0 := \Phi(-1) = 1 - p_0$. Define

$$a := \frac{1}{2}\log\frac{p_0}{q_0} > 0, \qquad \psi := p_0 e^{-a} + q_0 e^a = 2\sqrt{p_0 q_0} \in (0,1), \qquad W(x) := \exp\left(a\frac{|x|}{\eta}\right).$$

Denote $s = \sqrt{\lambda B}$. If $|\delta_k| \geq \eta$, then $|\delta_{k+1}| = |\delta_k| \pm \eta$ and

$$\mathbb{P}\big(|\delta_{k+1}| = |\delta| - \eta \mid \delta_k = \delta\big) = \Phi(s|\delta|), \qquad \mathbb{P}\big(|\delta_{k+1}| = |\delta| + \eta \mid \delta_k = \delta\big) = \Phi(-s|\delta|) = 1 - \Phi(s|\delta|).$$

Moreover, by $\rho = \eta s \geq 1$ and $|\delta| \geq \eta$, we have $\Phi(s|\delta|) \geq \Phi(\eta s) \geq \Phi(1) = p_0$. Hence

$$\mathbb{E}[W(\delta_{k+1}) \mid \delta_k = \delta] = W(\delta)\Big(\Phi(s|\delta|)e^{-a} + (1 - \Phi(s|\delta|))e^a\Big) = W(\delta)\,f(\Phi(s|\delta|)),$$

where $f(p) := pe^{-a} + (1-p)e^a$. Since $f$ is decreasing on $[0,1]$ and $\Phi(s|\delta|) \geq p_0$,

$$f(\Phi(s|\delta|)) \leq f(p_0) = p_0 e^{-a} + q_0 e^a = \psi,$$

and thus for all $|\delta| \geq \eta$,

$$\mathbb{E}[W(\delta_{k+1}) \mid \delta_k = \delta] \leq \psi \, W(\delta). \tag{41}$$

On the other hand, if $|\delta_k| \leq \eta$, then $|\delta_{k+1}| \leq |\delta_k| + \eta \leq 2\eta$, hence

$$\mathbb{E}[W(\delta_{k+1}) \mid \delta_k = \delta] \leq e^{2a}.$$

Combining with (41) yields the global inequality

$$\mathbb{E}[W(\delta_{k+1}) \mid \delta_k = \delta] \leq \psi \, W(\delta) + e^{2a}, \qquad \forall \, \delta \in \mathbb{R}.$$

By definition of $k_j^{**}$, we have $|\delta_{k_j^{**}}| \leq \eta$ and thus $W(\delta_{k_j^{**}}) \leq e^a$. Iterating the above equation gives that

$$\mathbb{E}[W(\delta_{k_j^{**}+t})] \leq \psi^t \, \mathbb{E}[W(\delta_{k_j^{**}})] + e^{2a} \sum_{i=0}^{t-1} \psi^i \leq e^a + \frac{e^{2a}}{1-\psi}. \tag{42}$$

Since $\sup_{x \geq 0} x^2 e^{-ax} = 4/(a^2 e^2)$, we have $x^2 \leq (4e^2/a^2) e^{ax}$ for all $x \geq 0$. With $x = |\delta|/\eta$, this implies

$$\delta^2 \leq \frac{4e^2}{a^2} \eta^2 \, W(\delta).$$

Plugging this to (42), we have for all $t \geq 0$ that

$$\mathbb{E}[\delta_{k_j^{**}+t}^2] \leq \frac{4e^2}{a^2} \eta^2 \, \mathbb{E}[W(\delta_{k_j^{**}+t})] \leq \frac{4e^2}{a^2} \eta^2 \left( e^a + \frac{e^{2a}}{1-\psi} \right) =: C \, \eta^2,$$

where $C < \infty$ is a universal constant. This proves $\sup_{t \geq 0} \mathbb{E}[\delta_{k_j^{**}+t}^2] \leq C\eta^2$. $\qquad \square$

**Proof of Proposition F.4.** The upper bound in Proposition F.4 follows immediately from Theorem H.4. For the lower bound, note that $|\delta_{k+1}| = |\delta_k| \pm \eta$. Hence

$$\max\{|\delta_k|, |\delta_{k+1}|\} \geq \frac{\eta}{2}.$$

## H.3 LINEAR RECURRENCE

In this section, we prove Lemma F.5.

*Proof.* The SignSGD update is

$$\delta_{k+1}^{(j)} = \delta_k^{(j)} - \eta \, \mathrm{sgn}(\lambda_j \delta_k^{(j)} + \xi_k^{(j)}).$$

Taking the square,

$$\mathbb{E}[(\delta_{k+1}^{(j)})^2 \mid \delta_k^{(j)}] = (\delta_k^{(j)})^2 + \eta^2 - 2\eta \delta_k^{(j)} \, \mathbb{E}_{\xi_k^{(j)}}[\mathrm{sgn}(\lambda_j \delta_k^{(j)} + \xi_k^{(j)})],$$

For $k > k_j^*$, with Assumption F.3, we have

$$\delta_k^{(j)} \in \mathcal{R}_j = \left[ -\frac{1}{\sqrt{\lambda_j B}}, \frac{1}{\sqrt{\lambda_j B}} \right],$$

and the linear approximation holds:

$$\mathbb{E}_{\xi_k^{(j)}}[\mathrm{sgn}(\lambda \delta_k^{(j)} + \xi_k^{(j)})] \approx \eta \sqrt{\lambda_j B} \, \delta_k^{(j)}$$

Therefore, for $k \geq k_j^*$ we have

$$\mathbb{E}[(\delta_{k+1}^{(j)})^2 \mid k_j^*] = (1 - \Theta(\eta \sqrt{\lambda_j B})) \, \mathbb{E}[(\delta_k^{(j)})^2 \mid k_j^*] + \eta^2.$$

$\qquad \square$

## I CONCLUDING REMARKS

In this paper, we developed a comprehensive scaling-law analysis of SignSGD in feature-space linear regression. Within this framework, we derived explicit scaling laws, explained how and when SignSGD can outperform vanilla SGD, and identified the distinctive roles of coordinate-wise adaptivity and the associated noise-smoothing effect in shaping scaling efficiency.

Extending our scaling framework to different learning rate schedules and other adaptive methods such as Signum and Adam is a natural next step, and we leave a systematic treatment of these extensions to future work.

