# OpenReview forum: "Scaling-Law Analysis of SignSGD: From Feature-Space Linear Regression to LLM Pre-training"
_ICLR.cc/2026/Workshop/Sci4DL — Sci4DL 2026_

### Official Review · Reviewer_eTgH · 2026-02-25

**Fit:** 3
**Significance:** 2
**Confidence:** 2

**Summary:**

The authors provide a scaling law for SignSGD, which they first show in regression tasks, providing theoretical arguments for the scaling law, which they then show transfers to LLM pre-training to find that the same scaling law is fit (almost exactly) for larger scale cases. Provided that scaling laws are of great interest to understand the computational benefits provided by different optimisation strategies, this work is timely and a good fit for the workshop.

**Strengths:**

1. The justification for the experimental setup exploring SGD compared to SignSGD is strong and aligns with experimental standards in this regime.
2. The transition from the regression task to show the scaling law and transition to LLM pre-training to show the same results is impressive.
3. All figured are very clear with verbose captions, which make the results and core takeaways easy to synthesise.
4. Varying hyperparameters such as batch size and learning rate is good to show that the scaling law is robust to hyperparameter changes. It is good that the authors then show that these results enable the accurate prediction of the loss curve with batch size 256.

**Suggestions:**

1. It would be interesting to see how the results scale to even larger batch sizes in the LLM pre-training (>256) to see how the scaling law fits. However, it is appreciated that this incurs increases computational expense, and the scale of experiments conducted is appreciated.
2. In the appendix, there are many missing references, which correspond to LaTeX errors, which makes reviewing some of the statements difficult. It would be good to ensure that all of these are fixed in the revision period.
3. While the figures are clear, it would be good to present the standard error of the mean to understand the error surrounding the mean values presented.
4. It would be interesting to see how these results change when different learning rate schedulers are used and if any fluctuations are captured by the theoretical findings in the paper. While it is by the authors acknowledge that this is future work, it would certainly serve as a worthwhile extension of this work.
5. For the LLM pre-training results, it would be good to provide details on what datasets are used and the parameter sizes of the model explored to ascertain the scale at which the experiment is conducted.

---

### Official Review · Reviewer_Jp5j · 2026-02-25

**Fit:** 3
**Significance:** 2
**Confidence:** 3

**Summary:**

The paper studies SignSGD from a scaling-law perspective, aiming to explain why sign-based optimizers can work well in large-scale training. It uses an analytically tractable setup, feature-space linear regression with power-law spectra and Hessian-aligned mini-batch noise, to derive explicit scaling laws for SignSGD loss dynamics.

**Strengths:**

1. The paper connects a clean theoretical model (power-law linear regression) to practical LLM pretraining behavior, and the main-text experiments are directly tied to the derived scaling predictions.

2. The paper gives a useful qualitative decomposition of batch-size effects in SignSGD, and then test the effects empirically in the LLM experiment.

**Suggestions:**

1. The core results depend on specific assumptions (diagonal Hessian / Hessian-aligned noise). While the paper acknowledges this, the main text could better clarify when these assumptions are expected to hold in real LLM training.

2. The main theorem is presented in an informal version with the full statement/proof deferred to the appendix, and the main text focuses on an “easy” regime. Authors should add a formal theorem in the main text.

---

### Official Review · Reviewer_3UG1 · 2026-02-27

**Fit:** 3
**Significance:** 2
**Confidence:** 2

**Summary:**

The paper develops a scaling-law theory for SignSGD by analyzing feature-space linear regression under power-law spectrum and a diagonal Hessian assumption, yielding a three-regime picture (burn-in, scaling, unstable) governed by the intrinsic time. Empirically, the authors fit a simple parametric form derived from their theory to 100m Llama pretraining experiments and accurately extrapolate later loss dynamics and unseen batch sizes.

**Strengths:**

* The characterization of signSGD across the joint scaling of learning rate and batch size via a phase diagram with three regimes is interesting and yields interpretable scaling signatures.
* The work is transparent about assumptions/limitations (proxy model, regime conditions)
* The experimental setup is sound and directly verifies the theory’s implied bounds/signatures in both controlled settings and a small pretraining setting, where the theory yields a compact parametric form and specific batch scalings that are then used to forecast held-out portions of LLM loss curves and unseen batch sizes

**Suggestions:**

* Do the authors have intuition for how results may differ for more commonly used learning rate schedulers, eg warmup-stable-decay?
* If learning rate is tuned differently across optimizers, can the authors report the LR tuning range for both SGD and SignSGD at each batch size?
Since the theory is explicitly data-limited and predicts B*(D) should shift with token budget D, do the authors test multiple token budgets (e.g., in the commonly adopted 'overtrained' regime) to verify how the inferred critical batch size and plateau behavior change with D?
* (minor) Undefined references throughout Appendix C

---

### Meta-Review · Area_Chair_kZux · 2026-03-02

**Recommendation:** Accept

**Metareview:**

This paper uses feature-space linear regression to develop a scaling-law theory for SignSGD, enabling extrapolation of later loss dynamics from 100m Llama pretraining experiments. The methodology and the findings are a good contribution to the workshop.

---

### Decision · Program_Chairs · 2026-03-02

Accept